# CD32 captures committed haemogenic endothelial cells during human embryonic development

Rebecca Scarfò[1,8], Lauren N. Randolph[1,8], Monah Abou Alezz[1], Mahassen El Khoury[2], Amélie Gersch[2], Zhong-Yin Li[3,4], Stephanie A. Luff [3,4], Andrea Tavosanis[1], Giulia Ferrari Ramondo[1], Sara Valsoni [1], Sara Cascione[1], Emma Didelon[1], Laura Passerini[1], Giada Amodio[1], Chiara Brandas [1], Anna Villa[1,5], Silvia Gregori[1], Ivan Merelli[1,6], Jean-Noël Freund[2,7], Christopher M. Sturgeon [3,4], Manuela Tavian [2] ✉ & Andrea Ditadi [1] ✉

During embryonic development, blood cells emerge from specialized endothelial cells, named haemogenic endothelial cells (HECs). As HECs are rare and only transiently found in early developing embryos, it remains difficult to distinguish them from endothelial cells. Here we performed transcriptomic analysis of 28- to 32-day human embryos and observed that the expression of Fc receptor CD32 (*FCGR2B*) is highly enriched in the endothelial cell population that contains HECs. Functional analyses using human embryonic and human pluripotent stem cell-derived endothelial cells revealed that robust multilineage haematopoietic potential is harboured within CD32+ endothelial cells and showed that 90% of CD32+ endothelial cells are bona fide HECs. Remarkably, these analyses indicated that HECs progress through different states, culminating in *FCGR2B* expression, at which point cells are irreversibly committed to a haematopoietic fate. These findings provide a precise method for isolating HECs from human embryos and human pluripotent stem cell cultures, thus allowing the efficient generation of haematopoietic cells in vitro.

During embryonic development, cells continuously alter their characteristics to specify new fates and generate the entire spectrum of cell types and tissues. Sometimes cell differentiation involves abrupt lineage conversions as in the case of embryonic haematopoiesis. In fact, various experimental models have informed us that, during development, haematopoietic cells are produced from haemogenic endothelial cells (HECs), a specialized subpopulation of embryonic endothelium[1–10]. HECs are thought to generate blood cells via an endothelial-to-haematopoietic transition, which involves considerable transcriptional and morphological changes leading to the identity switch from endothelial cell to blood[1,2,4–7,11]. HECs represent a central element of the distinct haematopoietic developmental programmes. In fact, HECs can be found in the different anatomical locations where haematopoiesis is observed, including the yolk sac (YS),

[1]San Raffaele Telethon Institute for Gene Therapy, IRCCS San Raffaele Scientific Institute, Milan, Italy. [2]Université de Strasbourg, Inserm, IRFAC/UMR-S1113, FHU ARRIMAGE, FMTS, Strasbourg, France. [3]Black Family Stem Cell Institute, Icahn School of Medicine at Mount Sinai, New York, NY, USA. [4]Department of Cell, Developmental and Regenerative Biology, Icahn School of Medicine at Mount Sinai, New York, NY, USA. [5]Institute of Genetic and Biomedical Research, Milan Unit, National Research Council, Milan, Italy. [6]Institute for Biomedical Technologies, National Research Council, Milan, Italy. [7]Present address: INSERM U1256—NGERE, Université de Lorraine, Vandoeuvre-lès-Nancy, France. [8]These authors contributed equally: Rebecca Scarfò, Lauren N. Randolph. ✉e-mail: manuela.tavian@inserm.fr; ditadi.andrea@hsr.it

where lineage-restricted haematopoietic progenitors are generated first, as well as the aorta–gonad–mesonephros (AGM), the site of the Notch-dependent haematopoietic stem cell (HSC) emergence[12–14]. As HECs represent a rare and transient population rapidly generating haematopoietic output, they have been difficult to characterize and therefore little is known about how the emergence of the haematopoietic lineage is orchestrated.

Traditionally, HECs have been isolated and characterized in animal models, using reporters under the control of the regulatory elements of the transcription factors that drive blood cell emergence, such as *Runx1* and *Gfi1* (refs. [15–17]), a strategy that cannot be used to study HECs in human embryos. Our previous data suggested angiotensin-converting enzyme (ACE, also known as CD143) as a potential marker of HSCs and their endothelial precursors in human embryo[18,19]. Recently, transcriptomic analyses have allowed the identification of putative HEC markers in both murine and human embryos, including ACE as well as CXCR4 and CD44 (refs. [20–23]), but these also enrich for arterial endothelial cells (AECs), anatomically associated with HECs[13,14,24]. This hinders the specific characterization of the unique endothelial population that generates blood cells and, consequently, the design of accurate protocols for the derivation of therapeutically relevant haematopoietic cells from human pluripotent stem cells (hPSCs).

In this Article, to overcome these limitations and identify broadly applicable cell-surface markers specific for human HECs both in vivo and in vitro, we performed transcriptomic analysis of endothelial populations displaying haematopoietic potential in the human embryo based on ACE expression. Here, we report that the Fc receptor CD32 is expressed on human embryonic endothelial cells with robust hemogenic potential. Likewise, CD32 allows for the identification of hPSC-derived HECs with higher specificity than other reported HEC markers. This provided an unprecedented opportunity to study how and when HECs initiate the haematopoietic programme. We show that HECs transit through different states and their haematopoietic commitment occurs before the expression of haematopoietic markers and cell cycle genes. In particular, CD32 marks a specific subpopulation of HECs that no longer require NOTCH activation and that are fully committed to haematopoiesis. The ability to capture this rare and rapid embryonic process allows the redefinition of how blood cells emerge in the embryo and will enable the efficient generation of haematopoietic cells in vitro for therapies.

## Results

### ACE identifies distinct human embryonic endothelial subsets

We have previously shown that, within the AGM region of the human embryo, haematopoietic clusters first emerge in the dorsal aorta (DA) at 27 days post-fertilization (dpf; Carnegie Stage (CS) 12)[25]. Human haematopoietic clusters are located on the ventral side of the aortic endothelium and express CD34, a surface marker that they share with the surrounding endothelial cells[25,26]. Intra-aortic CD34[+] haematopoietic clusters also express ACE, another marker that identifies cells with haematopoietic potential in the developing human embryo[18,19]. As ACE expression is also observed in endothelial cells within different haematopoietic sites during human development, including the DA[18,19], we further analysed its expression pattern in the AGM region in parallel with the expression of the transcription factor RUNX1, which regulates the emergence of blood cells in the embryo[15,16]. Immunofluorescence analysis on human embryonic sections revealed that, at 23 dpf (CS10), ACE[+] cells can be found in the mesenchyme surrounding the DA, but not in the aortic wall (Fig. 1a). At this stage, RUNX1 expression cannot be observed in the human AGM region. However, at 27 dpf (CS12), when the first haematopoietic clusters emerge inside the DA, ACE expression is also detected on endothelial cells. This suggests that ACE[+] precursors might migrate from the subaortic mesenchyme to the DA, although the lineage relationship between ACE[+] mesenchymal cells and ACE[+] endothelial cells remains to be elucidated. Strikingly, ACE and RUNX1 expression co-localized in the endothelial cells lining the ventral site of

the DA (Fig. 1a). As such, ACE expression emerges as a potential human HEC marker in vivo, similarly to what is described for murine HECs[22].

We then tested whether ACE can also distinguish subsets of endothelial cells in vitro and track haematopoietic potential in hPSC haematopoietic cultures. For this, we differentiated the human embryonic stem cell (hESC) line H1 using a method that specifies hPSCs into WNT-dependent (WNTd) NOTCH-dependent multipotent *HOXA*[+] HECs, indicative of intra-embryonic AGM-like haematopoiesis[14,27–29]. In this setting, we observed that ACE is expressed by virtually all day 8 CD34[+] cells, including 90 ± 1% of CD34[+]CD43[neg]CD73[neg]CD184[neg] cells that comprise HECs in day 8 WNTd hPSC haematopoietic cultures (Extended Data Fig. 1a,b)[14]. Since HECs represent only ~2% of WNTd CD34[+]CD43[neg]CD73[neg]CD184[neg] cells[14], we concluded that ACE expression is unable to distinguish HECs from other vascular endothelial cells in these hPSC differentiating cultures.

Having established that ACE expression segregates with the haemogenic transcription factor RUNX1 in endothelial cells of the human embryonic DA, we used this surface marker to better characterize embryonic endothelial cells with haematopoietic potential and identify markers that track with HECs both in vivo and in vitro. For this, we isolated CD34[+]CD45[neg] cells on the basis of their ACE expression from the AGM region of four human embryos (E1, E2, E3 and E4) staged between 28 dpf and 32 dpf (CS12–CS13; Extended Data Fig. 1c) and performed whole-transcriptomic analysis using bulk RNA sequencing (RNA-seq). Principal component analysis (PCA) and unsupervised hierarchical clustering by *k*-means highlighted a clear segregation of CD34[+]CD45[neg]ACE[+] and CD34[+]CD45[neg]ACE[neg] (herein ACE[+] and ACE[neg], respectively) transcriptomes (Fig. 1b,c and Extended Data Fig. 1d,e), with 785 differentially expressed genes (DEGs), of which 440 genes were upregulated and 345 were downregulated in ACE[+] (Fig. 1c and Supplementary Table 1). At the population level, both cell fractions expressed high levels of endothelial genes (*CD34*, *CDH5*, *PECAM1* and *TEK*) while genes associated with venous fate (*NR2F2* and *FLRT2*) had a lower expression in ACE[+] cells in comparison with ACE[neg] cells. In contrast, ACE[+] cells showed a higher expression of genes classically associated with arterial cells (*BMX*, *GJA5*, *DLL4*, *CXCR4* and *HEY2*) and HECs (*MYB*, *GFI1* and *CD44*; Fig. 1d). Overrepresentation analysis (ORA) and gene set enrichment analysis (GSEA) revealed that ACE[+] cells are positively associated with cell migration- and cell adhesion-related Gene Ontology (GO) terms, in line with the remodelling occurring during the emergence of blood cells from HECs[30] (Fig. 1e and Supplementary Tables 2–5). Conversely, ACE[+] cells are negatively associated with several cell cycle-related GO terms, suggesting that at a population level they are undergoing an active arterialization process that requires cell growth suppression[31]. These results indicated that, in the aortic endothelium of 5-week human embryos, ACE expression identifies cells showing an arterial gene signature as well as cells expressing genes pivotal for haematopoietic development.

### CD32 is expressed in HECs in human embryos

To identify putative markers for the isolation of HECs, we focused our attention on DEGs coding for cell surface proteins showing higher expression in ACE[+] cells (Fig. 2a, Extended Data Fig. 2a and Supplementary Table 6). We found that *FCGR2B*, which encodes for an isoform of the Fc receptor CD32, ranks among the top ten cell-surface genes whose expression is enriched in ACE[+] cells. Given that CD32 is a marker of other specialized endothelial cells[32–34] and that Fc receptors are expressed together with endothelial markers in a subset of YS-derived haematopoietic progenitors[35], we focused our attention on this gene. We analysed human embryo sections between 26 dpf and 30 dpf (CS12–CS13), which showed that CD32 is expressed together with CD34 in the aortic endothelial cells. CD32[+] endothelial cells localized close to the bifurcation with the vitelline artery (VA) in the AGM region, a site known to contain a high frequency of haematopoietic clusters, and therefore potentially of HECs, at this stage (Fig. 2b)[25]. Remarkably, in the DA, CD32

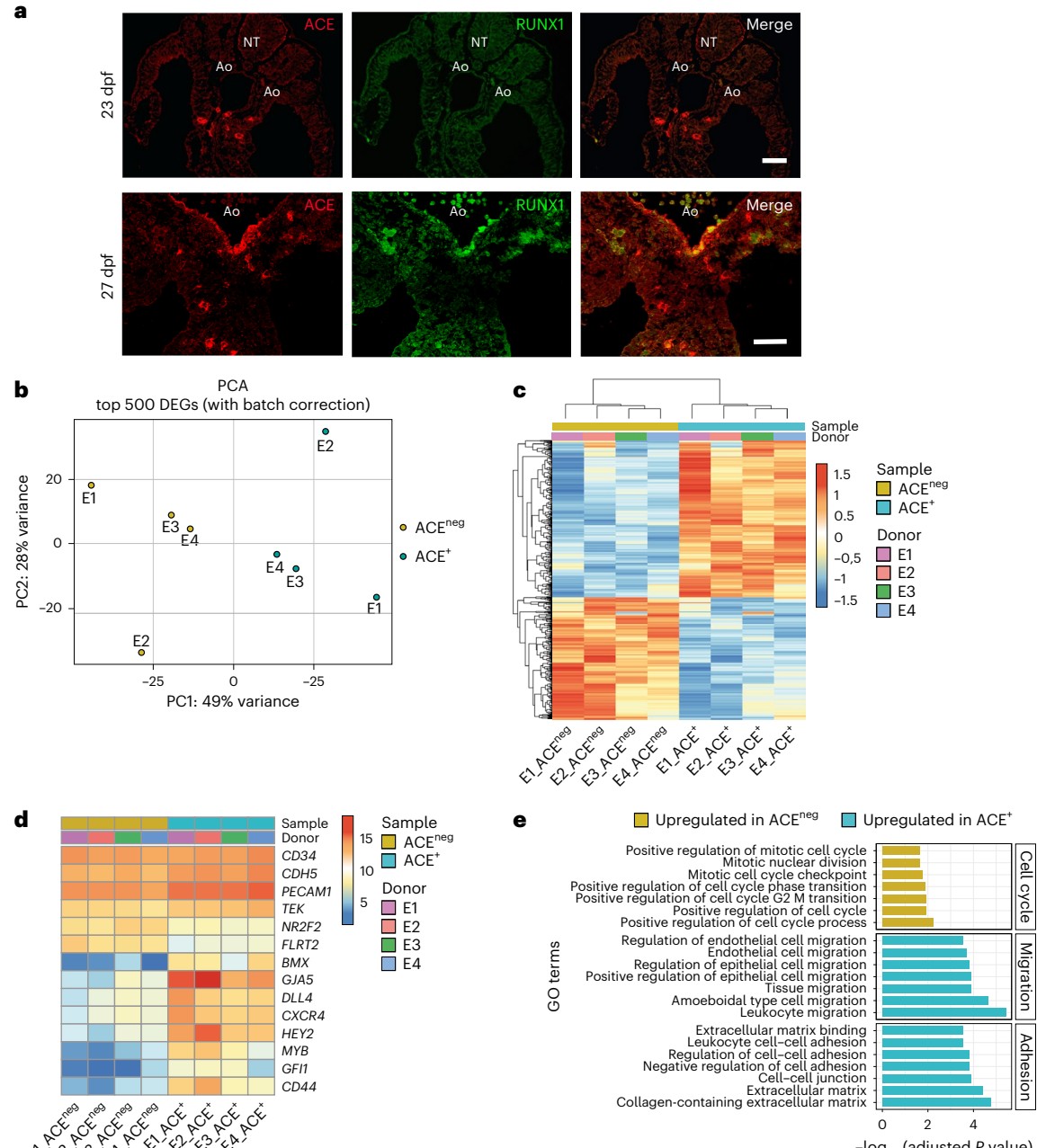

**Fig. 1 | Human embryonic ACE⁺ endothelial cells express arterial and haemogenic markers. a**, A transverse section of the AGM region of a 23 dpf (CS10, top, *n* = 2 independent) and a 27 dpf (CS12, bottom, *n* = 3 independent) human embryo, immunostained with ACE (left, in red), RUNX1 (middle, in green) and merge (right). Ao, aorta; NT, neural tube. Scale bars, 50 µm. **b**–**e**, An RNA-seq analysis of human embryonic populations isolated from four CS12–CS13 embryos, referred to as 'donor': E1, E2, E3 and E4. The ACE^neg population is coloured in beige and ACE⁺ population in light blue. PCA (**b**) of the top 500 DEGs within human embryonic populations. A heatmap of DEGs within human embryonic populations (**c**), where gene counts were corrected for donor and

the rlog gene expression values shown in rows and tiles referring to DEGs are coloured according to upregulation (red) or downregulation (blue). A heatmap of selected pan-endothelial (*CD34*, *CDH5*, *PECAM* and *TEK*), vein-specific (*NR2F2* and *FLRT2*), arterial-specific (*GJA5*, *DLL4*, *CXCR4* and *HEY2*) and haemogenic (*MYB*, *GFI1* and *CD44*) gene expression (**d**), where the rlog gene expression values shown in rows and tiles referring to DEGs are coloured according to upregulation (red) or downregulation (blue). A barplot (**e**) showing significantly enriched GO terms (Fisher exact test; FDR <0.05) using ORA on DEGs. The barplot shows enriched terms grouped by custom categories: cell cycle (upregulated in ACE^neg), migration and adhesion (upregulated in ACE⁺).

marks both the intra-aortic haematopoietic clusters bordering on the ventral site and the underlying endothelial cells. Immunofluorescence analysis on consecutive sections showed that CD32⁺ endothelial cells in the DA co-express CD34, ACE and RUNX1 (Fig. 2c). In the CS12 human embryo, CD32 is also expressed together with CD34 in endothelial cells surrounding the haemogenic regions of the YS (Fig. 2d). This specific expression pattern at active haemogenic embryonic sites led us

to functionally test whether CD32 identifies HECs. Thus, we isolated by fluorescence-activated cell sorting (FACS) the CD32⁺ and CD32^neg fractions of the CD34⁺CD43^negCD45^neg endothelial cell population containing HECs in the AGM and YS regions dissected from two CS13 embryos (E5 and E6) and assessed their potential to generate haematopoietic progenitors ex vivo (Fig. 2e and Extended Data Fig. 2b–d). The CD32⁺ endothelial fraction from both the AGM and YS regions of both

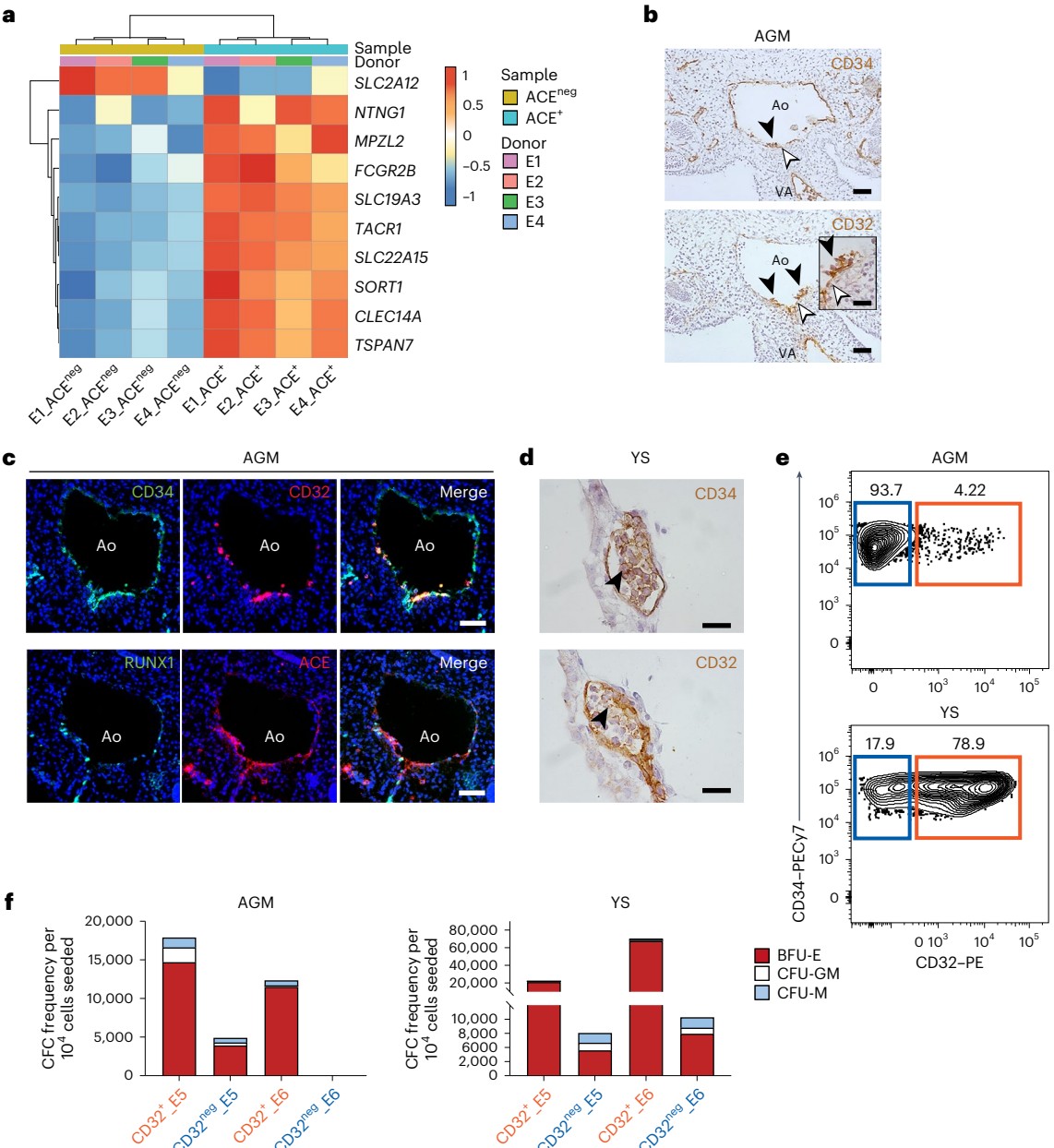

**Fig. 2 | CD32 is expressed in AGM and YS HECs during human embryonic development. a**, A heatmap of top ten differentially expressed surface genes within human embryonic ACE+ (light blue) and ACEneg (beige) cells derived from four CS12–CS13 human embryos (E1, E2, E3 and E4). rlog gene expression values are shown in rows. Colouring indicates differential expression by upregulation (red) or downregulation (blue). **b**, CD34 (top) and CD32 (bottom) expression by immunohistochemistry of consecutive sections of the AGM of a 29 dpf (CS13) human embryo (n = 5 independent). Inset shows high magnification of haematopoietic clusters (black arrowhead) and surrounding endothelial cells (white arrowhead) immunostained by CD32. Ao, aorta. Scale bars, 50 μm and 25 μm (inset). **c**, Transverse consecutive sections of the AGM region of a 29 dpf

(CS13) human embryo, immunostained with CD34 (green), CD32 (red) and merge (top, from left to right) and RUNX1 (green), ACE (red) and merge (bottom, from left to right). Scale bars, 50 μm, n = 3 independent. **d**, CD34 (top) and CD32 (bottom) expression by immunohistochemistry of consecutive sections of the YS of a 26 dpf (CS12) human embryo (n = 5 independent). Scale bars, 25 μm. **e,f**, Flow cytometric analysis (**e**) and quantification of erythro-myeloid CFC potential (**f**) of CD32+ (orange) and CD32neg (blue) cell populations isolated from the AGM and YS of CS13 human embryos (E5 and E6). n = 2, independent. PE, phycoerythrin; PE-Cy7, phycoerythrin-cyanine7; BFU-E, burst-forming unit erythroid; CFU-GM, colony-forming unit granulocyte macrophage; CFU-M, colony-forming unit macrophage.

embryos consistently generated more clonogenic progenitors with erythro-myeloid potential than the CD32neg cells (Fig. 2f). Altogether, these results demonstrate that CD32 can identify a subset of endothelial cells with robust haematopoietic potential in the human embryo.

## CD32 defines intra-embryonic hPSC-derived HECs

Based on our results obtained with the human embryos, we next investigated whether CD32 can be a reliable marker for isolating HECs from

hPSC differentiating cultures. We monitored its expression in WNTd intra-embryonic-like HOXA+ HECs (Extended Data Fig. 3a)[14,27,29,36]. Remarkably, CD32 expression identified only a small subset of the HEC-containing CD34+CD43negCD73negCD184negDLL4neg cell population, which was all ACE+ (Fig. 3a,b and Extended Data Fig. 3b,c). In parallel, using the H9 hESC reporter line to evaluate the expression of RUNX1C, a RUNX1 isoform expressed on WNTd HECs as they begin to generate haematopoietic progenitors[14,27], we observed that CD32 expression in

day 8 WNTd CD34[+]CD43[neg]CD73[neg]CD184[neg]DLL4[neg] cells demarcates endothelial cells that do not express *RUNX1C*–EGFP and thus might identify a population of HECs that have not initiated the haematopoietic programme yet (Fig. 3c,d). To monitor whether this population can generate blood cells, we isolated CD32[+] from CD34[+]CD43[neg]CD73[neg]CD184[neg]DLL4[neg] day 8 WNTd cells and cultured them on Matrigel in the presence of haematopoietic cytokines known to promote and sustain haematopoietic differentiation (HEC culture), as we previously described[14]. Under these conditions, the cells formed an adhesive monolayer that generated haematopoietic progeny as demonstrated by the presence of round cells and of a population of *RUNX1C*–EGFP[+] and CD45[+] cells 5 days later (Fig. 3e,f). As such, the CD32[+] fraction displays the defining behaviour of bona fide HECs.

We therefore assessed the haematopoietic potential of CD34[+]CD43[neg]CD73[neg]CD184[neg]DLL4[neg]CD32[+/neg] (referred to as CD32[+] and CD32[neg]) cell populations (Fig. 3b and Extended Data Fig. 3a,c). The CD32[+] cells generated erythro-myeloid clonogenic progenitors with significantly higher frequency than CD32[neg] cells in both H1 and H9 hESC lines (Fig. 3g), similar to what we showed in the human embryo (Fig. 2f). Given the residual haematopoietic potential observed in CD32[neg] cells isolated from the AGM region and hPSC-derived haematopoietic cultures (Figs. 2f and 3g), we isolated CD32[neg] cells from day 8 WNTd haematopoietic cultures and tested whether they could generate CD32[+] progeny harbouring haematopoietic potential. Indeed, the 2-day-long culture of CD32[neg] cells gave rise to ~40% (39 ± 1%) of CD32[+] cells (Extended Data Fig. 3d) that, when isolated, generated CD45[+] haematopoietic cells in HEC cultures (Extended Data Fig. 3e). These data demonstrate that the CD32[neg] cell population contains precursors of HECs that do not express CD32 yet. This suggests that the residual haematopoietic output observed in the CD32[neg] fraction is due to a further maturation of CD32[neg] into CD32[+] HECs that will then generate haematopoietic cells. We then assessed lymphoid potential by analysing the defining lymphoid lineage for the WNTd haematopoietic programme, that is, T cells[36]. Upon 24 days of co-culture on OP9DLL4 stromal cells, only CD32[+] cells could robustly generate CD4[+]CD8[+] T cells while the CD32[neg] fraction did not display T cell potential under the same conditions (Fig. 3h,i). CD32[+]-derived CD4[+]CD8[+] T cells could progress towards more mature stages and gave rise to CD3[+] cells expressing TCRαβ or TCRγδ and the activation markers CD45RA, CD25 and CD27 (Extended Data Fig. 3f,g). Of note, in contrast to cord blood CD34[+] blood progenitors able to generate only the definitive-restricted Vδ1[+] γδ population, CD32[+] HECs generated different subsets of γδ T cells, including the Vδ1[+] population as well as the Vδ2[+] subset that develops early in the human embryo (Extended Data Fig. 3h)[37]. In addition, CD32[+] HECs displayed the potential to robustly generate CD45[+]CD56[+] natural killer (NK) cells (Extended Data Fig. 3i). Collectively, these results show that CD32[+] cells represent a subpopulation endowed with robust multilineage intra-embryonic-like haematopoietic potential in hPSC cultures.

## CD32 is a specific HEC marker

We next compared the specificity of CD32 as an hPSC-derived HEC marker against CD44, a surface marker often used to define HECs[20,21]. In day 8 WNTd CD34[+]CD43[neg] cells, CD44 is expressed by most DLL4[+] cells (Extended Data Fig. 4a), in line with the reported CD44 expression in AECs[13,20], while it distinguishes two subpopulations of CD34[+]CD43[neg]CD184[neg]CD73[neg]DLL4[neg] cells (referred to as CD44[+] and CD44[neg]) (Extended Data Fig. 4b). We next assessed the haemogenic potential of CD44[+] and CD44[neg] subpopulations and observed that the CD44[+] fraction was enriched for HECs as it generates significantly more clonogenic progenitors than CD44[neg] cells (Fig. 3j), in accordance with recent reports[20,21]. Given that both CD32 and CD44 expression enriches for cells with haemogenic potential, we analysed the relationship between these two markers in hPSC differentiating cultures. Since CD32 identifies a minor subpopulation of CD44[+] cells (Extended Data Fig. 4c) in day 8

WNTd CD34[+]CD43[neg]CD184[neg]CD73[neg]DLL4[neg] cells, we investigated if CD44[+] HECs upregulate CD32 expression to give rise to haematopoietic cells. Kinetic analyses throughout HEC differentiation revealed that CD44[+]CD32[neg] isolated at day 8 of WNTd haematopoietic differentiation begin to express CD32 within 2 days of culture (Extended Data Fig. 4d). Given its expression dynamics, we hypothesized that CD32 might be a more specific marker for HECs than CD44. To compare the specificity of CD32 and CD44, we performed a single-cell HEC assay using CD32[+] or CD44[+] cells isolated at day 8 of WNTd haematopoietic cultures[14]. This clonal analysis revealed that the CD32[+] subfraction was highly enriched for HECs, as 87.0 ± 4.0% of the cells that formed a clone (143/168) in the HEC assay generated exclusively CD45[+] haematopoietic cells (Fig. 3k,l and Extended Data Fig. 4e,f). In stark contrast, the CD44[+] fraction contained equal proportions of progenitors with either haematopoietic or non-haematopoietic potential (Fig. 3k,l). This single-cell analysis indicates that CD32 is a reliable marker for hPSC-derived WNTd HECs, as nearly all CD32[+] cells harbour robust haematopoietic potential. In addition, the use of CD32 yields a significant improvement in enriching for HECs compared with CD44, a marker often used to identify human HECs.

## CD32 identifies HECs in a specific NOTCH-independent state

To further characterize hPSC-derived CD32[+] cells, we next asked if these cells have transcriptional similarity to HECs found in the developing human embryo. We analysed the transcriptomic profile of the CD32[+] fraction sorted from day 8 WNTd haematopoietic cultures and, since HECs in the DA are found in close contact with non-haemogenic cells arterial cells[2], we compared them with CD34[+]CD43[neg]CD184[+]CD73[+]DLL4[+] cells (referred to as DLL4[+]) as control sample[14]. While DLL4[+] cells displayed a significant enrichment for genes whose expression is associated with arterial fate in vivo (for example, *CXCR4*, *DLL4*, *HEY1*, *HEY2*, *SOX17* and *GJA5*)[38], CD32[+] cells were enriched for the expression of genes associated with HECs and their haematopoietic progression in vivo, including *RUNX1*, *GFI1*, *MYCN* and *RAB27B* (Fig. 4a and Supplementary Table 7)[38].

We next contrasted CD32[+] and DLL4[+] cells with human embryonic single-cell RNA sequencing (scRNA-seq) datasets publicly available[20,38], which, integrated, comprise populations of AECs and HECs from AGM and YS regions at different developmental stages spanning from CS10 to CS16 (Fig. 4b). Since CD32[+] cells express *HOXA9* and *HOXA10* similarly to AGM but in contrast to YS cells (Fig. 4c,d), we restricted our comparison to the datasets of CS10–CS16 AGM cells. While DLL4[+] cells resemble more closely CS14–CS16 embryonic AECs, CD32[+] cells exhibited strong similarity to HECs, regardless of the developmental stage (Fig. 4e), thus including those when HSC generation has been observed, that is, CS13–CS16.

We then investigated whether *FCGR2B* expression could further refine the identification of HECs using available human CS14–CS16 AGM scRNA-seq data[38]. In this dataset, while *CDH5*[+]*RUNX1*[+]*PTPRC*[neg]*FCGR2B*[neg] HECs display an enrichment for genes associated with arterial endothelium, the expression of genes characteristic of haematopoietic commitment segregates within *CDH5*[+]*RUNX1*[+]*PTPRC*[neg]*FCGR2B*[+]-expressing cells (Fig. 4f and Extended Data Fig. 5a). These data suggest that transcriptionally distinct states of HECs can be identified.

To assess HEC heterogeneity and interrogate whether *FCGR2B* expression could define a specific state of HECs, we performed scRNA-seq of day 8 WNTd CD34[+]CD43[neg]CD184[neg]CD73[neg] cells as they contain HECs as well as other endothelial progenitors and HEC precursors[14]. Unsupervised clustering revealed 22 transcriptionally distinct clusters which were mostly annotated to the major endothelial cell fates (Fig. 5a, Extended Data Fig. 5b and Supplementary Table 8). This scRNA-seq analysis confirmed that WNTd HECs showed transcriptional heterogeneity as a total of six clusters were enriched for cells differentially expressing *RUNX1* (Fig. 5a, Extended Data Fig. 5b,c and Supplementary Table 8), including one with enriched expression of *FCGR2B* (cluster 11 in Fig. 5a, Extended Data Fig. 5d and Supplementary Table 8).

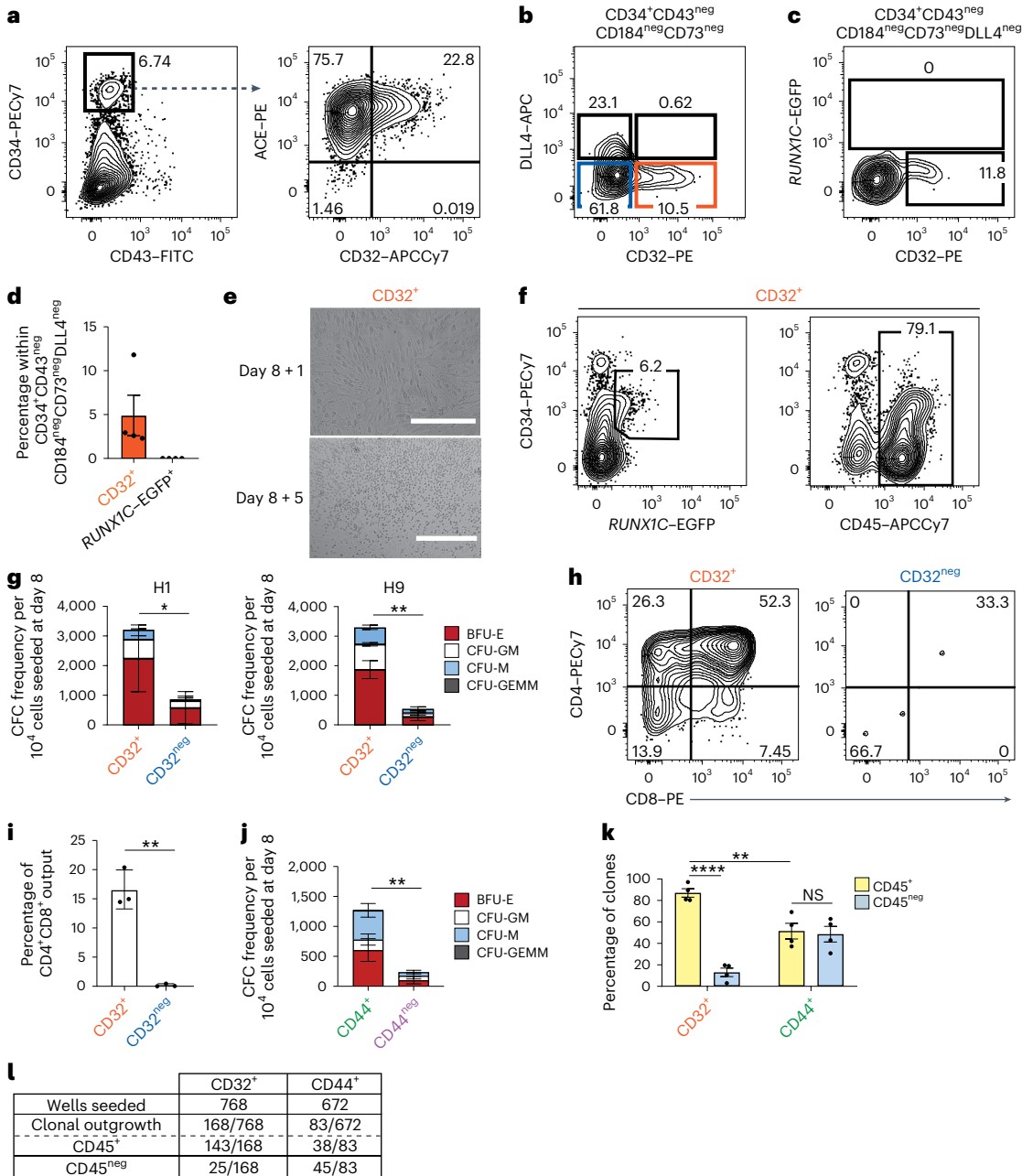

**Fig. 3 | CD32 is expressed in WNTd hPSC-derived HECs. a**, A flow cytometric analysis of day 8 WNTd hPSC-derived haematopoietic cultures. Left: CD43 and CD34 expression within SSC/FSC/Live. Right: CD32 and ACE expression within SSC/FSC/Live/CD34⁺CD43ⁿᵉᵍ. H1 hESCs, *n* = 3, independent. PE-Cy7, phycoerythrin-cyanine7; FITC, fluorescein isothiocyanate; PE, phycoerythrin; APC-Cy7, allophycocyanine-cyanine7. **b**, A flow cytometric analysis at day 8 of WNTd hPSC-derived haematopoietic cultures of CD32 and DLL4 expression within SSC/FSC/Live/CD34⁺CD43ⁿᵉᵍCD184ⁿᵉᵍCD73ⁿᵉᵍ. CD32⁺ is shown in orange and CD32ⁿᵉᵍ in blue. H1 hESCs, *n* = 7, independent. APC, allophycocyanin. **c,d**, Flow cytometric analysis (**c**) and barplot (**d**) showing the frequency of CD32 and *RUNX1C*–EGFP expression in day 8 WNTd hPSC-derived haematopoietic cultures within SSC/FSC/Live/CD34⁺CD43ⁿᵉᵍCD184ⁿᵉᵍCD73ⁿᵉᵍDLL4ⁿᵉᵍ. H9 hESCs, *n* = 4, independent, mean ± s.e.m. **e**, Photo-micrographs of CD32⁺ cells isolated at day 8 of WNTd hPSC-derived haematopoietic culture and after 1 (day 8 + 1, top) or 5 (day 8 + 5, bottom) days of HEC culture. H9 hESCs, *n* = 3, independent. Scale bars, 200 μm (top) and 400 μm (bottom). **f**, A flow cytometric analysis of endothelial (CD34⁺CD45ⁿᵉᵍ) and haematopoietic cells (CD45⁺, *RUNX1C*–EGFP⁺) derived from CD32⁺ cells isolated at day 8 of WNTd hPSC-derived haematopoietic culture and analysed after 5 days of HEC culture. Gated on SSC/FSC/Live. H9 hESCs, *n* = 3, independent. **g**, The quantification of erythro-myeloid CFC potential of CD32⁺ and CD32ⁿᵉᵍ populations isolated from day 8 WNTd hPSC-

derived haematopoietic cultures and cultured on OP9DLL1, H1 hESCs (*n* = 4), on the left, and H9 hESCs (*n* = 3), on the right. One-tail paired Student's *t*-test, non-parametric, for all biological replicates, statistics performed considering the total number of colonies, mean ± s.e.m., (*P* = 0.0155; **P* = 0.0092). BFU-E, burst-forming unit erythroid; CFU-GM, colony-forming unit granulocyte macrophage; CFU-M, colony-forming unit macrophage; CFU-GEMM, colony-forming unit granulocytes, erythrocytes, macrophages, megakaryocytes. **h,i**, Flow cytometric analysis (**h**) and barplot (**i**) of CD4⁺CD8⁺ T cell potential of CD32⁺ (orange) and CD32ⁿᵉᵍ (blue) cells isolated at day 8 of WNTd hPSC-derived haematopoietic cultures. Gated on SSC/FSC/Live/CD45⁺CD56ⁿᵉᵍCD7⁺CD5⁺ H1 hESCs, *n* = 3, independent; One-tail paired Student's *t*-test, non-parametric, for all biological replicates, mean ± s.e.m., **P* = 0.0073. **j**, The quantification of the erythro-myeloid CFC potential of CD44⁺ (green) and CD44ⁿᵉᵍ (purple) populations isolated at day 8 of WNTd hPSC-derived haematopoietic cultures and cultured on OP9DLL1. One-tail paired Student's *t*-test, non-parametric, for all biological replicates (H1 hESCs, *n* = 3, independent) considering the total number of colonies, mean ± s.e.m., **P* = 0.0098. **k,l**, A barplot showing the frequency (**k**) and a table showing the clonal analysis (**l**) of CD45⁺ and CD45ⁿᵉᵍ clones derived from CD32⁺ or CD44⁺ cells isolated at day 8 of WNTd hPSC-derived haematopoietic cultures. One-way ANOVA for all biological replicates (H1 hESCs, *n* = 4), mean ± s.e.m. (NS, not significant *P* = 0.9855; ****P* < 0.0001; **P* = 0.0028).

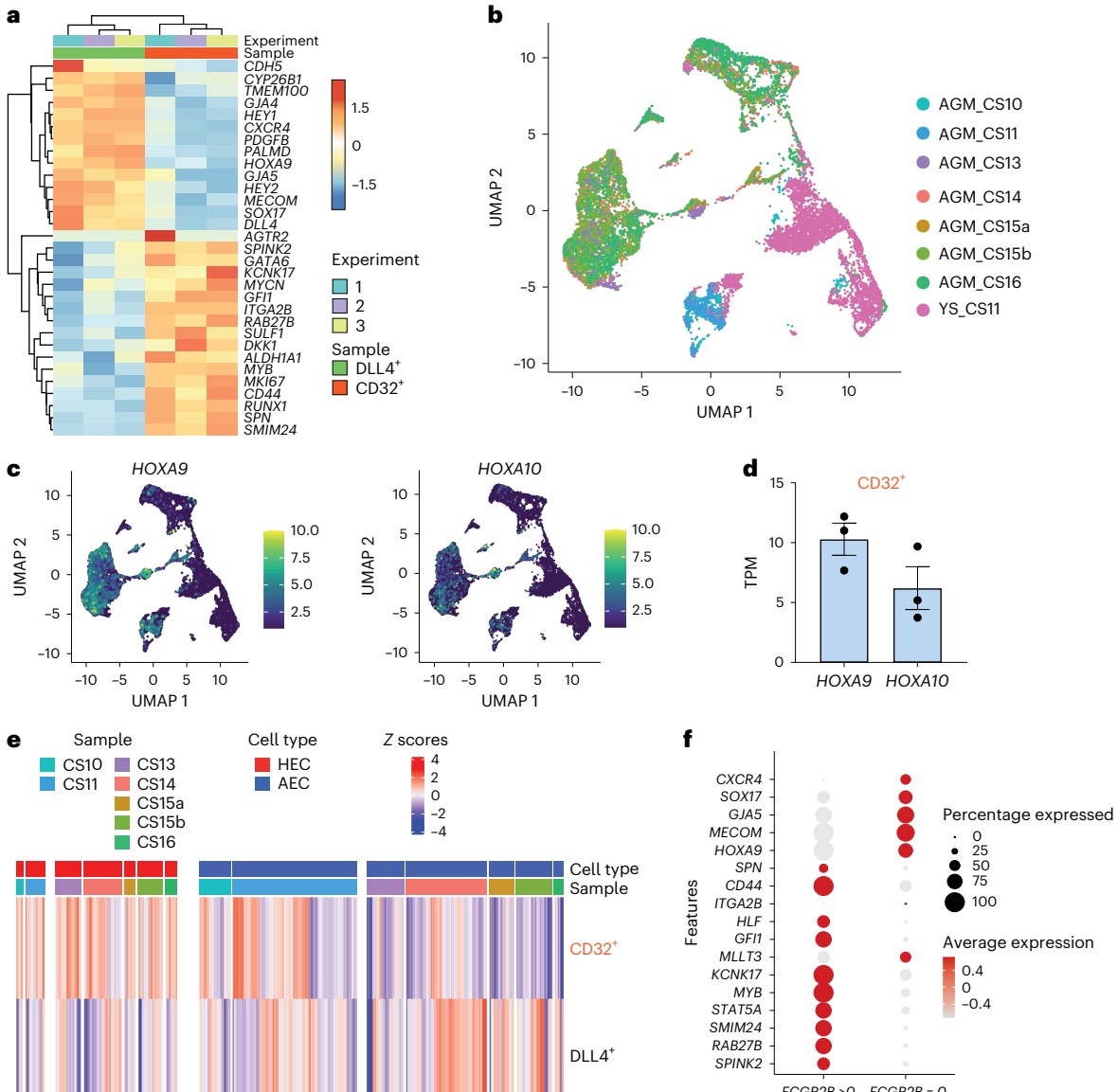

**Fig. 4 | Transcriptomic similarity between CD32⁺ cells and AGM-derived HECs.**
**a**, A heatmap showing a selection of DEGs in CD34⁺CD184⁺CD73⁺DLL4⁺CD43ⁿᵉᵍ (DLL4⁺, in green) or CD34⁺CD43ⁿᵉᵍCD184ⁿᵉᵍCD73ⁿᵉᵍCD32⁺ (CD32⁺, in orange) samples isolated at day 8 of WNTd hPSC-derived haematopoietic culture. H1 hESCs, *n* = 3 independent (#1 in light blue, #2 in purple and #3 in yellow). Scaled rlog gene expression values are shown in rows. The colouring indicates differential expression by upregulation (red) or downregulation (blue). **b**, UMAP of integrated publicly available single-cell datasets (accession codes GSE135202 and GSE162950) showing cell clusters of AGM (CS10–CS16) or YS (CS11). Cells clustered at resolution 1.2. **c**, Feature plots showing *HOXA9* and *HOXA10*

expression across the clusters shown in **b**. **d**, A barplot displaying the transcript per million (TPM) values for *HOXA9* and *HOXA10* genes analysed from the RNA-seq data of CD32⁺ cells as described in **a**, mean ± s.e.m. **e**, A heatmap visualizing the similarity scores between AGM samples from the single-cell data shown in **b** and annotated as HEC (red) or AEC (blue) in comparison with CD32⁺ and DLL4⁺ samples. Scale bar: *Z* scores of the relative Spearman coefficients. Each column is representative of a single embryonic cell scored across each population indicated by the row name. **f**, A scorecard dot plot showing landmark genes as reported in ref. 38. Differential expression was evaluated in *CDH5⁺RUNX1⁺PTPRCⁿᵉᵍ* cells that either express *FCGR2B* (*FCGR2B > 0*) or do not (*FCGR2B = 0*).

Pseudotime analysis by Monocle3 revealed that *FCGR2B⁺* HECs represent an intermediate state for the progression of *RUNX1⁺KCNK17⁺H19⁺FCGR2B*ⁿᵉᵍ HECs[39] to *RUNX1⁺* cells expressing haematopoietic genes such as *SPN* (despite being CD43ⁿᵉᵍ) and *MYB* (Fig. 5b and Extended Data Fig. 5e,f). Similarly to Monocle3, two additional trajectory inference methods[40,41] depicted a unified developmental sequence with minimal deviations characterized by a *FCGR2B⁺* intermediate HEC state (Extended Data Fig. 5g–i). Collectively these analyses suggested that clusters 0, 1, 2, 11, 16 and 17 identified by scRNA-seq represent progressive developmental states of HECs within day 8 WNTd CD34⁺CD43ⁿᵉᵍCD184ⁿᵉᵍCD73ⁿᵉᵍ cells. We next performed GSEA and ORA across this progression to dissect the unique features of *FCGR2B⁺*

HECs and identify state-specific gene expression profiles proper of HECs with distinct characteristics. In particular, we observed that the gradual loss of endothelial identity of *H19⁺* HECs begins with the downregulation of genes associated with extracellular organization, cell adhesion and cytoskeletal remodelling (Fig. 5c and Supplementary Tables 9–12). The progression to the *FCGR2B⁺* HEC cluster is associated with an enrichment of the expression of several ribosomal protein genes, consistent with the role of RUNX1 in regulating ribosome biogenesis[42], a key process for the emergence of blood cells (Fig. 5c and Supplementary Tables 9–12). Next, concomitant with the upregulation of haematopoietic genes, HECs also display increased expression of genes associated with cell motility as well as cell cycle progression

(Fig. 5c and Supplementary Tables 13–16). Indeed, using scRNA-seq data to infer the cell cycle state across HEC states, we observed that, during the progression from $H19^+$ to $FCGR2B^+$ cluster (clusters 0, 1 and 2 to cluster 11), cells are mostly in G1 phase, while cells in the $MYB^+$ HEC clusters (clusters 16 and 17) are mostly in S/G2/M phase (Fig. 5d).

Interestingly, the $MYB^+$ HEC clusters (that is, clusters 16 and 17) negatively correlated with the expression of genes associated with the activation of NOTCH signalling (Fig. 5c and Supplementary Tables 14–16). Since NOTCH signalling is an essential driver of stage-specific intra-embryonic emergence of haematopoietic cells[11], we further analysed the expression trend according to pseudotime of the well-characterized NOTCH target genes in the DA, that is, $HES1$, $HEY1$ and $HEY2$ (Fig. 5e). The analysis revealed that $HES1$ expression peaks in cells belonging to cluster 11, marked by $FCGR2B$ differential expression, while $HEY1$ and $HEY2$ expression peaks in cells that precede $FCGR2B^+$ cells in this pseudotime (Fig. 5e). This suggests that $FCGR2B$ expression might identify a NOTCH-independent state of HECs. To test this hypothesis, we added the chemical γ-secretase inhibitor L-685,458 (γSi) to the HEC culture of day 8 WNTd CD32$^+$ as well as CD32$^{neg}$ cells, as some of the latter will generate CD32$^+$ cells (Extended Data Fig. 3e). While CD32$^{neg}$ cells gave rise to haematopoietic progenitors in a NOTCH-dependent manner, the chemical inhibition of NOTCH signalling did not impair the generation of CD45$^+$ haematopoietic progenitors from CD32$^+$ cells (Fig. 5f,g). Altogether these results show that CD32 expression defines the temporal NOTCH requirement within the HEC continuum and that HECs activate a cell division programme to give rise to haematopoietic progeny.

To further elucidate the role of CD32 in the ontogeny of HECs, we performed an in silico perturbation analysis of CD32 across the six identified HEC clusters using CellOracle[41]. By simulating a knockout of $FCGR2B$ and observing the resultant effects on both direct and indirect gene targets, we identified a shift in the developmental trajectory of HEC differentiation, particularly noting a reversion in differentiation around cluster 11 (Extended Data Fig. 6a–c). This pattern suggests that a perturbation of CD32 signalling is likely to affect the differentiation process at these critical stages of HEC development. Therefore, we silenced FCGR2B expression in H1 hESCs (herein CD32 knock-down, KD) introducing a knockdown construct into the AAVS1 'safe harbour' locus to perform functional studies[43] (Extended Data Fig. 6d). This strategy effectively reduced the expression of CD32 without affecting the overall day 8 CD34$^+$ cell output (Extended Data Fig. 6e,f). Upon differentiation, CD32 KD cells displayed a significantly reduced haematopoietic output compared with H1 wild-type cells (Fig. 5h and Extended Data Fig. 6g). This suggests that, under these conditions, in the absence of FCGR2B the efficiency of haematopoietic differentiation is severely reduced.

**Stage-specific regulators of haematopoietic development**

We then leveraged the specificity of CD32 expression on HECs to identify pathways driving HEC specification that can be harnessed to increase the haematopoietic output from hPSC cultures. ORA analysis of the transcriptomic profiles of the CD32$^+$ and DLL4$^+$ hPSC-derived populations described above revealed that CD32$^+$ cells are positively associated with BMP signalling GO terms (Fig. 6a and Supplementary Table 17). This led us to hypothesize that the development of CD32$^+$ HECs may require active BMP signalling. We therefore tested the effects of either the activation or the inhibition of BMP4 signalling during the HEC specification stage (day 3 to day 8, Extended Data Fig. 3a) in WNTd hPSC differentiations. While the addition of BMP4 resulted in around a twofold increase of the proportion of CD32$^+$ cells at day 8, BMP signalling inhibition (BMPi) severely impaired their formation (Fig. 6b,d). We functionally validated the role of BMP signalling during HEC specification by testing the haematopoietic potential of day 8 CD144$^+$ cells isolated from BMP- or BMPi-treated WNTd haematopoietic cultures, as it should reflect the variation of the proportion of CD32$^+$ cells observed. As expected, CD144$^+$ cells isolated from BMP-treated cultures gave rise to twofold more CD45$^+$ cells and haematopoietic progenitors after HEC culture. On the other hand, BMP signalling inhibition almost abrogated the haematopoietic potential of day 8 CD144$^+$ cells (Fig. 6c,e).

We next leveraged the hPSC-derived scRNA-seq dataset described above to identify pathways regulating $RUNX1^+$ HECs progression to the blood fate. We focused on the progression to $FCGR2B^+$ state (that is, from clusters 0, 1 and 2 to cluster 11, Fig. 5b) given that CD32 marks a unique HEC stage. ORA revealed that this progression is associated with a downregulation of Rho/ROCK signalling (Fig. 6f and Supplementary Table 18). We then tested whether ROCK signalling inhibition during the HEC culture could increase the haematopoietic output from hPSCs. Indeed, ROCK signalling inhibition increased the proportion of CD45$^+$ and clonogenic progenitors generated during HEC culture (Fig. 6g,h), probably by synchronizing or facilitating the progression of HECs towards the blood fate. Collectively, these results indicate that HEC specification requires stage-specific BMP signalling activation and Rho/ROCK signalling inhibition, highlighting the value of using CD32 to study HEC biology.

**HEC CD32 expression is conserved across developmental programmes**

Since CD32 is also expressed in YS endothelial cells where it enriches for a population with haematopoietic potential (Fig. 2d,f), we investigated whether CD32 is a conserved marker of HECs with robust haematopoietic potential across different haematopoietic programmes specified from hPSCs. For this, we used the hPSC differentiation protocol that includes a stage-specific inhibition of WNT signalling by IWP2. This leads to the emergence by day 8 of WNT-independent (WNTi) $HOXA^{neg/low}$ extra-embryonic-like HECs, whose gene expression and potential are similar to the YS-derived EMP haematopoietic programme[9,35,36,44]. In accord with what we observed in WNTd cultures, CD32 identifies a subset of day 8 WNTi cells CD34$^+$CD43$^{neg}$CD73$^{neg}$CD184$^{neg}$DLL4$^{neg}$ (Extended Data Fig. 7a,b). Likewise, CD32$^+$ cells were enriched for HECs, since they generated significantly more erythro-myeloid clonogenic progenitors (Extended Data Fig. 7c). In addition, WNTi CD32$^+$ cells display robust capacity to generate NK-lymphoid cells (Extended Data Fig. 7d), the defining lymphoid lineage of YS-derived haematopoietic

**Fig. 5 | CD32 identifies a NOTCH-independent HEC state. a**, UMAP of single-cell data of day 8 WNTd CD34$^+$CD43$^{neg}$CD184$^{neg}$CD73$^{neg}$ cells. H1 hESCs, $n = 1$. Cells clustered at resolution 0.6. $RUNX1$-expressing clusters highlighted by a dashed black line. ECs, endothelial cells; EndoMT, endothelial-to-mesenchyme transition; M phase, mitotic phase. **b**, UMAP of the single-cell trajectory performed by Monocle3 on clusters 0, 1, 2, 11, 16 and 17 displaying $RUNX1$ as a differential marker. **c**, GSEA (significantly enriched GO terms, adjusted $P$ value <0.05) on FC pre-ranked genes from the comparison between cluster 11 and clusters 0, 1 and 2 (top) or cluster 11 and clusters 16 and 17 (bottom). Upregulated genes are shown in red, downregulated in blue. NES, normalized enrichment score; ECM, extracellular matrix. **d**, UMAP (top) and donut charts (bottom) showing the cells in clusters 0, 1 and 2, cluster 11, and clusters 16 and 17 coloured according to the cell cycle phase. G1 phase, yellow; S phase, purple; G2/M phase, green. **e**, Pseudotime kinetics of the expression variation of $FCGR2B$ and $HES1$, $HEY1$ and $HEY2$ along the clusters with differential $RUNX1$ expression. Cells coloured by cluster identity. Lines denote relative average expression of each gene in pseudotime. **f**, Flow cytometric analysis of CD45 and CD34 after HEC culture of CD32$^{+/neg}$ treated with DMSO, as control, or γ-secretase inhibitor L-685,458 (γSi) to block NOTCH signalling. Gated on SSC/FSC/Live. $n = 3$, independent. APC-Cy7, allophycocyanin-cyanine7. **g**, A barplot showing the frequency of CD45$^+$ cells derived from CD32$^+$ and CD32$^{neg}$ fraction treated with DMSO (in petroleum) or γ-secretase inhibitor L-685,458 (γSi, in white) to block NOTCH signalling. One-way ANOVA for all biological replicates ($n = 3$), mean ± s.e.m. (*$P = 0.0456$; NS, not significant, $P = 0.9623$). **h**, Flow cytometric analysis of CD45/CD34 after HEC cultures. of wild-type H1 hESCs (left) and H1 hESCs expressing 3× short hairpin RNAs (shRNAs) against $FCGR2B$ (referred to as CD32 knock-down, KD, right), gated on SSC/FSC/Live/GFP$^+$ cells.

progenitors[44]. Collectively, these results show that CD32 expression in hPSC cultures demarcates an endothelial subpopulation endowed with robust multilineage haematopoietic potential across different haematopoietic developmental programmes.

## Discussion

The precise identification of human HECs will enable the characterization of this transient population and unveil what regulates their transition to blood, thus allowing us to determine their identity, which is still subject to debate. In this study, using transcriptomic analysis of haemogenic populations sorted from human embryos, we identified *FCGR2B*, which encodes for a CD32 isoform, as a marker whose expression is upregulated in HECs. CD32 expression can be used in combination with other endothelial markers to precisely isolate HECs with robust haematopoietic potential from both the human embryo and hPSC differentiating cultures. In fact, CD32 expression is more specific for hPSC-derived HECs than CD44, another marker known to be expressed in HECs as well as in arterial cells[13,20,21].

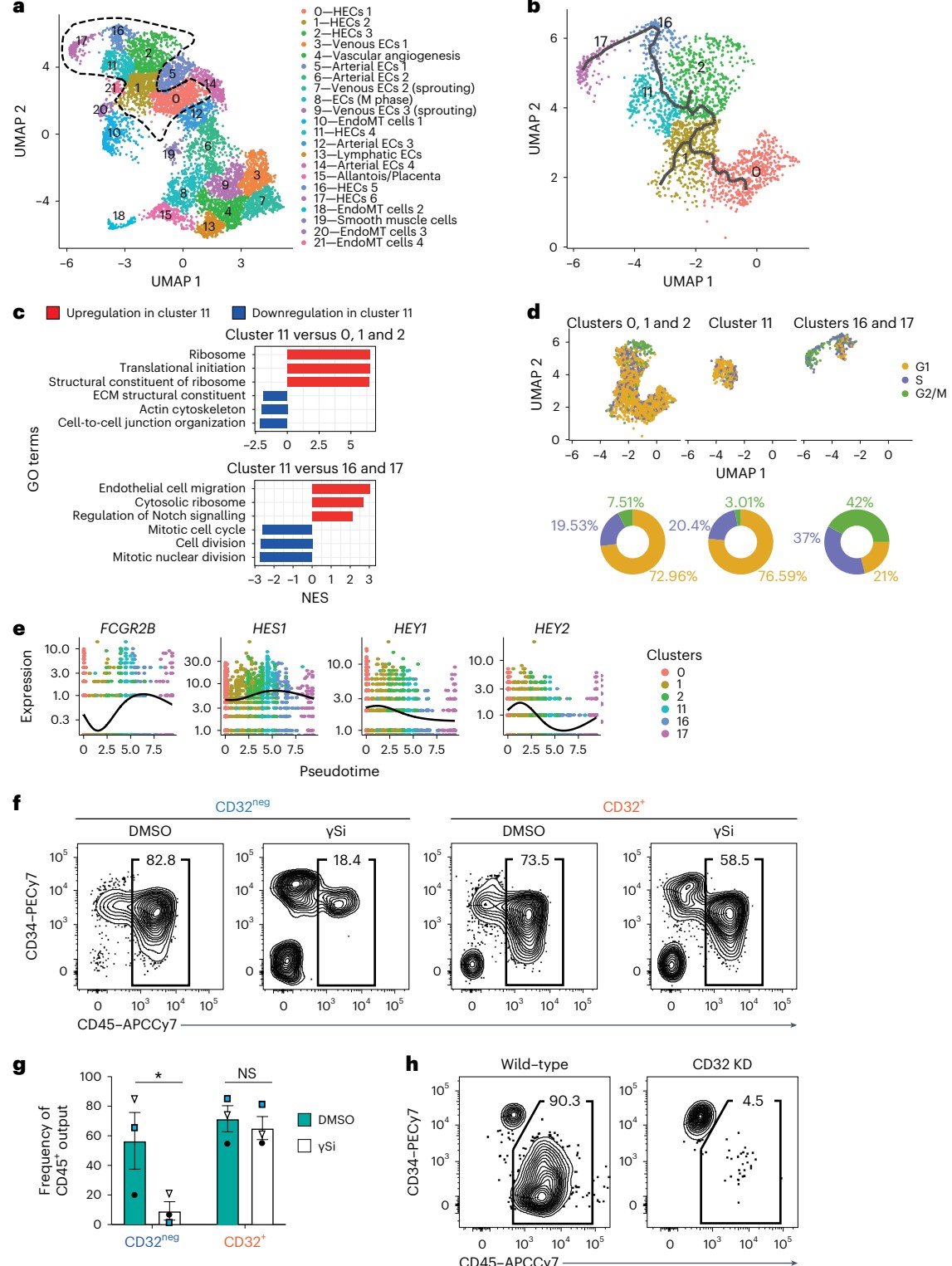

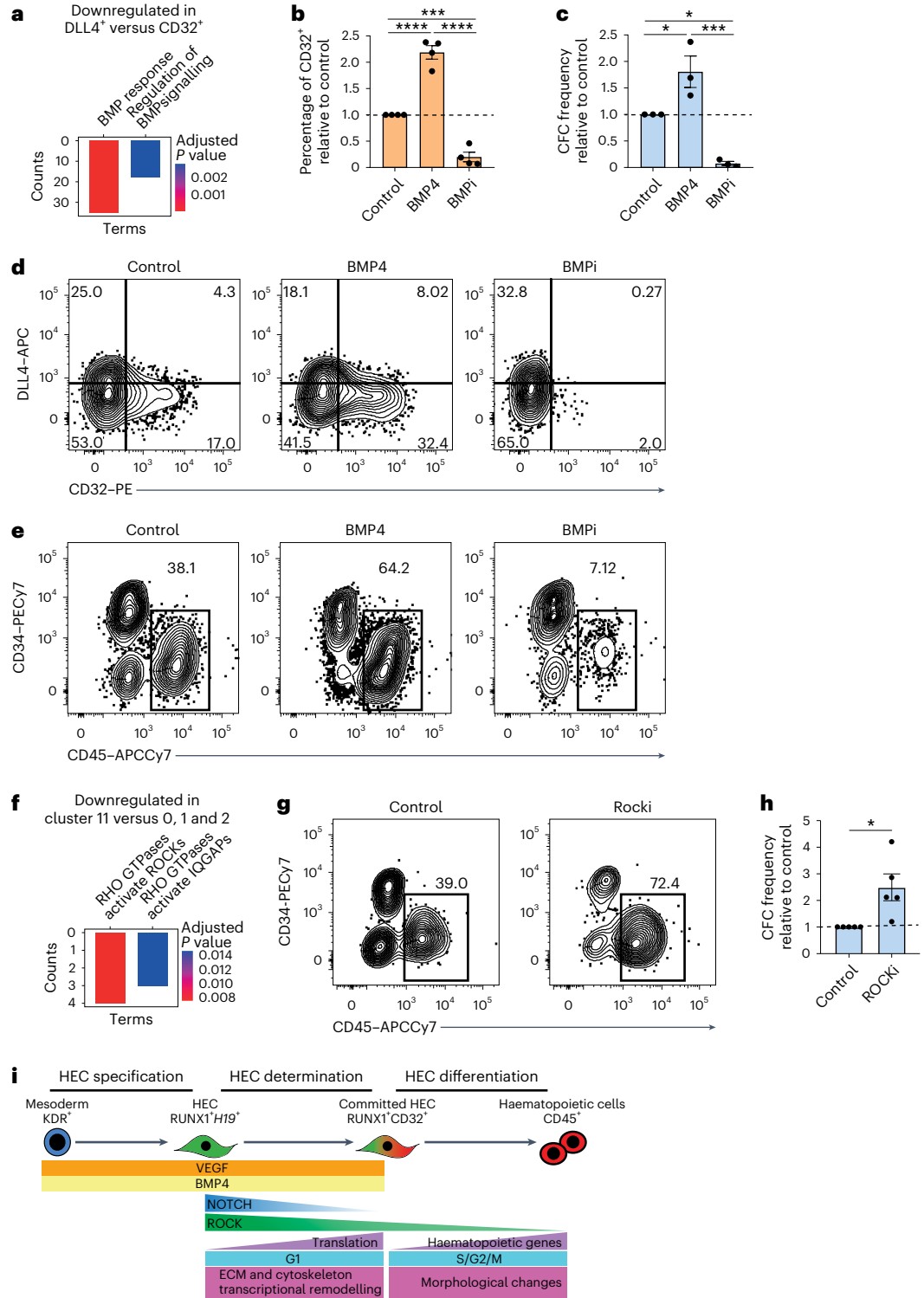

**Fig. 6 | HEC specification and haematopoietic cell emergence require stage-specific BMP and ROCK signalling modulation. a**, ORA (significant BMP-related BP terms, Fisher exact test; FDR <0.05) on downregulated DEGs between DLL4⁺ versus CD32⁺ samples. Bars coloured by adjusted *P* value. **b,c**, Fold difference of CD32⁺ cell frequency in day 8 WNTd cells (**b**) and of CFC frequency post HEC cultures of cells obtained with BMP signalling modulation (**c**). Frequencies are shown relative to control. One-way ANOVA for all biological replicates (*n* = 3, independent, H9 hESCs), mean ± s.e.m. (for **b**, ***P = 0.0004; ****P < 0.0001; for **c**, control versus BMP4: *P = 0.0379; control versus BMPi: *P = 0.02170; BMP4 versus BMPi: ***P = 0.001). **d,e**, Flow cytometry analysis of CD32/DLL4 expression in day 8 WNTd cultures (**d**) and CD45/CD34 expression after HEC cultures of cells obtained with BMP signalling modulation (**e**). For **d**, gated on SSC/FSC/Live/CD34⁺CD43ⁿᵉᵍCD184ⁿᵉᵍCD73ⁿᵉᵍ; for **e**, gated on SSC/FSC/Live. H9 hESCs, *n* = 3, independent. APC, allophycocyanin;

PE, phycoerythrin; PE-Cy7, phycoerythrin-cyanine7; APC-Cy7, allophycocyanin-cyanine7. **f**, ORA (significant RHO-related Reactome Pathways terms, Fisher exact test; FDR <0.05) on downregulated DEGs between cells of cluster 11 versus clusters 0, 1 and 2. Bars are coloured by adjusted *P* value. **g,h**, Flow cytometric analysis of CD45/CD34 expression (**g**) and fold difference of CFC frequency after ROCK inhibition (**h**). Gated on SSC/FSC/Live. Frequencies are shown relative to control. One-tail paired Student's *t*-test, non-parametric, for all biological replicates (H9 hESCs, *n* = 3, independent), mean ± s.e.m., *P = 0.0404. **i**, Model depicting the emergence of blood cells during human embryonic development. HECs are specified from mesodermal cells via VEGF and BMP signalling. HECs then undergo a NOTCH-dependent determination process that culminates with CD32 expression, which is followed by a cell cycle re-entry for the NOTCH-independent differentiation into blood cells. ECM, extracellular matrix.

The use of a hPSC-based model has allowed us to capture and dissect in fine detail the HEC progression to the generation of blood cells. By providing new granularity to a rare process that in vivo occurs very rapidly, our study demonstrates that HECs can be found in different intermediate states, which display transcriptional profiles indicative of distinct cellular characteristics. The progression of HECs towards the haematopoietic fate begins with a gradual change in the expression of genes associated with the remodelling of the extracellular matrix, the loss of adhesion molecules and the re-organization of the cytoskeleton (Fig. 5c). This suggests that the transcriptional changes associated with the release of haematopoietic cells in the bloodstream occur before the actual morphological remodelling can be observed. In addition, this HEC progression culminates with the enhancement of ribosome biogenesis and translation in CD32[+] committed HECs that appear to be irreversibly fated to haematopoiesis. Indeed, CD32[+] HECs can generate haematopoietic progeny independently of NOTCH signalling, the major driver of this process (Fig. 5f). As such, our study defines the temporal requirement of NOTCH signalling to initiate the haematopoietic programme in intra-embryonic HECs, with CD32 expression demarcating pre-NOTCH versus post-NOTCH states during HEC progression to the blood fate.

The initiation of the morphological remodelling and the concomitant expression of haematopoietic markers are often identified as the moment of the haematopoietic specification of HECs[45]. However, our data suggest that the commitment of HECs to the blood fate is temporally distinct from, and becomes NOTCH-independent before, the full execution of the haematopoietic programme, which occurs at a different cell cycle state. In fact, while the fate decision coincides with a timely suppression of the cell cycle, the emergence of blood cells is associated with cell cycle re-entry (Fig. 5d). Given that the acquisition of an active cell cycle to generate lineage output is a hallmark of cell differentiation[46], rather than developmental lineage transition[47,48], our findings support a model in which blood cells emerge via the differentiation of haematopoietic-restricted HECs rather than an endothelial-to-haematopoietic transition. In this model (Fig. 6i), we propose that HECs are specified from mesodermal cells in a vascular endothelial growth factor (VEGF)- and BMP-dependent manner. HECs expressing RUNX1 isoforms driven by the P2 proximal promoter then enter the determination stage, which is a process driven by BMP and NOTCH signalling. During this process, which occurs in G1 phase, RUNX1[+] HECs increase their translation and gradually downregulate Rho/ROCK signalling to become CD32[+]. This marks the irreversible commitment of HECs that progressively upregulate the expression of haematopoietic genes while re-entering the cell cycle to upregulate RUNX1C and differentiate into blood cells in a NOTCH-independent manner.

Our studies suggest that CD32 plays a functional role in the development of the human haematopoietic lineage. More refined studies are needed to determine the exact timing of its requirement and how mechanistically it exerts its role. In addition, the fact that in immuno-deficient humans HECs generate functional blood cells in the absence of circulating antibodies, the best-characterized ligands of CD32, suggests that CD32 could potentially functionally regulate blood development via the binding of alternative ligands.

Our findings indicate that CD32 is expressed in HECs harbouring multilineage haematopoietic potential isolated from different anatomic locations of the human embryo and in HECs derived from hPSC differentiations recapitulating distinct haematopoietic programmes (Figs. 2f and 3g, and Extended Data Fig. 7c). These data suggest that HEC progression to a CD32[+] stage before generating blood cells, is a conserved developmental process across ontogenies. However, we could not test whether CD32 is expressed on HSC-competent HECs, since culture conditions supporting human HSC specification from HECs, even from human embryonic explant cultures, have not been identified yet[49]. Indeed, additional challenging experiments are needed to formally prove a direct lineage relationship between CD32[+] HECs

and HSCs using stage-specific human embryos. As such, despite the functional results described, we cannot exclude that HECs can give rise to blood cells, potentially HSCs, independently of CD32 signalling.

In summary, this study demonstrates that expression of CD32 marks HECs fully committed to generate haematopoietic progeny and suggests that it could be used as a powerful tool to enrich haematopoietic precursors from a broad range of hPSC lines, including those for which current differentiation protocols into haematopoietic lineages are not optimal. Our findings will allow a deeper understanding of the specification of the HEC lineage, a central element of haematopoietic development, which will translate into optimized scaled-down, potentially more cost-effective, protocols to generate therapeutic blood products from hPSCs.

## Online content

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

## Methods

### Ethics declaration

The use of human embryonic tissues described in this study is compliant with the International Society for Stem Cell Research guidelines. All human embryonic tissue samples used in this study were discarded material from elective terminations that were obtained once informed written consent to the use of samples in research was obtained from patients. The donated human embryonic tissues were anonymized and did not carry any personal identifiers. In all cases, the decision to terminate the pregnancy occurred before the decision to donate tissue. No payments were made to donors, and the donors knowingly and willingly consented to provide research materials without restrictions for research and for use without identifiers. Human embryonic tissues employed for RNA-seq, immunohistochemistry and immunofluorescence were obtained from voluntary abortions performed according to the guidelines and with the approval of the French National Ethics Committee. The study was approved by Ospedale San Raffaele Ethical Committee (TIGET-HPCT protocol) and by the Institutional Review Board of the French Institute of Medical Research and Health (number 21-854). Human embryonic tissues employed for ex vivo haematopoietic cultures were collected by the Human Developmental Biology Resource (HDBR; HDBR project number 200430), Newcastle University, Newcastle, United Kingdom, with approval from the Newcastle and North Tyneside NHS Health Authority Joint Ethics Committee (08/H0906/21 + 5). The HDBR is regulated by the UK Human Tissue Authority (HTA; https://www.hta.gov.uk/) and operates in accordance with the relevant HTA Codes of Practice. This use was also approved by Ospedale San Raffaele Ethical Committee (TIGET-HPCT protocol). No embryos were created nor cultured for research purposes. The use of hESCs was approved by the Ospedale San Raffaele Ethical Committee, included in the TIGET-HPCT protocol.

### Human embryonic tissues

Human embryos were staged using anatomic criteria and the Carnegie classification. Samples employed for RNA-seq, immunohistochemistry and immunofluorescence were either used immediately as fresh tissues (ex vivo experiments and RNA-seq analysis) or fixed in phosphate-buffered saline (PBS) supplemented with 4% paraformaldehyde (Sigma-Aldrich), embedded in gelatin and stored at −80 °C (immunohistochemistry and immunofluorescence). Human embryonic tissues (CS12–CS13) analysed by RNA-seq were incubated in medium containing 0.23% w/v collagenase Type I (Worthington Biochemical Corporation, NC9482366) for 30 min at 37 °C, and the single-cell suspensions were filtered through a 70 μm cell strainer (BD Biosciences).

Human embryonic tissues (CS13) employed for ex vivo haematopoietic cultures were collected by HDBR (HDBR project number 200430), Newcastle University, Newcastle, United Kingdom, with written informed consent and approval from the Newcastle and North Tyneside NHS Health Authority Joint Ethics Committee (08/H0906/21 + 5). The human embryonic tissues were dissociated for 50 min at 37 °C with 10 mg ml$^{-1}$ collagenase/dispase (Sigma-Aldrich, 10269638001) in PBS with Ca$^{2+}$ and Mg$^{2+}$ (Sigma-Aldrich, D8662), supplemented with 7% heat-inactivated foetal bovine serum (FBS, Hyclone, 12389802), 1% penicillin–streptomycin (Lonza, DE17-603E) and 10 μg ml$^{-1}$ DNAse I (Calbiochem, 260913) and filtered through a 70 μm cell strainer (Falcon, 352235), similarly to what was previously described[50].

### Immunohistochemistry and immunofluorescence

The techniques employed have been previously described[19]. Briefly, 5-μm sections were incubated first with primary antibodies overnight at 4 °C, then for 1 h at room temperature (RT) with biotinylated secondary antibodies and finally with fluorochrome-labelled (BioLegend) or peroxidase-labelled streptavidin (Beckman Coulter).

Peroxidase activity was revealed with 0.025% 3,3-diaminobenzidine (Sigma-Aldrich) in PBS containing 0.03% hydrogen peroxide. Low amounts of antigens (CD32 and ACE) were revealed by Tyramide signal amplification biotin or fluorescence amplification systems (Akoya, Biosciences). An isotype-matched negative control was performed for each immunostaining. When 3,3-diaminobenzidine was used on slides, they were counterstained with Gill's haematoxylin (Sigma-Aldrich), mounted in XAM neutral medium (BDH Laboratory Supplies), analysed and imaged using an Optiphot 2 microscope (Nikon). Immunofluorescence-stained sections were cover-slipped in Prolong Gold Antifade Mountant with DAPI (Thermo Fisher Scientific) and analysed with an Axio Imager M2 microscope coupled to a Hamamatsu's camera Orca Flash 4v3 using the ApoTome.2 function (Zeiss) for optical sectioning. The antibodies employed are listed in Supplementary Table 19.

### RNA-seq

Human embryo sorted cells were collected in 6 μl of PBS supplemented with 0.5 μl of Protector RNAse inhibitor (Roche, 3335399001) and conserved in −80 °C. Full-length coding DNA (cDNA) was generated using Clontech SMART-Seq v4 Ultra Low Input RNA kit for Sequencing (Takara Bio Europe, 634891) according to the manufacturer's instructions with 15 cycles of PCR for cDNA amplification by Seq-Amp polymerase. A total of 600 pg of pre-amplified cDNA was then used as input for Tn5 transposon tagmentation by the Nextera XT DNA Library Preparation Kit (Illumina, FC-131-1096) followed by 12 cycles of library amplification. After purification with Agencourt AMPure XP beads (Beckman Coulter, A63882), the size and concentration of libraries were assessed by capillary electrophoresis. Libraries were then sequenced with the Illumina HiSeq 4000 sequencing platform in the single-end mode and with a read length of 50 bp. For day 8 WNTd hPSC-derived haematopoietic cultures, total RNA from sorted CD34$^+$CD32$^+$CD43$^{neg}$CD184$^{neg}$CD73$^{neg}$DLL4$^{neg}$ and CD34$^+$CD184$^+$CD73$^+$DLL4$^+$CD43$^{neg}$ was purified using the ReliaPrep RNA Cell Miniprep System and RNA-seq libraries were generated using the Smart-seq2 method. One nanogram of RNA was retrotranscribed, and cDNA was PCR amplified (15 cycles) and purified with AMPure XP beads. Sequencing was performed on an Illumina NovaSeq6000 (single-end, 100 bp read length) following the manufacturer's instruction.

For both RNA-seq datasets, raw reads quality control was accomplished using the FastQC tool (http://www.bioinformatics.babraham.ac.uk/projects/fastqc) and read trimming was performed using the Trim Galore software (https://doi.org/10.5281/zenodo.5127899) to remove residual adapters and low-quality sequences. Trimmed reads were aligned against the human reference genome (GRCh38) using STAR[51] with standard parameters. Uniquely mapped reads were then assigned to genes using the featureCounts tool from the Subread package[52], considering the GENCODE primary assembly v.34 gene transfer file as reference annotation for the genomic features. Gene count matrices were then processed by using the R/Bioconductor differential gene expression analysis packages DESeq2 (ref. 53) applying the standard workflow.

For the human embryos' dataset, a paired analysis was set up modelling gene counts using the following design formula: ~donor + condition. Gene $P$ values were corrected for multiple testing using false discovery rate (FDR). Genes with adjusted $P$ values <0.05 were considered differentially expressed.

ORA and a GSEA were then computed considering the GO Biological Process (BP) terms from the C5 collection of the Molecular Signatures Database (MSigDB version 7.2) using the R/Bioconductor package clusterProfiler[54] (http://bioconductor.org/packages/release/bioc/html/clusterProfiler.html, v 3.8.1). ORA was applied to the significantly DEGs, while GSEA was performed by pre-ranking genes according to fold change (FC) values. $P$ values were corrected for multiple testing using FDR and enriched terms with an adjusted

*P* value less than 0.05 were considered statistically significant. Volcano plots were generated using the R package ggplot2 (https://ggplot2.tidyverse.org) and have been used to display RNA-seq results plotting the statistical significance (adjusted *P* value) versus the magnitude of change (FC). Heatmaps were generated using the R package pheatmap (https://CRAN.R-project.org/package=pheatmap). Surface genes were extracted using the surfaceome database (http://wlab.ethz.ch/surfaceome/)[55].

## scRNA-seq

CD34[+]CD43[neg]CD73[neg]CD184[neg] cells were sorted at day 8 from WNTd hPSC-derived haematopoietic cultures that were treated at day 2 with 6 μM SB-431542 (Tocris, 1614)[56]. Libraries were prepared following the manufacturer's instructions using the Chromium platform (10x Genomics) with the 3′ gene expression (3′ GEX) V3 kit, using an input of ~10,000 cells. Briefly, Gel-Bead in Emulsions (GEMs) were generated on the sample chip in the Chromium controller. Barcoded cDNA was extracted from the GEMs by post GEM reverse transcription cleanup and amplified for 12 cycles. Amplified cDNA was fragmented and subjected to end-repair, poly A-tailing, adapter ligation and 10x-specific sample indexing following the manufacturer's protocol. cDNA libraries were sequenced in paired-end mode on a NovaSeq instrument (Illumina) targeting a depth of 50,000–100,000 reads per cell.

Sequencing reads were processed into gene count matrix by Cell Ranger (https://support.10xgenomics.com/single-cell-gene-expression/software/pipelines/latest/what-is-cell-ranger, v 4.0.0) from the Chromium Single Cell Software Suite by 10x Genomics. In detail, fastq files were generated using the Cell Ranger 'mkfastq' command with default parameters. Gene counts for each cell were quantified with the Cell Ranger 'count' command with default parameters. The human genome (GRCh38.p13) was used as the reference. The resultant gene expression matrix was imported into the R statistical environment (v 4.0.3) for further analyses. Cell filtering, data normalization and clustering were carried out using the R package Seurat[57] v 3.2.2. For each cell, the percentage of mitochondrial genes, number of total genes expressed and cell cycle scores (S and G1 phases) were calculated. Cells with a ratio of mitochondrial versus endogenous gene expression >0.2 were excluded as putative dying cells. Cells expressing <200 or >6,000 total genes were also discarded as putative poorly informative cells and multiplets, respectively. Cell cycle scores were calculated using the 'CellCycleScoring' function that assigns to each cell a score based on the expression of the S and G2/M phase markers and stores the S and G2/M scores in the metadata along with the predicted classification of the cell cycle state of each cell. Counts were normalized using Seurat function 'NormalizeData' with default parameters. Expression data were than scaled using the 'ScaleData' function, regressing on the number of unique molecular identifier, the percentage of mitochondrial gene expression and the difference between S and G2M scores. By using the most variable genes, dimensionality reduction was then performed with PCA by calculating 100 principal components (PCs) and selecting the top 55 PCs. Uniform manifold approximation and projection (UMAP) dimensionality reduction[58] was performed on the calculated PCs to obtain a two-dimensional representation for data visualization. Cell clusters were identified using the Louvain algorithm at resolution *r* = 0.6, implemented by the 'FindCluster' function of Seurat. To find the differentially expressed (marker) genes from each cluster, the 'FindAllMarkers' function (iteratively comparing one cluster against all the others) from the Seurat package was used with the following parameters: adjusted *P* values <0.05, average log FC >0.25, and percentage of cells with expression >0.1. A comprehensive manual annotation of the cell types was performed using the previously obtained markers list. DEGs between cells of cluster 11 against cells of clusters 0, 1 and 2 and clusters 16 and 17 were determined by the 'FindMarkers' function using the following parameters adjusted *P* values < 0.05, |average log FC| >0 and percentage of cells

with expression >0. GSEA was then performed considering GO BP terms from the C5 collection of the Molecular Signatures Database (MsigDB v 7.2) using the R/Bioconductor package clusterProfiler[54] (http://bioconductor.org/packages/release/bioc/html/clusterProfiler.html, v 3.8.1). ORA was computed on the significantly DEGs considering GO BP terms from the C5 collection of the Molecular Signatures Database (MsigDB v 7.2) and the Reactome Pathways Database using the R/Bioconductor package[54] (v 3.8.1). *P* values were corrected for multiple testing using FDR and enriched terms with an adjusted *P* value less than 0.05 were considered statistically significant. A barplot was constructed using the R package ggplot2 (https://ggplot2.tidyverse.org).

scRNA-seq samples from the public dataset GSE162950 were retrieved and processed as described in ref. 38. The 'DotPlot' and the 'VlnPlot' functions from the Seurat R package were used to construct a scorecard highlighting the expression pattern of selected cells *having RUNX1 + CDH5 + FCGR2B+* or *RUNX1 CDH5 + FCGR2B−* expression patterns.

Pseudotime trajectory was constructed using Monocle3 (https://cole-trapnell-lab.github.io/monocle3/, v 0.2.3)[59,60]. Expression and feature data were extracted from the Seurat object, and a Monocle3 'cell_data_set' object was constructed. The processed data were normalized followed by PCA analysis using the Monocle3 function 'preprocess_cds'. Dimensionality reduction was performed using the 'reduceDimension' function. Trajectory graph learning and pseudo-time measurement through reversed graph embedding were performed with 'learn_graph' function. Cells were ordered along the trajectory using the 'orderCells' method with default parameters. The 'plot_cells' function was used to generate the trajectory plots.

To corroborate the findings from Monocle3, a trajectory inference analysis was conducted using the dynverse workflow, a component of the R package dyno (v 0.1.2)[40]. The Dynbenchmark utility, which offers a comprehensive framework for selecting the most suitable trajectory inference method according to the available experimental data, was utilized via the 'guidelines_shiny()' function. Following these guidelines, the trajectory inference analysis was performed using the partition-based graph abstraction (PAGA)-tree algorithm[61]. Input data for dyno, including gene expression matrices, dimensionality reduction coordinates, clustering information and cell metadata, were derived from Seurat output and processed using the 'wrap_expression()' function. The cell trajectory was subsequently calculated by dyno using the 'infer_trajectory()' function, employing the 'ti_paga_tree()' method. Trajectory paths and pseudotime values were visualized on UMAP coordinates through the 'plot_dimred()' function provided by dyno.

To investigate the role of the CD32 gene in HEC ontogeny, genetic knockouts were simulated using the CellOracle tool (v 0.12.0)[41]. CellOracle integrates a gene regulatory network (GRN) with pseudotime analysis to predict shifts in cellular identities resulting from gene perturbations. This tool simulates alterations in gene expression due to perturbations and compares these changes with the cell's developmental trajectory within the GRN. This comparison allows for the estimation of transition probabilities between different cell states along the pseudotime axis. Following this, CellOracle generates a transition trajectory graph, illustrating the potential shifts in cellular identities after perturbation. This analysis was performed in a Python (version 3.8) environment, using Jupyter notebooks. scRNA-seq data, initially processed with Seurat, were converted to AnnData format using the anndata2ri tool (https://github.com/theislab/anndata2ri), ensuring content preservation for subsequent analysis. The CellOracle object construction utilized this data. Highly variable genes, critical for downstream analysis, were identified using the scanpy.pp.filter_genes_dispersion() function from scanpy[62], specifying n_top_genes = 3,000. A preliminary GRN was constructed using the oracle.get_links() function within the Oracle() class, based on ligand–receptor interactions from the CellTalkDB database[63]. This base GRN was further refined by incorporating the *CD32* gene and its interactors, as identified in

the STRING database (https://string-db.org/). Pseudotime analysis was conducted using the Pseudotime_calculator() class, employing the PAGA method from scanpy and integrating it into the CellOracle framework. This analysis culminated in the creation of a pseudotime gradient vector field with the Gradient_calculator() class from CellOracle, depicting the normal developmental trajectory. Subsequently, in silico perturbation of CD32 expression and simulation of resultant cell identity shifts were performed using the simulate_shift() and estimate_transition_prob() functions from the Oracle class. To compare the effects of CD32 perturbation with normal development, the Oracle_development_module class was used to calculate perturbation scores by computing the inner product of the respective vector fields with the calculate_inner_product() function.

## Human pluripotent stem cell maintenance and differentiation

The already-established H1 (WiCell Research Institute, WA01)[64] and *RUNX1C*–EGFP H9 (ref. 27) hESC lines were grown on irradiated mouse embryonic fibroblast feeders in hES medium defined as Dulbecco's modified Eagle medium/F12 medium (Corning, L022046-10092CVR) supplemented with 25% of KnockOut Serum Replacement (Thermo Fisher Scientific, 10828028), 1% penicillin–streptomycin (Lonza, DE17-603E), 2 mM L-glutamine (Lonza, BE17-605E), 0.1% β-mercaptoethanol (Sigma-Aldrich, M3148) and 0.7% of MEM non-essential amino acids solution (Thermo Fisher Scientific, 11140035). Right before usage, 1 µg ml⁻¹ ciprofloxacin HCl (Sigma-Aldrich, PHR1044-1G) and 20 ng ml⁻¹ human recombinant basic fibroblast growth factor (bFGF, R&D, 233-FB-500/CF) were added to hES medium. Alternatively, cells were cultured using Essential 8 medium (Thermo Fisher Scientific, A1517001) on Matrigel-coated plasticware (Corning Life Sciences, 356230). Cells were maintained and expanded at 37 °C, 21% $O_2$, 5% $CO_2$.

For differentiation, hPSCs were processed for embryoid body (EB) generation. EB aggregates were resuspended in SFD medium defined as 75% (Iscove's modified Dulbecco's medium (Corning, 15343531), 25% Ham's F12 (Corning, 10-080-CVR), 0.005% bovine serum albumin–fraction V, B27 supplement (Thermo Fisher Scientific, cat. no. 12587010), N2 supplement (Thermo Fisher Scientific, cat. no. 17502048), 1% penicillin–streptomycin and 1 µg ml⁻¹ ciprofloxacin HCl. The differentiation medium was supplemented as previously described[14,56]. Briefly, the first day of differentiation, SFD medium was supplemented with 2 mM L-glutamine, 1 mM ascorbic acid (Sigma-Aldrich, A4544), 400 µM 1-thioglycerol solution (Sigma-Aldrich, M6145), 150 µg ml⁻¹ transferrin (R&D, 2914-HT) and 10 ng ml⁻¹ BMP4 (R&D, 314-BP-MTO). Twenty-four hours later, 5 ng ml⁻¹ bFGF (R&D, 233-FB-500/CF) was added. At the second day of differentiation, 3 µM (or 5 µM for feeder-free cultures) CHIR99021 (Cayman Chemical Company, CT99201) was added, as indicated. On the third day, EBs were changed to StemPro-34 medium (Thermo Fisher Scientific, 10639011) supplemented with penicillin–streptomycin, L-glutamine, ascorbic acid, 1-thioglycerol and transferrin, as above, with additional 5 ng ml⁻¹ bFGF and 15 ng ml⁻¹ VEGF (R&D, MAB3572). On day 6, 10 ng ml⁻¹ interleukin (IL)6 (130-093-934), 25 ng ml⁻¹ insulin-like growth factor 1 (IGF1, 130-093-887), 5 ng ml⁻¹ IL11 (130-103-439), 50 ng ml⁻¹ stem cell factor (SCF) (130-096-696) and 2 U ml⁻¹ erythropoietin (EPO) (Peprotech, 100-64) were added. Where indicated, 10 ng ml⁻¹ of BMP4 or 250 nM of LDN193189 dihydrochloride (Tocris, 6053) were added from day 3 to day 8. To assess the emergence of CD32⁺ cells from CD32ⁿᵉᵍ, CD32ⁿᵉᵍ cells were isolated at day 8 of WNTd haematopoietic cultures and cultured in day 6 medium for 48 h. All cytokines were purchased from Miltenyi Biotec, unless indicated differently. All differentiation cultures were maintained at 37 °C. All EBs and mesodermal aggregates were cultured in 5% $CO_2$, 5% $O_2$, 90% $N_2$.

## Generation of CD32 KD hESC line

An AAVS1-CA-GFP-Puro donor plasmid comprising homology arms for the integration into the *AAVS1* locus, puromycin resistance gene (Puro) and GFP sequence fused to a polylinker site was used[43]. In this

donor plasmid, we synthetized three different short hairpin RNAs (shRNAs) against the 3′ untranslated region of *FCGR2B* (shRNA1: 3′-GGTTGGAGTGTAGACTGAACTGCCT-5′, shRNA2: 3′-TCAAG GCTGTATTGGTTGGAGTGTA-5′, shRNA3: 3′-CAAGGCTGTATTGGTTG GAGTGTAG-5′). H1 cells cultured under feeder-free conditions (Essential 8) were nucleofected using a Lonza 4D nucleofector and P3 Primary Cell reagents. Cells were nucleofected with 16 µg total of plasmid DNA including 7 µg of eCas9 + gRNA plasmid (Addgene #71814, modified to contain T2 gRNA: 5′-GGGGGCCACTAGGGACAGGA-3′), 7 µg of donor plasmid containing the 3×-CD32 miRNA insert, and 2 µg of p53DD (Addgene #4156). The modified cells were then enriched by puromycin selection and purified by FACS sorting based on GFP expression.

## T cell and NK cell differentiation

To test the T cell potential, candidate cells isolated by FAC sorting as indicated were seeded on OP9DLL4-coated 24-well plates. OP9DLL4 were a kind gift from Juan-Carlos Zúñiga-Pflücker and described previously[65]. The cells were cultured in Alpha MEM (Thermo Fisher, 12000063) supplemented with 2.2 g l⁻¹ sodium bicarbonate (Corning, 61-065-RO), 20% FBS (HyClone), 1% penicillin–streptomycin, 2 mM glutamine (Thermo Fisher Scientific) and 400 µM 1-thioglycerol solution. Cells were supplemented with 5 ng ml⁻¹ IL7, 5 ng ml⁻¹ FLT3L and, for the first 5 days of differentiation, 50 ng ml⁻¹ SCF. The cells were split every 4–5 days by vigorous pipetting and passaging through a 40-µm cell strainer and plated on freshly seeded stromal cells. T lymphoid output was assayed by FACS analysis after 21–24 days of differentiation. For the analysis of the T cell maturation and activation markers, the three-dimensional artificial thymic organoid (ATO)[66] system was adapted to our protocol. CD32⁺ cells were mixed with MS5DLL4 cells (kind gift of Tom Taghon, University of Ghent)[67] in a 1:50 ratio. Haematopoietic progenitors and stromal cells were centrifuged together at 1,500 rpm for 5 min and resuspended in the T cell differentiation medium, consisting of RPMI 1640 (Corning 10-040-CV), 4% B27 (Thermo Fisher Scientific 17504044), 30 µM ascorbic acid and 1% penicillin/streptomycin and GlutaMAX (Thermo Fisher #35050061). Each single aggregate was seeded in a volume of 5 µl onto the cell insert membrane for the air–liquid interphase culture. Cells were analysed after 6 weeks of culture. For NK-specific differentiation, CD32⁺ cells were cultured in Alpha MEM (Thermo Fisher, 12000063) supplemented with 2.2 g l⁻¹ sodium bicarbonate (Corning, 61-065-RO), 20% FBS (HyClone), 1% penicillin–streptomycin, 2 mM glutamine (Thermo Fisher Scientific) and 400 µM 1-thioglycerol solution onto OP9DLL4 stromal cells. Cultures were supplemented with 5 ng ml⁻¹ IL7 (130-095-362), 5 ng ml⁻¹ FLT3L (130-096-479), 10 ng ml⁻¹ IL15 (130-95-765) and, for the first 7 days of differentiation, 30 ng ml⁻¹ IL3 (130-095-070). All cytokines were purchased from Miltenyi Biotec. The cells were maintained for 14 days on the same stromal cells before performing FACS analysis. Every 7 days, the culture was supplemented with fresh medium.

## CFC generation assay

The colony-forming cell (CFC) generation assay was performed as previously described[68]. Briefly, sorted cells were cultured on irradiated OP9DLL1 (a kind gift from Juan-Carlos Zúñiga-Pflücker) monolayers in Alpha MEM (ThermoFisher, 12000063) supplemented with 20% FBS (HyClone), 1% penicillin–streptomycin 2 mM L-glutamine, 30 ng ml⁻¹ thrombopoietin (TPO) (Miltenyi Biotec, 130-095-747), 10 ng ml⁻¹ BMP4, 50 ng ml⁻¹, 25 ng ml⁻¹ IGF1, 10 ng ml⁻¹ IL11, 10 ng ml⁻¹ FLT3L and 4 U ml⁻¹ EPO. After 5 days, cells were collected using 0.25% trypsin–EDTA (Thermo Fisher Scientific, 25-200-056) for 3 min at 37 °C. Cells were then filtered through a 40 µM filter and seeded on methylcellulose medium (STEMCELL Technologies, H4034). Cells were seeded on methylcellulose supplemented with 150 µg ml⁻¹ transferrin, 50 ng ml⁻¹ TPO, 10 ng ml⁻¹ VEGF, 10 ng ml⁻¹ IL6, 50 ng ml⁻¹ IGF1, 5 ng ml⁻¹ IL11 and 4 U ml⁻¹ EPO. Colonies' number and morphology were evaluated after 15 days by light microscopy.

## HEC culture

CD34$^+$CD43$^{neg}$CD184$^{neg}$CD73$^{neg}$DLL4$^{neg}$CD32$^{+/neg}$ or CD34$^+$CD43$^{neg}$CD184$^{neg}$CD73$^{neg}$DLL4$^{neg}$CD32$^{neg}$CD44$^+$ were isolated at day 8 of WNTd haematopoietic culture and re-aggregated overnight at $3 \times 10^5$ cells ml$^{-1}$ as previously described[14]. The cells were seeded in Stempro medium, supplemented with 1% glutamine, 50 µg ml$^{-1}$ ascorbic acid, 150 µg ml$^{-1}$ transferrin, 400 µM 1-thioglycerol solution, 30 ng ml$^{-1}$ TPO, 10 ng ml$^{-1}$ VEGF, 5 ng ml$^{-1}$ bFGF, 30 ng ml$^{-1}$ IL3, SCF, 50 ng ml$^{-1}$ IGF1, 10 ng ml$^{-1}$ IL6, 5 ng ml$^{-1}$ IL11 and 4 U ml$^{-1}$ EPO. Aggregates were then transferred onto thin-layer Matrigel-coated plasticware where they were cultured for an additional 1–7 days in the same media. Where indicated 10 µM γ-secretase inhibitor (γSi) L-685,458 (Tocris, 2627) or equal volume of dimethyl sulfoxide (DMSO, Sigma-Aldrich, D4540), as control, were added to the HEC culture test the effect of NOTCH signalling inhibition. Where indicated, 10 µM of the Rho kinase inhibitor (ROCKi) Y-27632 dihydrochloride (Cayman Chemical, TB1254-GMP) or equal volume of DMSO, as control, were added to the HEC culture to test the effect of ROCK signalling inhibition.

Single CD34$^+$CD43$^{neg}$CD184$^{neg}$CD73$^{neg}$DLL4$^{neg}$CD32$^+$/CD44$^+$ cells were FACS-sorted directly onto a Matrigel-coated well of 96-well plate at day 8 of WNTd haematopoietic cultures. Cells were cultured as above. Haematopoietic and non-haematopoietic clones were evaluated by light microscopy and FACS analysis after 10–14 days of culture.

## Cell staining, flow cytometry and cell sorting

Samples for FACS analysis or cell sorting were incubated with antibody mixes for 15–30 min at 4 °C. Dead cells were excluded using 7-aminoactinomycin D (7AAD) during staining. For the analysis of the T cell maturation, ATO aggregates were stained with Maleimide PromoFluor840 for dead cell exclusion. Cells were then incubated with antibody mixes diluted in Brilliant stain buffer (BD Biosciences, 563794) + PBS (Corning, 21-040-CM) + 2% FBS (HyClone, SH30066.03) + 4% FcR blocking for 15 min at RT. After washing, cells were fixed in 1% paraformaldehyde and analysed. The antibodies employed are listed in Supplementary Table 19. Cells were sorted with FACSAria II with the FACSDiva software (BD Biosciences). Sorting gates were set using appropriate fluorescence minus one and single staining controls. FACS analysis was performed using FACS Canto with the FACSDiva software (BD Biosciences) or Cytoflex S or Cytoflex LX with Coulter CytExpert software (Beckman Coulter) for the acquisition and the FlowJo software (BD Biosciences) for the analysis. Where indicated, WNTd day 8 CD144$^+$ cells were selected using magnetic bead-based separation with CD144 MicroBeads (Miltenyi Biotec, 130-097-857) following the manufacturer's instructions.

## Statistics and reproducibility

For all multivariate statistical analyses, analyses of variance (ANOVAs) were performed with the appropriate corrections for multiple comparisons. One-way ANOVA with Tukey's multiple comparison test was chosen for single metrics with more than two populations. The data distribution was not formally tested but was assumed to be normally distributed with equal variance. Sample size and replication were determined by historical controls[14]. For bivariate statistical analyses, Student's $t$-test was performed with the appropriate corrections (one tail, non-parametric). In general, biological replicates were excluded only if internal controls failed and technical replicates were not excluded. Experimental conditions were not randomized, but covariates were controlled by an equal distribution of sorted cells across controls and experimental conditions. Blinding of experimental conditions was not relevant as our studies do not require grading of the results.

## Reporting summary

Further information on research design is available in the Nature Portfolio Reporting Summary linked to this article.

## Data availability

Sequencing data that support the findings of this study have been deposited in the Gene Expression Omnibus (GEO) under accession code GSE223223. Previously published data that were re-analysed here are available under accession code GSE135202 and GSE162950. Genome alignments were performed with GENCODE GRCh38.p13 version 34. Source data are provided with this paper and have also been deposited within the San Raffaele Open Research Data Repository and are available at https://doi.org/10.17632/ds6rcgfp7y.1 (ref. 69). All other data supporting the findings of this study are available from the corresponding author on reasonable request.

## Code availability

Scripts used for data analysis and for the generation of all the figures in the paper are available at link http://www.bioinfotiget.it/gitlab/custom/scarfo_hec2023 (ref. 70).

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

## Acknowledgements

We thank G. Keller, R. Ostuni and L. Naldini as well as C.M.S., M.T. and A.D. lab members for inputs and critical reading of the manuscript; M. Genua and R. Ostuni for the help in library preparation, the Flow cytometry Resource, Advanced Cytometry Technical Applications Laboratory (FRACTAL), the Centro di Statistica per le Scienze Biomediche (CUSSB) at Ospedale San Raffaele and the Human Developmental Biology Resource. Sequencing was performed by the GenomEast platform (at the IGBMC Illkirch, France), a member of the 'France Génomique' consortium (ANR-10-INBS-0009). J.-N.F. and M.T. were supported by INSERM and by grants from ANR (ANR-14-CE11-0008) and the Ligue Contre le Cancer Région Grand Est Bourgogne Franche Comté – CCIR Est and the FHU ARRIMAGE. M.E.K. was awarded a fellowship from EURIdoc programme (H2020-MSCA-COFUND-2020). L.N.R. was awarded a Michele and Carlo Ardizzone fellowship from the Italian Cancer Research Association (AIRC Italy ID 26779-2021). S.A.L. is supported by an American Society of Hematology Scholar Award. C.M.S. is supported by the Bill & Melinda Gates Foundation INV-002414, and NIH R01HL145290 and R01HL151777. This study was supported by grants to A.D. from the Italian Telethon Foundation (SR-Tiget grant awards C4 and G3b) and San Raffaele Hospital (Seed Grant). Research in the A.D. laboratory is supported by the European Research Council (consolidator grant 101044032, HSC-reNEW). R.S. conducted this study as partial fulfilment of an international PhD in Molecular Medicine, Vita-Salute San Raffaele University.

## Author contributions

A.D. formulated the initial concept. R.S., L.N.R., M.E.K., J.-N.F., A.V., S.G., C.M.S., M.T. and A.D. designed the experiments and analysed the data. R.S., L.N.R., M.E.K., A.G., S.A.L, Z.-Y.L., S.C., E.D., S.A.L., A.T., G.F.R., L.P., G.A., C.B., M.T. and A.D. performed the experiments. M.A.A., S.V. and I.M. performed bioinformatics analyses. R.S., M.A.A., C.M.S., M.T. and A.D. wrote the manuscript.

## Competing interests

A.D. and R.S. have filed a patent application for the methodology described in this manuscript. C.M.S. is a scientific founder and Scientific Advisory Board member of Clade Therapeutics. The other authors declare no competing interests.

## Additional information

**Extended data** is available for this paper at https://doi.org/10.1038/s41556-024-01403-0.

**Correspondence and requests for materials** should be addressed to Manuela Tavian or Andrea Ditadi.

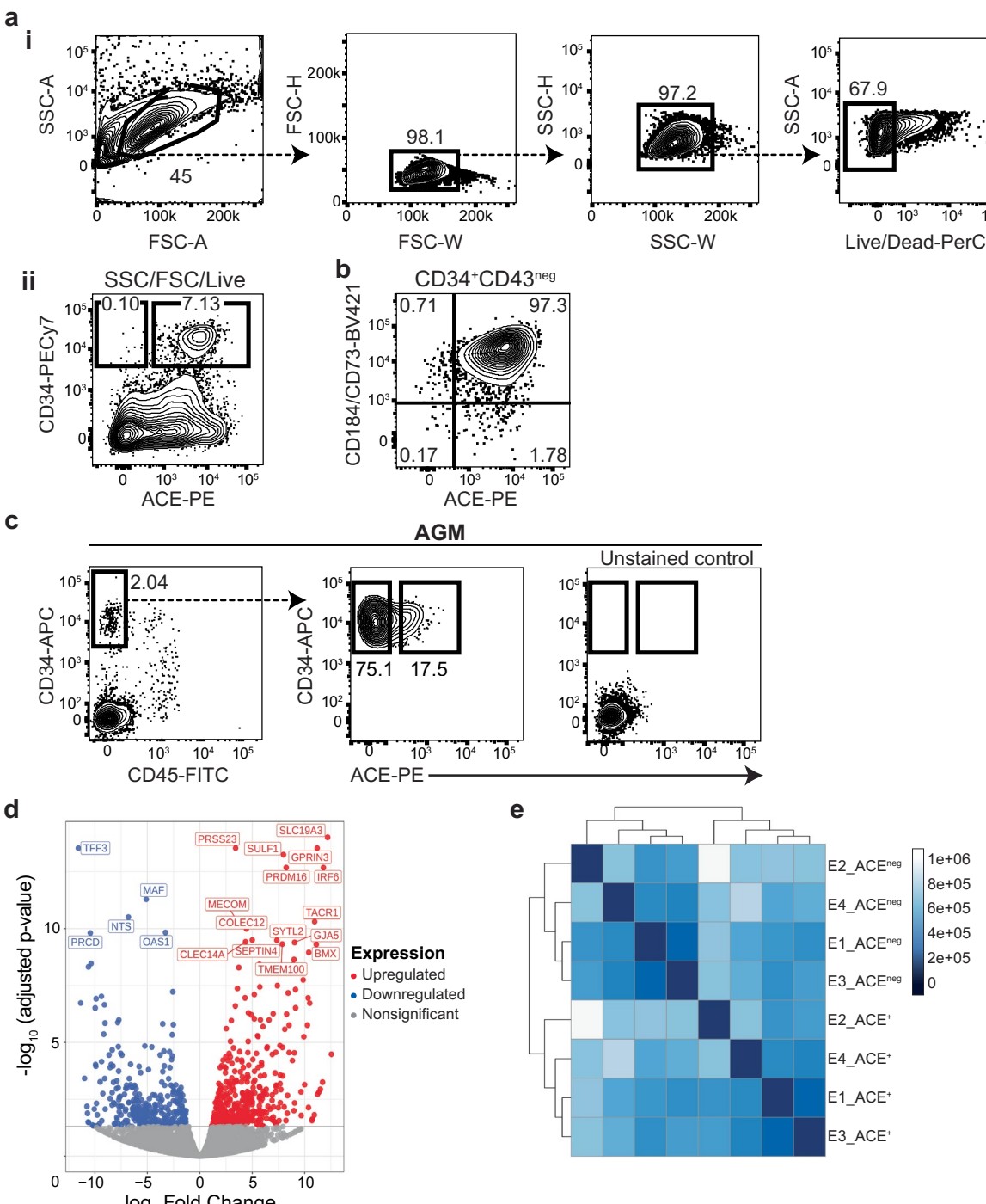

**Extended Data Fig. 1 | ACE is expressed within CD34⁺ cells during intra-embryonic hematopoiesis. a)** Representative flow cytometric analysis showing i) the gating strategy and ii) ACE and CD34 expression in day 8 WNTd hPS cell-derived hematopoietic cultures, gated on SSC/FSC/Live. H1 hESCs, *n* = 4, independent; **b)** Representative flow cytometric analysis showing CD184/CD73 and ACE expression in CD34⁺CD43ⁿᵉᵍ cells at day 8 of WNTd hPSC-derived hematopoietic cultures. Anti-CD184 and anti-CD73 antibodies are in the same color. Gated on SSC/FSC/Live/CD34⁺CD43ⁿᵉᵍ. H1 hESCs, *n* = 4, independent; **c)** Representative flow cytometric analysis showing the gating strategy to isolate ACE⁺ and ACEⁿᵉᵍ cells from the AGM of *n* = 4 CS12-CS13 human embryos. Left

panel: CD45 and CD34 expression, gated on SSC/FSC/Live. Middle panel: ACE and CD34 expression, gated on SSC/FSC/Live/CD34⁺CD45ⁿᵉᵍ. Right panel: unstained control, gated on SSC/FSC/Live; **d)** Volcano plot showing the differentially expressed genes in ACE⁺ and ACEⁿᵉᵍ cells isolated from the AGM of four CS12-CS13 human embryos (Wald test; FDR < 0.05). The top 20 differentially expressed genes are highlighted, upregulated in red, downregulated in blue, nonsignificant in grey; **e)** DESeq2 heatmap distance analysis of four CS12-CS13 human embryos used for RNA-seq. Samples were clustered using unsupervised hierarchical clustering by *k*-means.

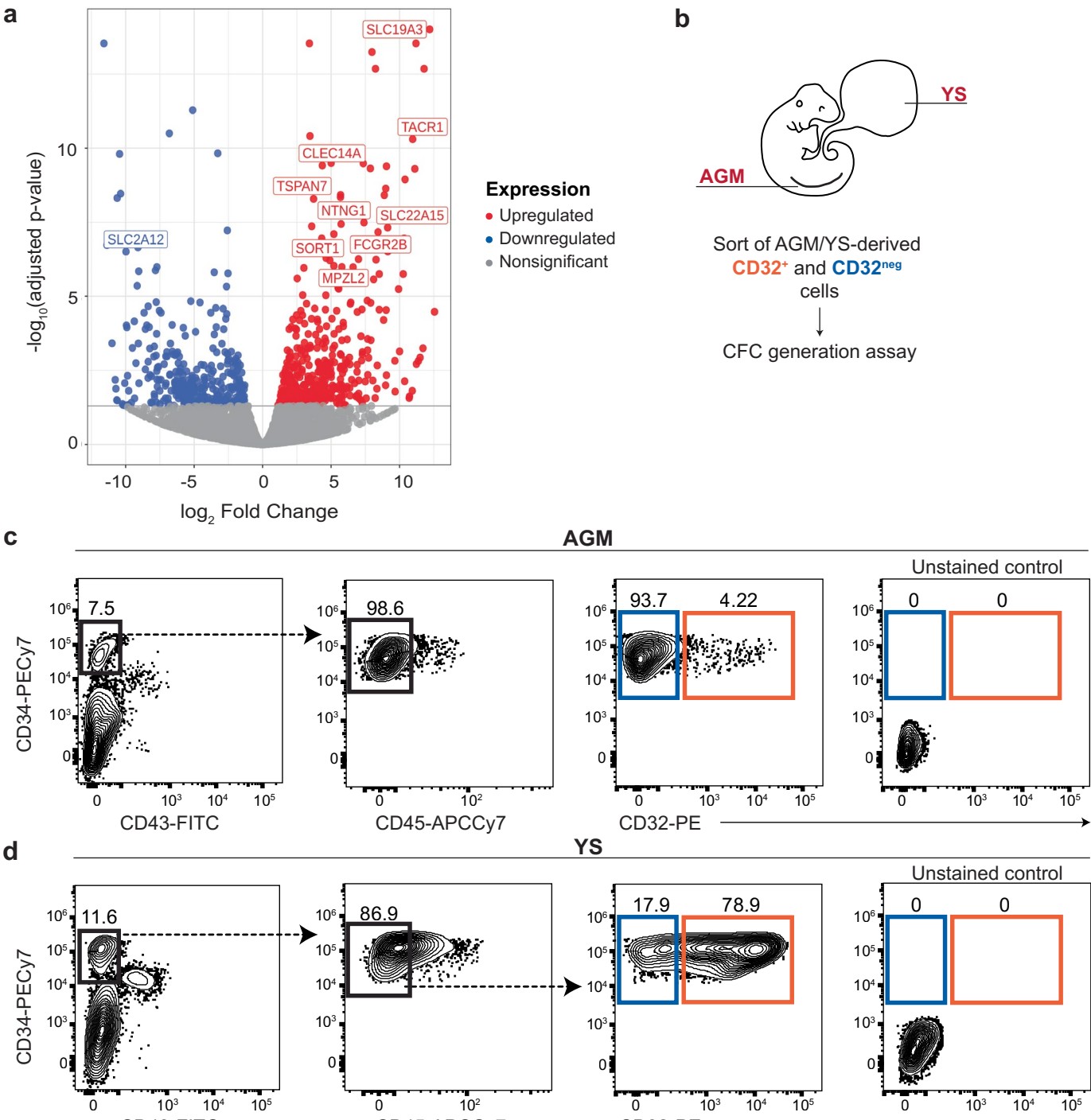

**Extended Data Fig. 2 | Identification of CD32 in the AGM and the YS of human embryos. a**) Volcano plot showing the differentially expressed cell surface genes in ACE[+] vs ACE[neg] cells isolated from the AGM of four CS12-CS13 human embryos (Wald test; FDR < 0.05). The top 10 differentially expressed cell surface genes are highlighted, upregulated in red, downregulated in blue, nonsignificant in grey; **b**) Experimental layout: CD34[+]CD43[neg]CD45[neg]CD32[+/neg] (referred to as CD32[+] and CD32[neg]) cells were FAC-sorted from the AGM and YS of two CS13 human embryos. Isolated cells were tested for their hematopoietic potential via CFC generation assay; **c**) Representative flow cytometric analysis showing the gating strategy to isolate CD32[+] (orange) and CD32[neg] (blue) cells from the AGM

of CS13 human embryo. From the left, first panel: gated on SSC/FSC/Live. Second panel: gated on SSC/FSC/Live/CD34[+]CD43[neg]. Third panel: gated on SSC/FSC/Live/CD34[+]CD43[neg]CD45[neg]. Fourth panel: unstained control, gated on SSC/FSC/Live. n = 2, independent; **d**) Representative flow cytometric analysis showing the gating strategy to isolate CD32[+] (orange) and CD32[neg] (blue) cells from the YS of CS13 human embryo. n = 2, independent. From the left, first panel: gated on SSC/FSC/Live. Second panel: gated on SSC/FSC/Live/CD34[+]CD43[neg]. Third panel: gated on SSC/FSC/Live/CD34[+]CD43[neg]CD45[neg]. Fourth panel: unstained control, gated on SSC/FSC/Live.

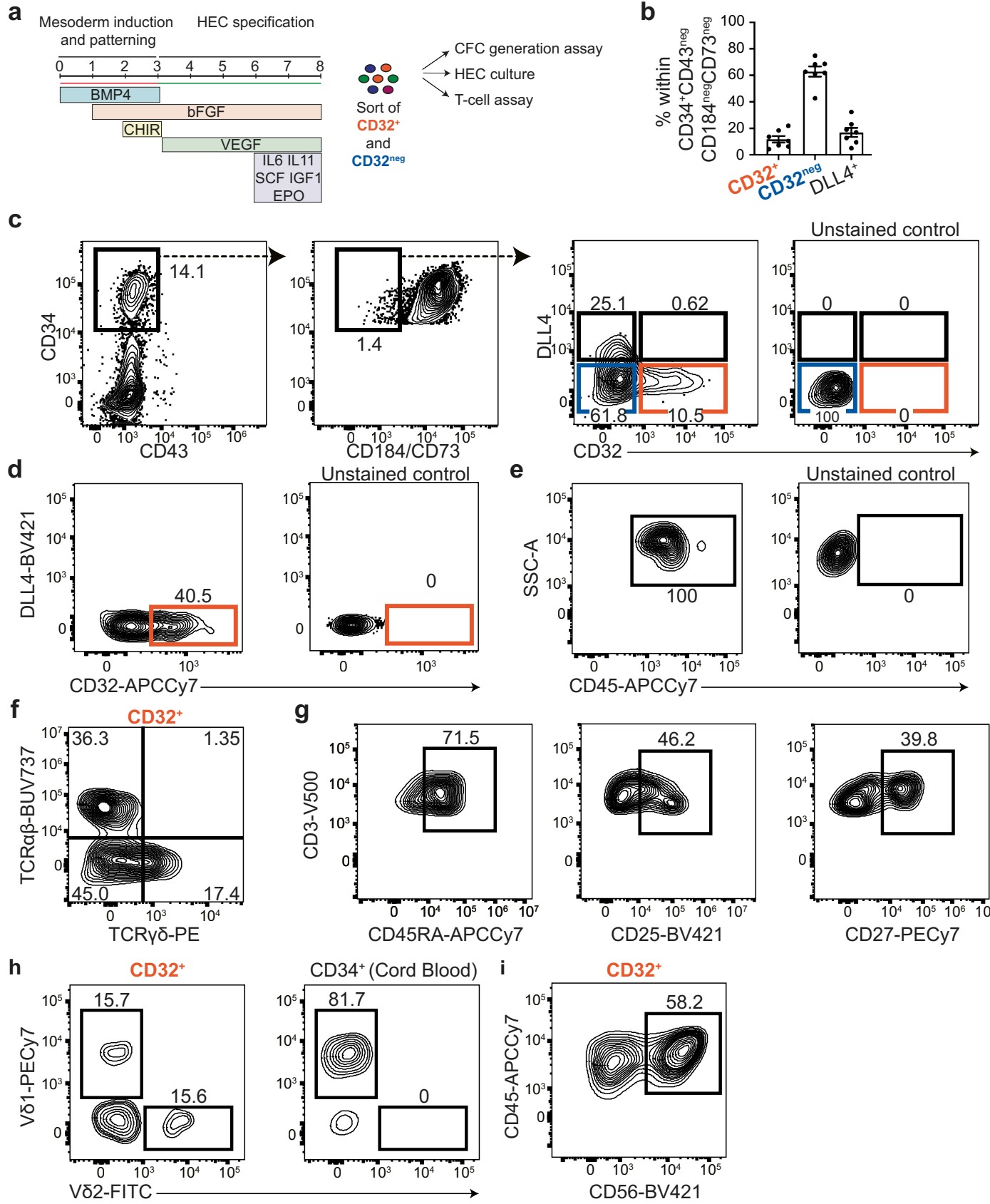

**Extended Data Fig. 3 | See next page for caption.**

**Extended Data Fig. 3 | Characterization of the hematopoietic potential of CD32⁺ cells in WNTd hPSC-derived hematopoietic differentiations.**
**a**) Experimental layout showing the timeline of WNTd hPSC-derived hematopoietic cultures obtained by adding the WNT agonist CHIR 99021. CD34$^+$CD43$^{neg}$CD184$^{neg}$CD73$^{neg}$DLL4$^{neg}$CD32$^{+/neg}$ (referred to as CD32$^+$, CD32$^{neg}$) cells were isolated at day 8 and further cultured to assay the CFC generation, T-lymphoid potential or the generation of hematopoietic progeny through HEC culture; **b**) Bar plot showing the frequency of DLL4$^{neg}$CD32$^{+/neg}$ cells and DLL4$^+$ within day 8 of WNTd hPSC-derived CD34$^+$CD43$^{neg}$CD184$^{neg}$CD73$^{neg}$ cells. Mean ± SEM. H1 hESCs $n$ = 7, independent; **c**) Representative flow cytometric analysis showing the gating strategy to isolate CD32$^+$ (orange) and CD32$^{neg}$ (blue) cells from day 8 of WNTd hPSC-derived hematopoietic cultures. DLL4$^+$ cells are highlighted in black. From the left, first panel: gated on SSC/FSC/Live. Second panel: gated on SSC/FSC/Live/CD34$^+$CD43$^{neg}$. Third panel: gated on SSC/FSC/Live/ CD34$^+$CD43$^{neg}$CD184$^{neg}$CD73$^{neg}$. Fourth panel: unstained control, gated on SSC/FSC/Live. $n$ = 7, independent; **d**) Representative flow cytometric analysis of the generation of CD32$^+$ cells from CD32$^{neg}$ cells isolated at day 8 of WNTd hPSC-derived hematopoietic cultures and cultured for 2 extra days using the same culture conditions. Gated on SSC/FSC/Live/CD34$^+$CD43$^{neg}$CD184$^{neg}$CD73$^{neg}$. H9 hESCs, $n$ = 3, independent; **e**) Representative flow cytometric analysis of the

hematopoietic CD45$^+$ progeny derived from CD32$^+$ cells isolated 2 days after the sorting of the CD32$^{neg}$ fraction at day 8 of WNTd hPSC-derived hematopoietic cultures as shown in d). Gated on SSC/FSC/Live. H9 hESCs, $n$ = 3, independent; **f**) Representative flow cytometric analysis of TCRs expression from CD32$^+$ cells isolated at day 8 of WNTd hPSC-derived hematopoietic cultures and then cultured as artificial thymic organoid (ATO)[67] for 6 weeks. Gated on SSC/FSC/Live/CD45$^+$CD3$^+$. H9 hESCs, $n$ = 4, independent; **g**) Representative flow cytometric analysis of CD3 and markers of T cell maturation and activation (CD45RA, left panel; CD25, middle panel and CD27, right panel) from CD32$^+$ cells isolated at day 8 of WNTd hPSC-derived hematopoietic cultures and then cultured as artificial thymic organoid (ATO)[67] for 6 weeks. Gated on SSC/FSC/Live/CD45$^+$CD3$^+$. H9 hESCs, $n$ = 2, independent; **h**) Representative flow cytometric analysis of Vδ2 and Vδ1 from CD32$^+$ cells isolated at day 8 of WNTd hPSC-derived hematopoietic cultures and Cord blood-derived CD34$^+$ cells cultured as artificial thymic organoid (ATO)[67] for 6 weeks. Gated on SSC/FSC/Live/CD45$^+$CD56$^{neg}$TCRγδ$^+$. H9 hESCs, $n$ = 3, independent; **i**) Representative flow cytometric analysis showing CD45$^+$CD56$^+$ NK-cells derived from CD32$^+$ cells isolated at day 8 of WNTd hPS cell-derived hematopoietic culture. Gated on SSC/FSC/Live. H1 hESCs. $n$ = 3, independent.

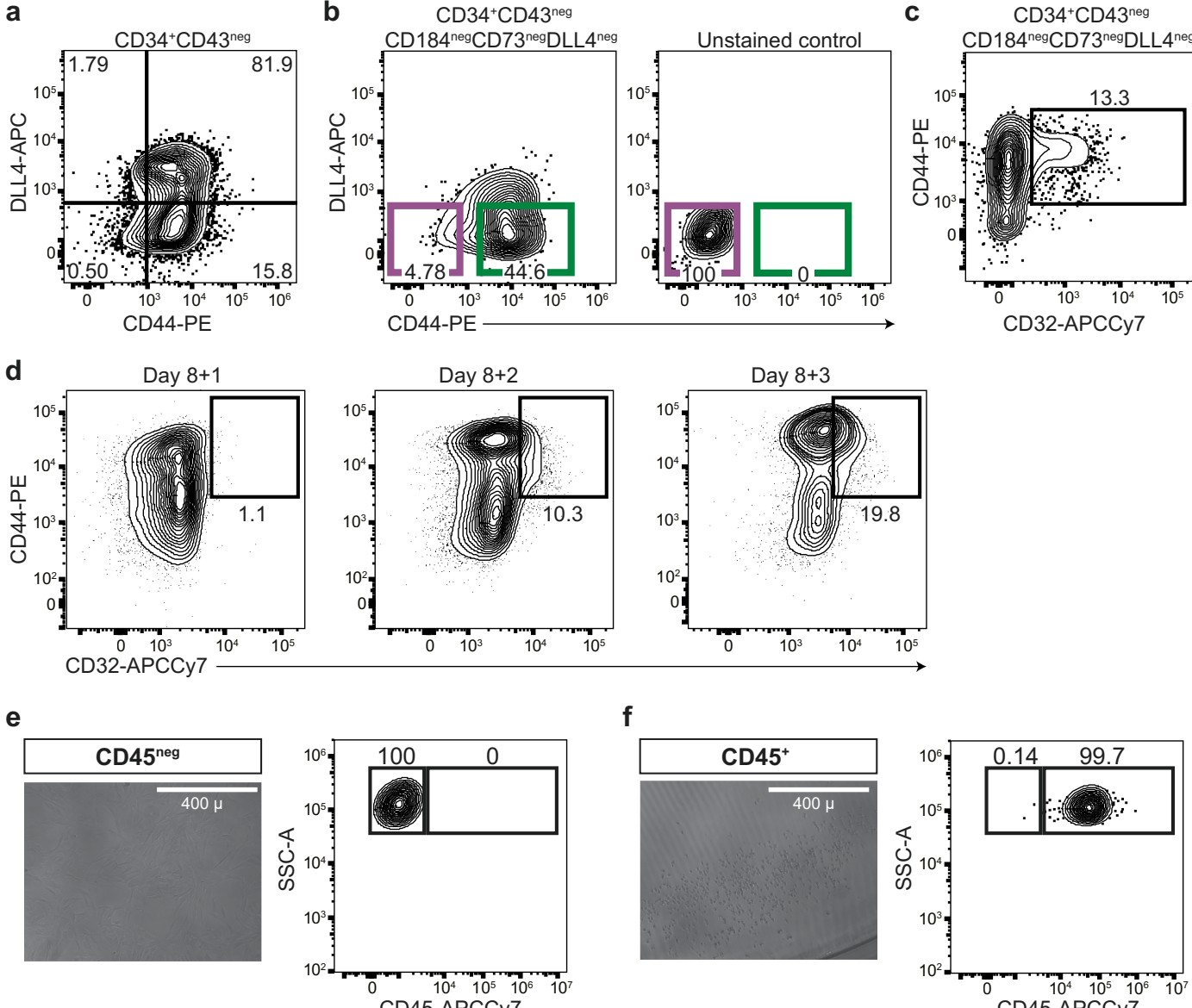

**Extended Data Fig. 4 | Characterization of the relationship between CD32+ and CD44+ cells in WNTd hPSC-derived hematopoietic differentiations.**
**a)** Representative flow cytometric analysis of CD44 and DLL4 expression in day 8 WNTd hPSC-derived CD34+CD43neg cells. Gated on SSC/FSC/Live/CD34+CD43neg. H1 hESCs, *n* = 4, independent; **b)** Representative flow cytometric analysis of CD44 and DLL4 expression in day 8 WNTd hPSC-derived CD34+CD43negCD184negCD73neg cells. Two populations are highlighted within DLL4neg fraction: CD44+ in green and CD44neg in purple. Left panel: gated on SSC/FSC/Live/ CD34+CD43negCD184negCD73neg. Right panel, unstained control, gated on SSC/FSC/Live. H1 hESCs, *n* = 4, independent; **c)** Representative flow cytometric analysis of CD32 and CD44 expression in day 8 WNTd hPSC-derived

hematopoietic cultures. Gated on SSC/FSC/Live/CD34+CD43negCD184negCD 73negDLL4neg. *n* = 3, independent; **d)** Representative flow cytometric analysis of CD32 and CD44 expression in day 8 + 1 (left panel), day 8 + 2 (middle panel) and day 8 + 3 (right panel) after HEC culture of day 8 WNTd CD44+CD32neg cells. Gated on SSC/FSC/Live/CD34+CD45neg. H9 hESCs, *n* = 3, independent; **e)** and **f)** Photo-micrograph (left panel) and representative CD45 flow cytometric analysis of a clone composed by adherent non-hematopoietic cells **e)** or round hematopoietic cells **f)** derived from single CD32+ cell isolated at day 8 of WNTd hPSC-derived hematopoietic culture. Gated on SSC/FSC/Live. H1 hESCs, *n* = 3, independent.

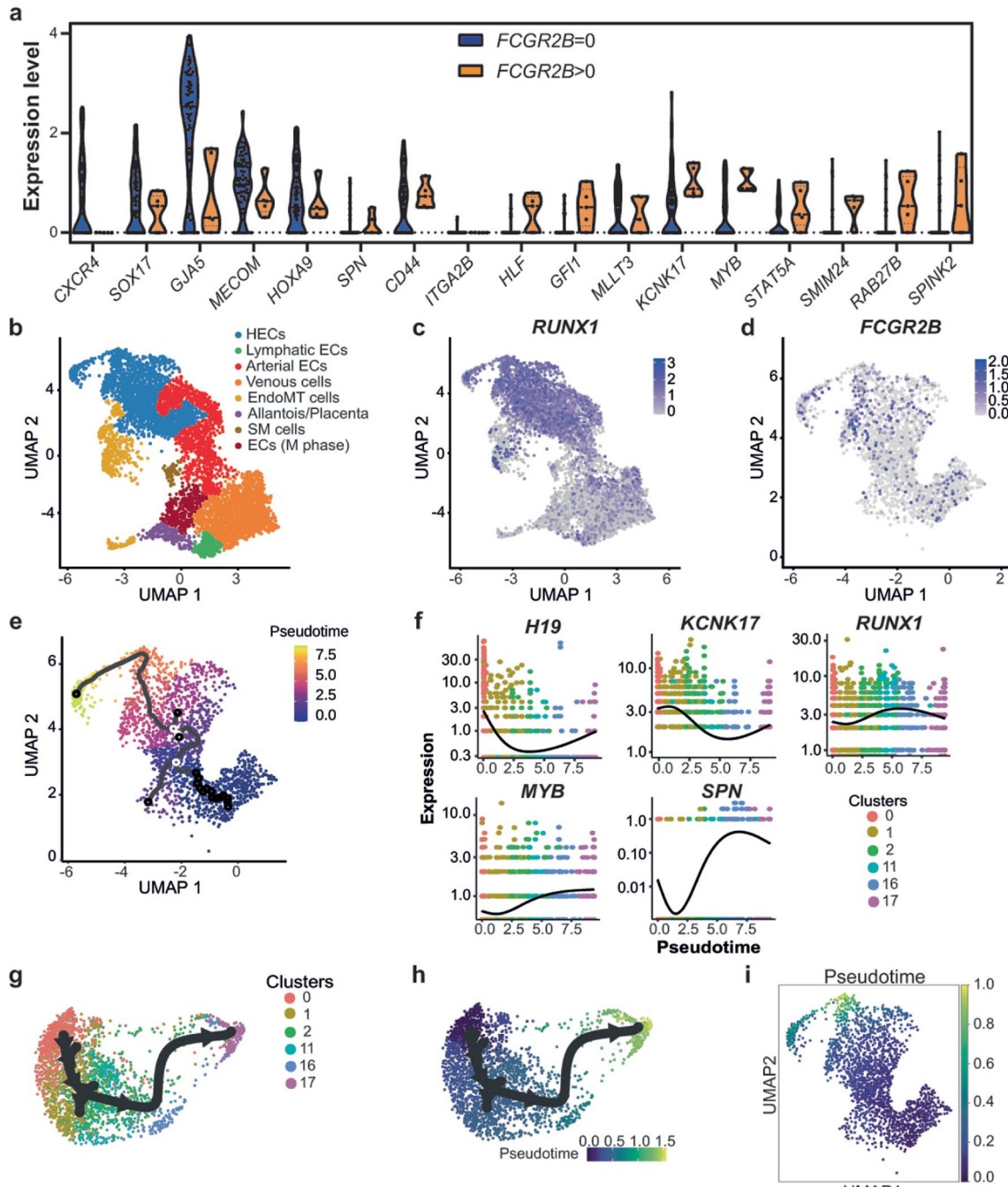

**Extended Data Fig. 5 | Single-cell transcriptomic analysis identifies the heterogeneity of hPSC-derived HECs. a**) Violin plot showing normalized expression level of landmark genes as reported in[38]. Differential expression was evaluated in CDH5 + RUNX1+PTPRCneg cells that either express FCGR2B (FCGR2B > 0), in orange, or do not (FCGR2B = 0), in blue; **b**) UMAP visualization of manually annotated cells and colour-coded by cell type. H1 hESCs, n = 1. HECs: hemogenic endothelial cells; ECs: endothelial cells; EndoMT: endothelial-to-mesenchyme transition; M phase: mitotic phase; **c**) Feature plot showing RUNX1 expression across clusters as in Fig. 5a; **d**) Feature plot showing FCGR2B expression across clusters 0, 1, 2, 11, 16, 17 that display RUNX1 as a differential marker; **e**) UMAP visualization of the pseudotime analysis by monocle3 showing the principal graph nodes and trajectory on the clusters with differential

expression of RUNX1. Cells are coloured according to pseudotime; **f**) Pseudotime kinetics of the expression alteration of selected genes (H19, KCNK17, RUNX1, MYB, SPN) along the clusters with differential expression of RUNX1 (clusters 0, 1, 2, 11, 16, 17). Cells are coloured by the cluster identity. Lines denote relative average expression of each gene in pseudotime; **g**) UMAP visualization of the single-cell trajectory, as inferred by the PAGA-tree algorithm, focusing on clusters displaying RUNX1 as a differential marker; **h**) UMAP representation of the pseudotime analysis conducted using the PAGA-tree algorithm, highlighting cells from clusters 0, 1, 2, 11, 16, and 17, with color-coding corresponding to their respective pseudotime values;. **i**) UMAP visualization of the pseudotime analysis performed by CellOracle, illustrating cells from clusters 0, 1, 2, 11, 16, and 17, with each cell color-coded according to its pseudotime value.

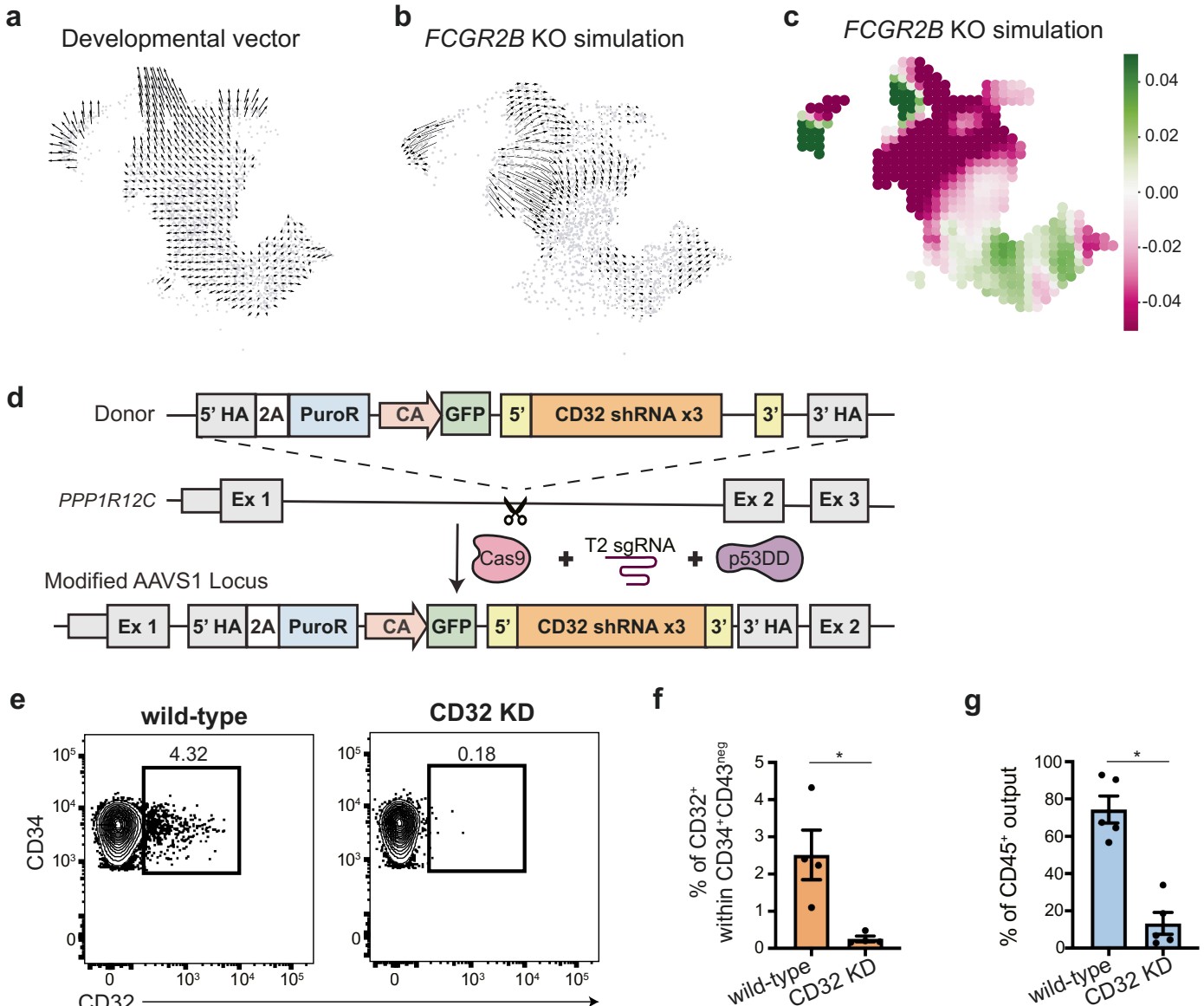

**Extended Data Fig. 6 | Downregulation of CD32 signalling impairs hematopoietic development. a)** Quiver plot depicting summarized cell-state transition vectors, as simulated by CellOracle from clusters 0, 1, 2, 11, 16, and 17, with arrows indicating the predicted direction of developmental flow; **b)** Quiver plot illustrating the simulated cell-state shift vectors resulting from an in-silico CD32 KO, as modeled by CellOracle, with arrows highlighting the predicted alterations in developmental flow due to the *FCGR2B* knockout; **c)** Digitized grid representation of the CD32 KO simulation by CellOracle, with each section colored relying on Perturbation Scores (PS). Green indicates a negative PS, suggesting that the transcription factor perturbation inhibits differentiation, while purple signifies a positive PS, indicating promotion of differentiation. Single-cell transition vectors are aggregated at specific grid points; **d)** Representative map of the donor plasmid used to engineer H1 hPSCs to

express 3x shRNAs to silence *FCGR2B* expression. Donor plasmid was inserted in AAVS1 locus by nucleofection with eCas9, the T2 gRNA plasmid, and p53 dominant negative (DD); **e)** Representative flow cytometry analysis showing CD32 and CD34 expression in day 8 wild-type (left panel) or CD32 KD (right panel) WNTd hPSC-derived CD34$^+$CD43$^{neg}$ cells. Gated on SSC/FSC/Live/CD34$^+$CD43$^{neg}$. $n = 4$, independent; **f)** Bar plot showing the frequency of CD32$^+$ within CD34$^+$CD43$^{neg}$ cells at day 8 of WNTd hPSC-derived hematopoietic culture as in **e)**. Mean ± SEM. One-tail paired Student's $t$-test, nonparametric, for all biological replicates ($n = 4$,), *p = 0.02203. **g)** Bar plot showing the frequency of CD45$^+$ derived from CD34$^+$ cells isolated at day 8 of WNTd hPSC-derived hematopoietic culture as in Fig. 4g. Mean ± SEM. One-tail paired Student's $t$-test, nonparametric, for all biological replicates ($n = 5$), *p = 0.0312.

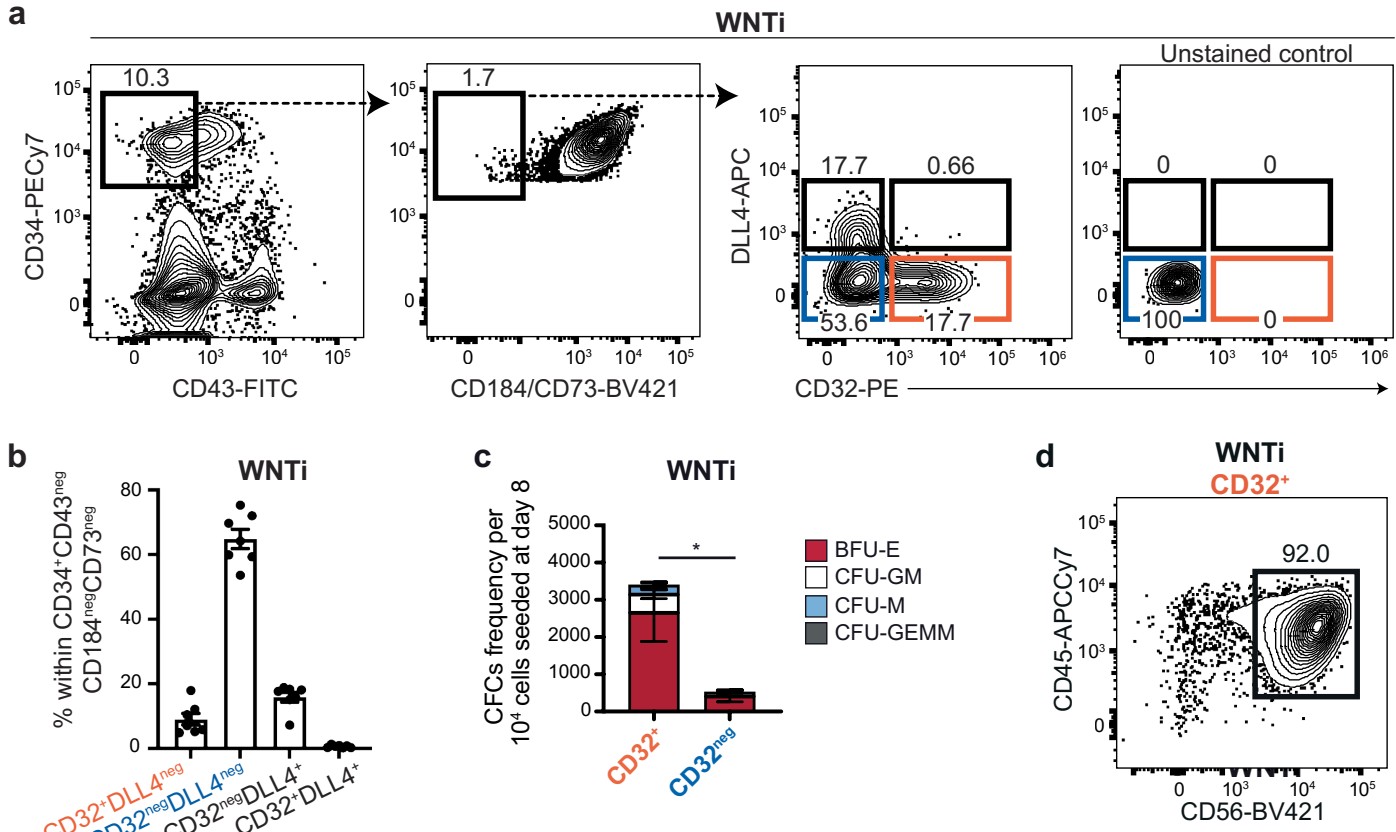

**Extended Data Fig. 7 | CD32 is expressed in WNT-independent hPSC-derived HECs. a**) Representative flow cytometric analysis showing the gating strategy to isolate CD32+ (orange) and CD32neg (blue) cells from day 8 of WNTi hPSC-derived. From the left, first panel: gated on SSC/FSC/Live. Second panel: gated on SSC/FSC/Live/CD34+CD43neg. Third and fourth panels: gated on SSC/FSC/Live/CD34+CD43negCD184negCD73neg. Fourth panel: unstained control. *n* = 7, independent; **b**) Bar plot showing the frequency of CD32+/negDLL4+/neg cells within day 8 of WNTi hPSC-derived CD34+CD43negCD184negCD73neg cells. Gated on SSC/FSC/Live/CD34+CD43negCD184negCD73neg. *n* = 7, independent, mean ± SEM; **c**) Quantification of erythro-myeloid CFC potential of CD32+/neg populations isolated at day 8 of WNTi hPSC-derived hematopoietic cultures. Mean ± SEM. One-tail paired Student's *t*-test, nonparametric, for all biological replicates (*n* = 6, H1 hESCs), considering the total number of colonies, *p = 0.0111; **d**) Representative flow cytometric analysis showing CD45+CD56+ NK-cells derived from CD32+ cells isolated at day 8 of WNTi hPSC-derived hematopoietic culture. Gated on SSC/FSC/Live. H1 hESCs. *n* = 3, independent.

| | |
|---|---|

# Reporting Summary

## Statistics

For all statistical analyses, confirm that the following items are present in the figure legend, table legend, main text, or Methods section.

| n/a | Confirmed | |
|---|---|---|
| ☐ | ☒ | The exact sample size (*n*) for each experimental group/condition, given as a discrete number and unit of measurement |
| ☐ | ☒ | A statement on whether measurements were taken from distinct samples or whether the same sample was measured repeatedly |
| ☐ | ☒ | The statistical test(s) used AND whether they are one- or two-sided<br>*Only common tests should be described solely by name; describe more complex techniques in the Methods section.* |
| ☐ | ☒ | A description of all covariates tested |
| ☐ | ☒ | A description of any assumptions or corrections, such as tests of normality and adjustment for multiple comparisons |
| ☐ | ☒ | A full description of the statistical parameters including central tendency (e.g. means) or other basic estimates (e.g. regression coefficient) AND variation (e.g. standard deviation) or associated estimates of uncertainty (e.g. confidence intervals) |
| ☐ | ☒ | For null hypothesis testing, the test statistic (e.g. *F*, *t*, *r*) with confidence intervals, effect sizes, degrees of freedom and *P* value noted<br>*Give P values as exact values whenever suitable.* |
| ☒ | ☐ | For Bayesian analysis, information on the choice of priors and Markov chain Monte Carlo settings |
| ☒ | ☐ | For hierarchical and complex designs, identification of the appropriate level for tests and full reporting of outcomes |
| ☒ | ☐ | Estimates of effect sizes (e.g. Cohen's *d*, Pearson's *r*), indicating how they were calculated |

*Our web collection on statistics for biologists contains articles on many of the points above.*

## Software and code

Policy information about availability of computer code

| | |
|---|---|
| Data collection | For RNA sequencing, total RNA from human embryonic samples was prepared using Clontech SMART-Seq v4 Ultra Low Input RNA kit and sequenced using Illumina HiSeq 4000 with 1x50 single reads. Total RNA from human pluripontet stem cell-derived cultures was prepared using ReliaPrep RNA Cell Miniprep System, RNA-Seq libraries were generated using the Smart-seq2 method and sequenced using Illumina 395 NovaSeq6000 1x100 single reads. Reads were aligned to GENCODE GRCh38 using STAR v2.7.6a with standard parameters.<br><br>For single cell RNA-sequencing, cells from day 8 differentiation culture condition were methanol-fixed as previously described (doi:10.1186/s12915-017-0383-5). Libraries were prepared following the manufacturer's instructions using the Chromium platform (10x Genomics, Pleasanton, CA) with the 3' gene expression (3' GEX) V3 kit, using an input of ~10,000 cells. Libraries were sequenced in paired end mode on a NovaSeq instrument (Illumina, San Diego, CA) targeting a depth of 50,000-100,000 reads per cell. Sequencing reads were processed and akigned to GRCh38 using the Cell Ranger software pipeline (v4.0.0)<br><br>Flow cytometric data was collected using BD FACSDiva or Beckman Coulter CytExpert. |
| Data analysis | For RNA sequencing datasets, raw reads quality control was accomplished using the FastQC tool (http://www.bioinformatics.babraham.ac.uk/projects/fastqc) and read trimming was performed using the Trim Galore software (https://doi.org/10.5281/zenodo.5127899) to remove residual adapters and low-quality sequences. Trimmed reads were aligned against the human reference genome (GRCh38) using STAR (Dobin et al., 2013; https://doi.org/10.1093/bioinformatics/bts635) with standard parameters. Uniquely mapped reads were then assigned to genes using the featureCounts tool from the Subread package (Liao et al., 2014; https://doi.org/10.1093/bioinformatics/btt656), considering the GENCODE primary assembly v.34 gene transfer file (GTF) as reference annotation for the genomic features. Gene count matrices were then processed by using the R/Bioconductor differential gene expression analysis package DESeq2 (Love et al., 2014; https://doi.org/10.1186/s13059-014-0550-8) applying the standard workflow. For the human embryos' dataset, a paired analysis was set up modeling gene counts |

using the following design formula: ~donor + condition. Gene p-values were corrected for multiple testing using FDR. Genes with adjusted p-values < 0.05 were considered differentially expressed. For single cell RNA sequencing gene counts for each cell were quantified with the Cell Ranger 'count' command with default parameters. The human genome (GRCh38.p13) was used as the reference. The resultant gene expression matrix was imported into the R statistical environment (version 4.0.3) for further analyses. Cell filtering, data normalization, and clustering were carried out using the R package Seurat (Stuart et al., 2019; https://doi.org/10.1016/j.cell.2019.05.031) v3.2.2. For each cell, the percentage of mitochondrial genes, number of total genes expressed and cell cycle scores (S and G1 phase) were calculated. Cells with a ratio of mitochondrial vs. endogenous gene expression > 0.2 were excluded as putative dying cells. Cells expressing <200 or >6,000 total genes were also discarded as putative poorly informative cells and multiplets, respectively. Cell cycle scores were calculated using the 'CellCycleScoring' function that assigns to each cell a score based on the expression of the S and G2/M phase markers and stores the S and G2/M scores in the metadata along with the predicted classification of the cell cycle state of each cell. Counts were normalized using Seurat function 'NormalizeData' with default parameters. Expression data were than scaled using the 'ScaleData' function, regressing on the number of unique molecular identifier, the percentage of mitochondrial gene expression, and the difference between S and G2M scores. By using the most variable genes, dimensionality reduction was then performed with principal component analysis (PCA) by calculating 100 PCs and selecting the top 55 PCs. Uniform Manifold Approximation and Projection (UMAP) dimensionality reduction (McInnes et al., 2018) was performed on the calculated principal components to obtain a 2D representation for data visualization. Cell clusters were identified using the Louvain algorithm at resolution r = 0.6, implemented by the 'FindCluster' function of Seurat. To find the differentially expressed (marker) genes from each cluster, the 'FindAllMarkers' function (iteratively comparing one cluster against all the others) from the Seurat package was used with the following parameters: adjusted P values <0.05, average log FC >0.25, and percentage of cells with expression > 0.1. A comprehensive manual annotation of the cell types was performed using the previously obtained markers list. Differentially expressed genes between cells of cluster 11 against cells of clusters 0,1, and 2 and clusters 16 and 17 were determined by the 'FindMarkers' function using the following parameters adjusted P values <0.05, |average log FC| > 0, and percentage of cells with expression > 0. GSEA was then performed considering Gene Ontology (GO) Biological Process (BP) terms from the C5 collection of the Molecular Signatures Database (MSigDB version 7.2) using the R/Bioconductor package clusterProfiler51 (v 3.8.1, http://bioconductor.org/packages/release/bioc/html/clusterProfiler.html). ORA was computed on the significantly differentially expressed genes considering Gene Ontology (GO) Biological Process (BP) terms from the C5 collection of the Molecular Signatures Database (MSigDB version 7.2) and the Reactome Pathways Database using the R/Bioconductor package 51 (v 3.8.1). P-values were corrected for multiple testing using FDR and enriched terms with an adjusted p-value less than 0.05 were considered statistically significant. Barplot was constructed using the R package ggplot2 (https://ggplot2.tidyverse.org). Single-cell RNA-seq samples from the public dataset GSE162950 were retrieved and processed as described in Calvanese et al., 2022 (https://doi.org/10.1038/s41586-022-04571-x) . The 'DotPlot' function from the Seurat R package was used to construct a scorecard highlighting the expression pattern of selected cells with specific expression patterns. Pseudotime trajectory was constructed using Monocle3 (version 0.2.3) (https://cole-trapnell-lab.github.io/monocle3/). Expression and feature data were extracted from the Seurat object and a Monocle3 'cell_data_set' object was constructed. The processed data was normalized followed by Principal Component Analysis (PCA) analysis using the Monocle3 function 'preprocess_cds'. Dimensionality reduction was performed using the 'reduceDimension' function. Trajectory graph learning and pseudo-time measurement through reversed graph embedding were performed with 'learn_graph' function. Cells were ordered along the trajectory using the 'orderCells' method with default parameters. The 'plot_cells' function was used to generate the trajectory plots.

To corroborate the findings from Monocle3, a trajectory inference analysis was conducted using the dynverse workflow, a component of the R package dyno (version 0.1.2).The Dynbenchmark utility, which offers a comprehensive framework for selecting the most suitable trajectory inference method according to the available experimental data, was utilized via the 'guidelines_shiny()' function. Following these guidelines, the trajectory inference analysis was performed using the Partition-based graph abstraction (PAGA)-tree algorithm. Input data for dyno, including gene expression matrices, dimensionality reduction coordinates, clustering information, and cell metadata, were derived from Seurat output and processed using the 'wrap_expression()' function. The cell trajectory was subsequently calculated by dyno using the 'infer_trajectory()' function, employing the 'ti_paga_tree()' method. Trajectory paths and pseudotime values were visualized on UMAP coordinates through the 'plot_dimred()' function provided by dyno.

To investigate the role of the CD32 gene in HEC ontogeny, genetic knockouts were simulated using the CellOracle tool (version 0.12.0). CellOracle integrates a gene regulatory network (GRN) with pseudotime analysis to predict shifts in cellular identities resulting from gene perturbations. This tool simulates alterations in gene expression due to perturbations and compares these changes to the cell's developmental trajectory within the GRN. This comparison allows for the estimation of transition probabilities between different cell states along the pseudotime axis. Following this, CellOracle generates a transition trajectory graph, illustrating the potential shifts in cellular identities after perturbation. This analysis was performed in a Python (version 3.8) environment, using Jupyter notebooks. Single-cell RNA sequencing (scRNA-seq) data, initially processed with Seurat, were converted to AnnData format using the anndata2ri tool (https://github.com/theislab/anndata2ri), ensuring content preservation for subsequent analysis. The CellOracle object construction utilized this data. Highly variable genes, critical for downstream analysis, were identified using the scanpy.pp.filter_genes_dispersion() function from scanpy, specifying n_top_genes=3000. A preliminary gene regulatory network (GRN) was constructed using the oracle.get_links() function within the Oracle() class, based on ligand-receptor interactions from the CellTalkDB database. This base GRN was further refined by incorporating the CD32 gene and its interactors, as identified in the STRING database (https://string-db.org/). Pseudotime analysis was conducted using the Pseudotime_calculator() class, employing the PAGA method from scanpy and integrating it into the CellOracle framework. This analysis culminated in the creation of a pseudotime gradient vector field with the Gradient_calculator() class from CellOracle, depicting the normal developmental trajectory. Subsequently, in-silico perturbation of CD32 expression and simulation of resultant cell identity shifts were performed using the simulate_shift() and estimate_transition_prob() functions from the Oracle class. To compare the effects of CD32 perturbation with normal development, the Oracle_development_module() class was used to calculate Perturbation Scores (PS) by computing the inner product of the respective vector fields with the calculate_inner_product() function.

All flow cytometric data was analyzed using FlowJo (v10)

For manuscripts utilizing custom algorithms or software that are central to the research but not yet described in published literature, software must be made available to editors and reviewers. We strongly encourage code deposition in a community repository (e.g. GitHub). See the Nature Portfolio guidelines for submitting code & software for further information.

# Data

Policy information about availability of data

All manuscripts must include a data availability statement. This statement should provide the following information, where applicable:
- Accession codes, unique identifiers, or web links for publicly available datasets
- A description of any restrictions on data availability
- For clinical datasets or third party data, please ensure that the statement adheres to our policy

All new gene expression analysis datasets are available in the Gene Expression Omnibus (GEO) under the accession number GSE199578 and GSE223223. Accession is currently private, protected by a password and is scheduled to be released upon publication.
GSE199578 is used in Fig 1b-e; Fig 2a; Ext. Data Fig. 1d-e; Ext. Data Fig. 2a.
The GSE223223 dataset contains:
Subseries GSE223221 (Ext. Data Fig. 4a)
Sample GSM6943617 (Fig 4a-e; Ext. Data Fig. 4b-g)
Single-cell RNA-seq samples from the public dataset GSE162950 were retrieved and processed as described in Calvanese et al., 2022 (https://doi.org/10.1038/s41586-022-04571-x).
Reads were aligned against the human reference genome (GRCh38) using STAR v2.7.6a with GENCODE GRCh38.p13 version 34
Scripts used for data analysis and for the generation of all the figures in the paper are available at this link http://www.bioinfotiget.it/gitlab/custom/scarfo_hec2023.

# Human research participants

Policy information about studies involving human research participants and Sex and Gender in Research.

**Reporting on sex and gender**

The reference to human participants included in this study refers to the patients who donate aborted tissue after signing the consent form after voluntary interruption of pregnancy.
Sex of the fetuses was not determined nor collected. All information derived from human material is completely anonymous, so they cannot in any way generate sensitive personal data, which affect the patient's privacy.

**Population characteristics**

Human embryos are obtained immediately after voluntary terminations of pregnancy induced with the RU 486 antiprogestative compound. Embryos are collected in sterile cold physiological solution containing antibiotics. Embryonic age is estimated based on several anatomic criteria: number of somite pairs at Carnegie stages 9 to 11, limb bud shape and eye pigmentation at stages 12 to 15 and 16 to 17, respectively.
As most of the tissues were destined for cell culture and sorting, specimens were preferably harvested under pseudo-sterile conditions. Thus, infected samples or samples from parents at risk for infectious pathologies (HIV, CMV, Hep-B-C, etc.) were excluded. Abortions performed on minors will not be sampled.

**Recruitment**

The tissues were obtained from elective pregnancy terminations. The decision to terminate pregnancy had occurred prior to consent for tissue donation. The tissues were entrusted, respecting anonymity (without any identity), to the research team which assigned a simple registration code allowing neither identification nor return to the patient. The only data necessary for the researcher is the term of pregnancy (weeks of amenorrhea). All documents concerning the patients, including the signed information and consent form, are kept by the clinical department in the patient's clinical file. This is stated clearly in the consent form that the patient signs.No payment were made to donors and the donors knowingly and willingly consented to provide research materials without restrictions for research and for use without identifiers.

**Ethics oversight**

Human embryonic tissues employed for RNA sequencing, immunohistochemistry and immunofluorescence were obtained from voluntary abortions performed according to the guidelines and with the approval of the French National Ethics Committee. Written consent to the use of samples in research was obtained from patients. The study was approved by Ospedale San Raffaele Ethical Committee (TIGET-HPCT protocol) and by the Institutional Review Board of the French Institute of Medical Research and Health (Number 21-854). Human embryonic tissues employed for RNA sequencing, immunohistochemistry and immunofluorescence were obtained from voluntary abortions performed according to the guidelines and with the approval of the French National Ethics Committee. Written consent to the use of samples in research was obtained from patients. The study was approved by Ospedale San Raffaele Ethical Committee (TIGET-HPCT protocol) and by the Institutional Review Board of the French Institute of Medical Research and Health (Number 21-854).

Note that full information on the approval of the study protocol must also be provided in the manuscript.

# Field-specific reporting

Please select the one below that is the best fit for your research. If you are not sure, read the appropriate sections before making your selection.

☒ Life sciences  ☐ Behavioural & social sciences  ☐ Ecological, evolutionary & environmental sciences

For a reference copy of the document with all sections, see nature.com/documents/nr-reporting-summary-flat.pdf

# Life sciences study design

All studies must disclose on these points even when the disclosure is negative.

| | |
|---|---|
| Sample size | Experiments on human embryonic samples were performed at minimum in triplicates when feasible, i.e. in all but one case: in fact, the isolation of CD32+ and CD32neg cells from human embryonic samples for functional analysis was performed in duplicate. Differentiation experiments were performed in triplicate and including further replicates (up to seven) when feasible. Bulk RNA sequencing was performed in triplicate, as prior experience instructs is sufficient for statistical power. No statistical method was used to pre-determine sample size but sample size and replication were determined by historical controls e.g. Ditadi et al, 2015 (https://doi.org/10.1038/ncb3161).Replication was consistent with our prior publications. Single cell RNA sequencing was performed in single replicate as is standard within the field. |
| Data exclusions | Differentiation resuls were excluded when internal control failed to produce hematopoietic progenitors. |
| Replication | Experiments on human embryonic samples were performed at minimum in biological triplicate as reported in all figure legends. The isolation of CD32+ and CD32neg cells from human embryonic samples for functional analysis was performed in biological duplicate. Differentiation experiments were performed at minimum in triplicate. The specific number of replicates are indicated in all figure legends. Two additional, separate hPSC lines were employed in Figure 3g. |
| Randomization | Experimental conditions were not randomized but covariates were controlled by equal distribution of sorted cells across controls and experimental conditions. |
| Blinding | Blinding of experimental conditions was not relevant as our studies do not require grading of the results. |

# Reporting for specific materials, systems and methods

We require information from authors about some types of materials, experimental systems and methods used in many studies. Here, indicate whether each material, system or method listed is relevant to your study. If you are not sure if a list item applies to your research, read the appropriate section before selecting a response.

## Materials & experimental systems

| n/a | Involved in the study |
|---|---|
| ☐ | ☒ Antibodies |
| ☐ | ☒ Eukaryotic cell lines |
| ☒ | ☐ Palaeontology and archaeology |
| ☒ | ☐ Animals and other organisms |
| ☒ | ☐ Clinical data |
| ☒ | ☐ Dual use research of concern |

## Methods

| n/a | Involved in the study |
|---|---|
| ☒ | ☐ ChIP-seq |
| ☐ | ☒ Flow cytometry |
| ☒ | ☐ MRI-based neuroimaging |

## Antibodies

| | |
|---|---|
| Antibodies used | Anti-human uncoupled antibodies used for immunohistochemistry and immunofluoresce includes: anti-human CD34 (Beckman, QBEnd/10, 1:500), anti-human ACE (BB9, BD Biosciences, 557813, 1:50), anti-human CD32 (Biolegend, 303202, 1:750) and rabbit anti-human/mouse Runx1 (Abcam, ab92336, 1:100). Secondary biotinylated antibodies were: goat anti-mouse IgG (Jackson Immuno Research, 115-066-072, 1:1000) and goat anti-rabbit IgG antibody (Jackson Immuno Research, 111-066-144, 1:500). Double-immunofluorescence staining used the Dylight488 coupled streptavidin (Biolegend, 405218, 1:350) and the TSA Fluorescent Plus System. For hPSC and human embryonic sample FACS analysis antibodies include: CD184 BV421 (BD, 562448, 1:100), CD32 APC Cyanine 7 (BD, 303229), CD32 PE (BD; 303206), CD34 APC (Beckman Coulter, IM2472), CD34 PE Cyanine 7 (Biolegend, 343616, 1:400), CD34 PE Cyanine 7 (eBioscience, 25034942, 1:400), CD4 PE Cyanine 7 (BD, 560649), CD43 APC (BD, 560198), CD43 FITC (BD, 555475, 10:100), CD44 PE (Miltenyi, 130-113-342), CD45 APC Cyanine 7 (Biolegend, 368516, 1:100), CD45 FITC (Beckman Coulter, A07782), CD45 BV421 (Biolegend, 304032), CD5 PE (Biolegend, 300607, 5:100), CD56 BV421 (BD, 740076, 3:100), CD7 APC (BD, 561604), CD73 BV421 (BD, 562430, 3:100), CD8 PE (Biolegend, 31051), DLL4 APC (Biolegend, 346508, 1:100), DLL4 BV421 (BD, 744840, 1:100), anti-IgG1 PE (Southern Biotechnology, 1070-05), 7AAD (BD, 559925, 1:100), Fcr Blocking reagent human (Miltenyi, 130-059-901, 20:100), CD3 V550 (BD, 561416, 1:30), CD45RA APC Cyanine 7 (Biolegend, 304128, 1:100), CD27 PE Cyanine 7, (eBioscience, 25027942, 1:100), CD25 BV421 (BD, 582442, 1:50), TCRb (BD, 749196, 1:100), TCRgd (Beckman Coulter, B49176, 1:100), CD45 BUV395 (BD, 563792), CD8 PerCP-Cy5.5 (Biolegend, 344710, 1:100), CD4 BV605 (BD, 562658, 1:50), TCRVd1 (Beckman Coulter, B49309, 1:50), TCRVd2 (Beckman Coulter, IM1464, 1:50), <br> All the antibodies used for FACS analysis were used at 1:200 dilution unless differently specified. |
| Validation | All antibodies are commercially validated. Validation statements found on manifacturer's website indicate the following validations: ACE (human bone marrow); CD32, CD34, CD4, CD43, CD45, CD44, CD5, CD56, CD7, CD73, CD3, CD45RA, CD27, CD25, TCRab, TCRgd, CD8,CD4, TCRVd1, TCRVd2 (human peripheral blood), FcR Blocking reagent (THP-1 cell line), RUNX1 (Molt-4 cell line). <br> Each lot of an antibody is tested for conformance with charachteristics of a standard reagent and representative flow cytometric data is included in data sheets to demonstrate specificity and/or sensitivity to a relevant cell population. |

# Eukaryotic cell lines

Policy information about cell lines and Sex and Gender in Research

| | |
|---|---|
| Cell line source(s) | WA01 (H1) and WA09 (H9) were obtained from WiCell Stemcell bank. OP9DLL4 and OP9DLL1 were generated as described in Schmitt et al 2002 (https://doi.org/10.1016/S1074-7613(02)00474-0), Mohtashami et al, 2010 (https://doi.org/10.4049/jimmunol.1000782). MS5-DLL4 were a kind gift from Dr. Tom Taghon who described their generation in Dolens et al, 2020 (https://doi.org/10.15252/embr.201949006) |
| Authentication | Cell lines were authenticated. |
| Mycoplasma contamination | H1, H9, OP9DDL4 and OP9DDL1 were tested negative for mycoplasma contamination. |
| Commonly misidentified lines (See ICLAC register) | No commonly misidentified lines in the ICLAC registry are used in this study. |

# Flow Cytometry

## Plots

Confirm that:

☒ The axis labels state the marker and fluorochrome used (e.g. CD4-FITC).

☒ The axis scales are clearly visible. Include numbers along axes only for bottom left plot of group (a 'group' is an analysis of identical markers).

☒ All plots are contour plots with outliers or pseudocolor plots.

☒ A numerical value for number of cells or percentage (with statistics) is provided.

## Methodology

| | |
|---|---|
| Sample preparation | Day 8 cells were trypsinized for 8 minutes and washed in IMDM, 10% FBS, 10 ug/ml DNAse. They were further dissociated with Collagenase II for 30 minutes and washed in Stempro 34. Human embryonic tissues were dissociated for 30-60 minutes with Collagenase I or Collagenase/Dispase. All samples were stained in Stempro 34 medium. |
| Instrument | Cells were sorted with FACSAria II (BD). FACS-analysis were performed at FACS Canto (BD Biosciences) Cytoflex S or Cytoflex LX (both Beckman Coulter). |
| Software | BD FACS Diva and Coulter CytExpert were used for data acquisition and FLowJo was used for analysis. |
| Cell population abundance | Single cells or 100-30000 CD32+ or CD44+ cells were isolated for each WNTd experiment. 100-30000 CD32neg cells were isolated for each WNTd experiment. |
| Gating strategy | Viable cells were gated using FSC-A/SSC-A and doublets were removed using FSC-H/FSC-W ans SSC-H/SSC-W. Autoflorescent cells were removed using the PerCP channel. 7AAD or Maleimide stain was used on human embryonic sample to remove the dead cells. Representative flow plots and gating strategy demonstrated in Extended Data Figure 1c, 2c, 2d, 3c, 8a<br>Figure 2e is gated on CD34+CD43negCD45neg.<br>Figure 3a, 3f, 4f, 4g, Extended data 4d, Extended data 4e, Extended data 4f, Extended data 6n, Extended data 7d, Extended data 7g and Extended Data Figure 8b are gated on live cells.<br>Figure 3b, Extended Data Figure 3d and Extended data 7b are gated on CD34+CD43negCD184negCD73neg cells.<br>Extended Data Figure 3f on CD34+CD43neg cells.<br>Figure 3c, Extended Data Figure 4a and Extended Data Figure 4b are gated on CD34+CD43negCD184negCD73negDLL4neg cells.<br>Figure 3h is gated on CD45+CD56negCD7+CD5+ cells.<br>Extended Figure 3f is gated on CD45+GFPnegCD3+ cells<br>Extended Figure 3g is gated on CD45+CD56neg cells<br>Extended Figure 3h is gated on CD45+CD56negTCRgd+ cells |

☒ Tick this box to confirm that a figure exemplifying the gating strategy is provided in the Supplementary Information.

