## [Peer Review File · Nature Cell Biology]

Peer Review Information

Journal: Nature Cell Biology

Manuscript Title: CD32 captures committed haemogenic endothelial cells during human embryonic development

Corresponding author name(s): Dr Andrea Ditadi

Editorial Notes:

comments, excluding minor textual revisions, have been copied into this Peer Review File.

Reviewer Comments & Decisions:

Decision Letter, initial version:

Dear Andrea,

Your manuscript, "CD32 allows capturing blood cells emergence in slow motion during human embryonic development", has now been seen by 3 referees, who are experts in developmental hematopoiesis, clonal analysis, scRNAseq (referee 1); developmental hematopoiesis, hemogenic endothelium, scRNAseq (referee 2); and human hematopoiesis, clonal analysis, iPSCs (referee 3). As you will see from their comments (attached below) they find this work of potential interest, but have raised substantial concerns, which in our view would need to be addressed with considerable revisions before we can consider publication in Nature Cell Biology.

Nature Cell Biology editors discuss the referee reports in detail within the editorial team, including the chief editor, to identify key referee points that should be addressed with priority, and requests that are overruled as being beyond the scope of the current study. To guide the scope of the revisions, I have listed these points below. I should stress that the referees' concerns point to a premature dataset and these points would need to be addressed with experiments and data, and reconsideration of the study for this journal and re-engagement of referees would depend on strength of these revisions.

In particular, it would be essential to:

(A) Provide data to support claims about the role of CD32 in HECs and the capacity of CD32+ HECs to contribute to functional HSCs, as indicated by all referees:

Referee #1:

"It is uncertain whether CD32 is a critical receptor for the differentiation of hemogenic endothelium or whether this is just a consequence of maturation. To determine this, in-vitro differentiation of CD32+ cells using an anti-CD32 blocking antibody should be done or other functional experiments. Please perform sequential paneling for the endothelial to hematopoietic transition using CD43/CD45 acquisition".

"Although it is clear that CD32 is an important marker for maturation of endothelial cells towards a hemogenic fate, it is still unclear how much better this is than CD44. The data was convincing that transient culture followed by differentiation was better if the authors start from CD32+ cells; however, this may simply be an issue with timing since CD32 is acquired in culture of CD44 cells. To address this, the authors should perform sorting of the D8 CD34+/CD43-/DLL4-/CD32-/CD44+ cell fraction. In EHT conditions, perform flow cytometry for CD44/CD32 to see just how much of the culture acquires CD32 and how long this takes. It is possible that the described short term EHT culture does not allow

for sufficient in-vitro maturation of CD44+ cells".

"It is still unclear to me how the endothelium generated resembles molecularly in vivo AGM-like endothelium. It could be interesting to include a more in depth bioinformatic analysis using the Mikola lab's time course data instead of just the score card analysis and show how this compares from a trajectory standpoint. Furthermore, it could also be interesting to look at other marker genes for HSC-competent HE (for example CD27/Neur13) and see if this overlaps with CD32 expressing HE".

Referee #2:

"How does the author demonstrate that the CD32+ HEC is bona fide HEC (HSC primed) or EMP primed HEC? Whether the HECs labelled by CD32 in human embryos or human PSC hematopoietic cultures can generate transplantable HSCs after hematopoietic induction remains unknown".

Referee #3:

"Is CD32+ an obligate stage of HEC ontogeny? In Fig. S4e, HEC precursors at a specific stage do not appear to uniformly express CD32, raising the possibility that only some HECs transition through a CD32+ stage during EHT".

"In human embryo experiments, it remains unclear if CD32 marks HECs that gives rise to HSCs. CFC studies in Fig. 2f show that CD32+ cells are more hemogenic compared to CD32-, however it remains possible that HSCs arise from a distinct HEC population which may be CD32 negative".

(B) Address the questions of referee #2 requiring clarifications and/or pointing at discrepancies:

"At 23 dpf, the authors show that ACE expression is observed in the mesenchyme surrounding the DA; however, at 27 dpf, the ACE expression in the sub-aortic mesenchyme cannot be observed and is observed in the endothelial cells lining the ventral wall of DA (Fig. 1a). How to explain the changes of location or fate of the ACE+ mesenchyme from 23 dpf to 27 dpf? What is the relationship between ACE+ mesenchyme and ACE+ endothelium?"

"In the human DA, CD32 and RUNX1 show co-localization in the HECs; however, CD32 and RUNX1C-EGFP show no co-staining in the day 8 CD34+CD43negCD73negCD184negDLL4neg cells (containing HECs) in human PSC hematopoietic cultures. How to explain this difference?"

"Given that 6 HEC clusters (annotated in single-cell transcriptome atlas) form a developmental continuum (Monocle3-based trajectory analysis, Fig. 4b) featured by arterial identity loss and hematopoietic identity gain, it remains controversial whether these HEC clusters are considered as 6 heterogeneous subtypes or 6 developmental stages".

(C) Address the question by referee #3 concerning the underlying mechanisms driving hematopoietic fate specification:

"The authors perform single cell RNAseq analysis of CD32+ HECs from hPSCs. However, it remains unclear what are the mechanisms and pathways driving hematopoietic fate specification, which can be harnessed to regulate EHT and/or hematopoietic output from hPSCs. The finding that the CD32+ HEC is Notch-independent is interesting, though it essentially confirms published data showing that Notch activity and dependence progressively declines during EHT (e.g. Lizama et al. 2015; Richard et al. 2013)".

(D) All other referee concerns pertaining to strengthening existing data, providing controls, methodological details, clarifications and textual changes, should also be addressed.

(E) Finally please pay close attention to our guidelines on statistical and methodological reporting (listed below) as failure to do so may delay the reconsideration of the revised manuscript. In particular please provide:

- a Supplementary Figure including unprocessed images of all gels/blots in the form of a multi-page pdf file. Please ensure that blots/gels are labeled and the sections presented in the figures are clearly indicated.
- a Supplementary Table including all numerical source data in Excel format, with data for different figures provided as different sheets within a single Excel file. The file should include source data giving rise to graphical representations and statistical descriptions in the paper and for all instances where the figures present representative experiments of multiple independent repeats, the source data of all repeats should be provided.

We would be happy to consider a revised manuscript that would satisfactorily address these points, unless a similar paper is published elsewhere, or is accepted for publication in Nature Cell Biology in the meantime.

In contrast, although we agree with referee #2 that showing conservation in other species would provide valuable insights, we consider this point to be beyond the scope of the present study. Thus, addressing experimentally the following point by referee #2 will not be necessary for reconsideration of the manuscript at this journal:

"Is CD32 a conserved marker for enrichment of HECs in mammals, including mice?"

- ensure that it conforms to our format instructions and publication policies (see below and www.nature.com/nature/authors/).
- provide a point-by-point rebuttal to the full referee reports verbatim, as provided at the end of this letter.
- provide the completed Editorial Policy Checklist (found here <https://www.nature.com/authors/policies/Policy.pdf>), and Reporting Summary (found here <https://www.nature.com/authors/policies/ReportingSummary.pdf>). This is essential for

reconsideration of the manuscript and these documents will be available to editors and referees in the event of peer review. For more information see <http://www.nature.com/authors/policies/availability.html> or contact me.

Nature Cell Biology is committed to improving transparency in authorship. As part of our efforts in this direction, we are now requesting that all authors identified as 'corresponding author' on published papers create and link their Open Researcher and Contributor Identifier (ORCID) with their account on the Manuscript Tracking System (MTS), prior to acceptance. ORCID helps the scientific community achieve unambiguous attribution of all scholarly contributions. You can create and link your ORCID from the home page of the MTS by clicking on 'Modify my Springer Nature account'. For more information please visit www.springernature.com/orcid.

[Redacted]

We would like to receive a revised submission within six months. We would be happy to consider a revision even after this timeframe, however if the resubmission deadline is missed and the paper is eventually published, the submission date will be the date when the revised manuscript was received.

We hope that you will find our referees' comments, and editorial guidance helpful. Please do not hesitate to contact me if there is anything you would like to discuss.

Best wishes,

Stelios

Stylios Lefkopoulos, PhD
He/him/his
Associate Editor
Nature Cell Biology
Springer Nature
Heidelberger Platz 3, 14197 Berlin, Germany

E-mail: stylios.lefkopoulos@springernature.com
Twitter: @s_lefkopoulos

Reviewers' Comments:

Reviewer #1:
Remarks to the Author:

The Scarfo et al article reports the identification of specific markers for hemogenic endothelial cells (HECs) in human embryos and their potential use in generating hematopoietic cells in vitro. The authors performed transcriptomic analysis of human embryos and identified FCGR2B, which encodes the Fc receptor CD32, as highly enriched in the endothelial cell population containing HECs. Functional analysis confirmed that CD32+ endothelial cells have multilineage hematopoietic potential. The study also revealed that CD32+ HECs no longer require Notch signaling for hematopoietic commitment and display full commitment to hematopoiesis before the expression of hematopoietic markers. The findings of the ms can be significant if these provide a precise method for isolating HECs and advancing the generation of therapeutic hematopoietic cells from human pluripotent stem cells in vitro.

While the paper is done well technically, there are a significant number of concerns that need to be addressed to fully establish the utility of CD32 as a HEC marker.

1. It is uncertain whether CD32 is a critical receptor for the differentiation of hemogenic endothelium or whether this is just a consequence of maturation. To determine this, in-vitro differentiation of CD32+ cells using an anti-CD32 blocking antibody should be done or other functional experiments. Please perform sequential paneling for the endothelial to hematopoietic transition using CD43/CD45 acquisition.
2. Although it is clear that CD32 is an important marker for maturation of endothelial cells towards a hemogenic fate, it is still unclear how much better this is than CD44. The data was convincing that transient culture followed by differentiation was better if the authors start from CD32+ cells; however, this may simply be an issue with timing since CD32 is acquired in culture of CD44 cells. To address this, the authors should perform sorting of the D8 CD34+/CD43-/DLL4-/CD32-/CD44+ cell fraction. In EHT conditions, perform flow cytometry for CD44/CD32 to see just how much of the culture acquires CD32 and how long this takes. It is possible that the described short term EHT culture does not allow for sufficient in-vitro maturation of CD44+ cells.
3. The characterization of the T cell potential of CD32+ cells is very limited. In the paper, T-cell populations are displayed as enriched from CD56- populations. None of these upstream flow plots are provided. It would be nice to know how much of the cells in-vitro generate CD56+ NK cells as well. Additionally, longer term T-cell differentiation cultures to define what T-cell fractions are generated would be useful. Analysis of activation and mature markers like TCRab would be helpful.
4. It is still unclear to me how the endothelium generated resembles molecularly in vivo AGM-like endothelium. It could be interesting to include a more in depth bioinformatic analysis using the Mikola lab's time course data instead of just the score card analysis and show how this compares from a trajectory standpoint. Furthermore, it could also be interesting to look at other marker genes for HSC-competent HE (for example CD27/Neur13) and see if this overlaps with CD32 expressing HE.

Reviewer #2:

Remarks to the Author:

To capture rare and transient hemogenic endothelial cells (HECs) for characterizing endothelial-to-hematopoietic transition in human embryos, the authors performed functional and transcriptomic analyses in human primary tissues and human PSC hematopoietic cultures. They identify CD32 (encoded by FCGR2B), together with CD34 and ACE, as surface marker combinations to enrich HECs in vivo and in vitro. In the human dorsal aorta and yolk sac at 26-30 dpf, CD32 shows co-staining with classical HEC markers, such as Runx1, CD34, and ACE, in the endothelial cells; meanwhile, sorted

CD32+ endothelial fraction shows higher hematopoietic outputs than the CD32neg fraction after OP9-DL1 co-culture. In human PSC hematopoietic culture, CD32-enriched HEC fraction shows robust multilineage differentiation potential. Finally, the authors reveal that the acquisition of hematopoietic fate of CD32+ HECs is independent of Notch signaling.

Overall, this study provides some insights of the molecular characteristics during endothelial-to-hematopoietic transition (EHT) in human embryos and hPSC system in vitro. In this regard, however, EHT has been already transcriptionally characterized at single cell level in a large number of previous studies using human embryos and human PSC differentiation cultures.

Major concerns:

1. How does the author demonstrate that the CD32+ HEC is bona fide HEC (HSC primed) or EMP primed HEC? Whether the HECs labelled by CD32 in human embryos or human PSC hematopoietic cultures can generate transplantable HSCs after hematopoietic induction remains unknown.
2. At 23 dpf, the authors show that ACE expression is observed in the mesenchyme surrounding the DA; however, at 27 dpf, the ACE expression in the sub-aortic mesenchyme cannot be observed and is observed in the endothelial cells lining the ventral wall of DA (Fig. 1a). How to explain the changes of location or fate of the ACE+ mesenchyme from 23 dpf to 27 dpf? What is the relationship between ACE+ mesenchyme and ACE+ endothelium?
3. In the human DA, CD32 and RUNX1 show co-localization in the HECs; however, CD32 and RUNX1C-EGFP show no co-staining in the day 8 CD34+CD43negCD73negCD184negDLL4neg cells (containing HECs) in human PSC hematopoietic cultures. How to explain this difference?
4. Given that 6 HEC clusters (annotated in single-cell transcriptome atlas) form a developmental continuum (Monocle3-based trajectory analysis, Fig. 4b) featured by arterial identity loss and hematopoietic identity gain, it remains controversial whether these HEC clusters are considered as 6 heterogeneous subtypes or 6 developmental stages.
5. Is CD32 a conserved marker for enrichment of HECs in mammals, including mice?

Reviewer #3:

Remarks to the Author:

In this manuscript, Scarfò et al. identify CD32 as a marker of HEC and use it to purify and molecularly profile HECs in human embryos and hPSCs. The authors isolate and profile CD34+ACE+ HECs from human embryos, showing that CD32 is co-expressed on a subset of HECs. CD32+ HECs are committed to hematopoietic fate and undergo EHT in a Notch-independent fashion. This is a well-developed and technically excellent study that impressively uses human embryos to uncover a specific marker of HECs. Despite molecular characterization of CD32+ HECs, the biological insights into how hematopoietic cell fate is specified or approaches for improving hematopoietic output are somewhat limited. Specific comments below.

1. The authors perform single cell RNAseq analysis of CD32+ HECs from hPSCs. However, it remains unclear what are the mechanisms and pathways driving hematopoietic fate specification, which can be harnessed to regulate EHT and/or hematopoietic output from hPSCs. The finding that the CD32+ HEC is Notch-independent is interesting, though it essentially confirms published data showing that Notch activity and dependence progressively declines during EHT (e.g. Lizama et al. 2015; Richard et al. 2013).

2. CD32 marks only the penultimate stage of EHT. This raises the question of what are the earlier

stages of HEC ontogeny and can more specific markers of those stages be identified.

3. Is CD32+ an obligate stage of HEC ontogeny? In Fig. S4e, HEC precursors at a specific stage do not appear to uniformly express CD32, raising the possibility that only some HECs transition through a CD32+ stage during EHT.

4. In human embryo experiments, it remains unclear if CD32 marks HECs that gives rise to HSCs. CFC studies in Fig. 2f show that CD32+ cells are more hemogenic compared to CD32-, however it remains possible that HSCs arise from a distinct HEC population which may be CD32 negative.

Minor comments

1. Fig. 2e shows that nearly ~80% of CD34+ cells in the YS are also CD32+ (vs. 5% in AGM and PSCs). What is a possible reason for this apparent abundance of CD32+ in YS? Are these CD32+ cells also hemogenic?

2. Fig. 3c shows that CD32+ cells are RUNX1Cneg, however single cell analysis in Fig. 4 shows that they are RUNX1+?

3. Fig. 3e images demonstrating that CD32+ cells are bona fide EC are important and could be enlarged to better show classical endothelial morphology.

Methods should be written concisely, but should contain all elements necessary to allow interpretation and replication of the results. As a guideline, Methods sections typically do not exceed 3,000 words. The Methods should be divided into subsections listing reagents and techniques. When citing previous methods, accurate references should be provided and any alterations should be noted. Information must be provided about: antibody dilutions, company names, catalogue numbers and clone numbers for monoclonal antibodies; sequences of RNAi and cDNA probes/primers or company names and catalogue numbers if reagents are commercial; cell line names, sources and information on cell line

identity and authentication. Animal studies and experiments involving human subjects must be reported in detail, identifying the committees approving the protocols. For studies involving human subjects/samples, a statement must be included confirming that informed consent was obtained. Statistical analyses and information on the reproducibility of experimental results should be provided in a section titled "Statistics and Reproducibility".

All Nature Cell Biology manuscripts submitted on or after March 21 2016 must include a Data availability statement at the end of the Methods section. For Springer Nature policies on data availability see <http://www.nature.com/authors/policies/availability.html>; for more information on this particular policy see <http://www.nature.com/authors/policies/data/data-availability-statements-data-citations.pdf>. The Data availability statement should include:

- Accession codes for primary datasets (generated during the study under consideration and designated as "primary accessions") and secondary datasets (published datasets reanalysed during the study under consideration, designated as "referenced accessions"). For primary accessions data should be made public to coincide with publication of the manuscript. A list of data types for which submission to community-endorsed public repositories is mandated (including sequence, structure, microarray, deep sequencing data) can be found here <http://www.nature.com/authors/policies/availability.html#data>.
- Unique identifiers (accession codes, DOIs or other unique persistent identifier) and hyperlinks for datasets deposited in an approved repository, but for which data deposition is not mandated (see here for details <http://www.nature.com/sdata/data-policies/repositories>).
- At a minimum, please include a statement confirming that all relevant data are available from the authors, and/or are included with the manuscript (e.g. as source data or supplementary information), listing which data are included (e.g. by figure panels and data types) and mentioning any restrictions on availability.
- If a dataset has a Digital Object Identifier (DOI) as its unique identifier, we strongly encourage including this in the Reference list and citing the dataset in the Methods.

We recommend that you upload the step-by-step protocols used in this manuscript to the Protocol Exchange. More details can found at www.nature.com/protocolexchange/about.

All imaging data should be accompanied by scale bars, which should be defined in the legend. Cropped images of gels/blots are acceptable, but need to be accompanied by size markers, and to retain visible background signal within the linear range (i.e. should not be saturated). The boundaries

of panels with low background have to be demarked with black lines. Splicing of panels should only be considered if unavoidable, and must be clearly marked on the figure, and noted in the legend with a statement on whether the samples were obtained and processed simultaneously. Quantitative comparisons between samples on different gels/blots are discouraged; if this is unavoidable, it should only be performed for samples derived from the same experiment with gels/blots were processed in parallel, which needs to be stated in the legend.

The total number of Supplementary Figures (not including the “unprocessed scans” Supplementary Figure) should not exceed the number of main display items (figures and/or tables (see our Guide to Authors and March 2012 editorial <http://www.nature.com/ncb/authors/submit/index.html#suppinfo>; <http://www.nature.com/ncb/journal/v14/n3/index.html#ed>). No restrictions apply to Supplementary Tables or Videos, but we advise authors to be selective in including supplemental data.

GUIDELINES FOR EXPERIMENTAL AND STATISTICAL REPORTING

REPORTING REQUIREMENTS – To improve the quality of methods and statistics reporting in our

papers we have recently revised the reporting checklist we introduced in 2013. We are now asking all life sciences authors to complete two items: an Editorial Policy Checklist (found here <https://www.nature.com/authors/policies/Policy.pdf>) that verifies compliance with all required editorial policies and a reporting summary (found here <https://www.nature.com/authors/policies/ReportingSummary.pdf>) that collects information on experimental design and reagents. These documents are available to referees to aid the evaluation of the manuscript. Please note that these forms are dynamic 'smart pdfs' and must therefore be downloaded and completed in Adobe Reader. We will then flatten them for ease of use by the reviewers. If you would like to reference the guidance text as you complete the template, please access these flattened versions at <http://www.nature.com/authors/policies/availability.html>.

Author Rebuttal to Initial comments
--

We would like to take the opportunity to thank each of the reviewers for their thoughtful comments and critiques. These have been helpful in the preparation of this revised manuscript, which includes several additional experiments and bioinformatic analyses. Most importantly, we have conducted functional studies on the role of CD32 for hemogenic endothelial cells (HECs) specification and maturation and we have found that CD32 activity is required for the robust generation of blood cells. The addition of these new data as well as of the required clarifications have been highlight in **blue font** in the revised manuscript.

Reviewer #1:

Remarks to the Author:

The Scarfo et al article reports the identification of specific markers for hemogenic endothelial cells (HECs) in human embryos and their potential use in generating hematopoietic cells in vitro. The authors performed transcriptomic analysis of human embryos and identified *FCGR2B*, which encodes the Fc receptor CD32, as highly enriched in the endothelial cell population containing HECs. Functional analysis confirmed that CD32+ endothelial cells have multilineage hematopoietic potential. The study also revealed that CD32+ HECs no longer require Notch signaling for hematopoietic commitment and display full commitment to hematopoiesis before the expression of hematopoietic markers. The findings of the ms can be significant if these provide a precise method for isolating HECs and advancing the generation of therapeutic hematopoietic cells from human pluripotent stem cells in vitro.

While the paper is done well technically, there are a significant number of concerns that need to be addressed to fully establish the utility of CD32 as a HEC marker.

1. It is uncertain whether CD32 is a critical receptor for the differentiation of hemogenic endothelium or whether this is just a consequence of maturation. To determine this, in-vitro differentiation of CD32+ cells using an anti-CD32 blocking antibody should be done or other functional experiments. Please perform sequential paneling for the endothelial to hematopoietic transition using CD43/CD45 acquisition.

For testing CD32 functionality during HEC cultures we have not used blocking antibodies as suggested given that most of the antibodies available are: 1) designed to block the non-specific labeling in antibody-based detection and cell separation experiments and it is not known whether they activate downstream CD32 signaling; 2) not specific for the isoform encoded by *FCGR2B*. As such, we turned to a genetic method. We inserted a cassette that expresses short hairpin RNAs against *FCGR2B* under the control of the constitutive chicken actin (CA) promoter into the AAVS1 “safe harbor” locus of H1 human embryonic stem cells (Extended Data Figure 6m)¹. This targeted integration results in the stable CD32 gene expression knock-down (KD) during the entire differentiation. Using these CD32 KD hESCs, we demonstrated that indeed CD32 function is required for the robust emergence of blood cells during hPSC differentiations. In fact, CD32 KD cells display a severely reduced capacity of generating CD45⁺ cells. These results are described in page 12 and are shown in figures 4g and Extended Data Fig 6p of the revised manuscript.

2. Although it is clear that CD32 is an important marker for maturation of endothelial cells towards a hemogenic fate, it is still unclear how much better this is than CD44. The data was convincing that transient culture followed by differentiation was better if the authors start from CD32+ cells; however, this may simply be an issue with timing since CD32 is acquired in culture of CD44 cells. To address this, the authors should perform sorting of the D8 CD34+/CD43-/DLL4-/CD32-/CD44+ cell fraction. In EHT conditions, perform flow cytometry for CD44/CD32 to see just how much of the culture acquires CD32 and how long this takes.

It is possible that the described short term EHT culture does not allow for sufficient in-vitro maturation of CD44⁺ cells.

We have investigated the kinetic of expression patterns of sorted CD44⁺CD32^{neg} endothelial cells upon HEC culture. Under these conditions a fraction of CD44⁺ cells begin to acquire CD32 expression after 48 hours of culture (Extended Data Figure 4d, page 8). However, as we show below, at end stage HEC culture not all the cells become CD45⁺ and indeed there is a residual CD44⁺ cell population that is CD32^{neg} that acquires an arterial fate and consequently DLL4 expression, in agreement with previous studies²⁻⁴.

Figure rebuttal 1: Representative flow cytometric analysis of end stage HEC cultures showing CD44, CD45, CD34, CD32 and DLL4 expression. Gated on SSC/FSC/Live.

Collectively, our data show that CD32 expression is restricted to a fraction of CD44⁺ cells. In particular, the clonal analysis described in figure 3k and 3l shows that nearly all CD32⁺ cells are committed to the hematopoietic fate and generate exclusively hematopoietic cells, while half of the CD44⁺ fraction give rise to a non-hematopoietic progeny. It must be noted that, already in the previous version of the manuscript, for this clonal HEC assay, we had extended to culture duration up to 14 days, exactly to allow enough time for the maturation of endothelial cells. We believe that altogether our results support the interpretation CD32 is a more specific HEC marker than CD44 as, within the CD44⁺ cell fraction, CD32 expression marks the subpopulation of cells committed to the blood lineage.

3. The characterization of the T cell potential of CD32⁺ cells is very limited. In the paper, T-cell populations are displayed as enriched from CD56⁻ populations. None of these upstream flow plots are provided. It would be nice to know how much of the cells in-vitro generate CD56⁺ NK cells as well. Additionally, longer term T-cell differentiation cultures to define what T-cell fractions are generated would be useful. Analysis of activation and mature markers like TCRab would be helpful.

The *in vitro* differentiation of human pluripotent stem cell (hPSC)-derived progenitors into the T-cell lymphoid lineage is associated with the development of CD56⁺ NK-lymphoid cells. To avoid confounding CD8⁺ NK-cells, we show the flow cytometric analysis of the CD4⁺CD8⁺ double positive T-lymphoid output, used as a measure of the T-lymphoid potential, performing a first gate on CD45⁺CD56^{neg} cells. We show below representative flow cytometric plots that illustrate the gating strategy as well as the expression of T-cell differentiation markers that we use to analyse the T-cell potential. Of note, most of the CD45⁺CD56^{neg} cells are also CD7⁺CD5⁺, further demonstrating their commitment to the T-lymphoid lineage. The presence of CD4⁺CD8⁺ double positive T-lymphoid cells is analyzed on CD45⁺CD56^{neg}CD7⁺CD5⁺ cells. In this revised version of the manuscript, we have amended the figure legend of Figure 3h, in which we had initially omitted the CD7⁺CD5⁺ gate.

Figure rebuttal 2: Representative flow cytometric analysis of CD45, CD54, CD7, CD5, CD4 and CD8 expression upon differentiation of CD32⁺ cells under T-lymphoid conditions. Left panel: gated on SSC/FSC/Live; middle panel: gated on SSC/FSC/LiveCD45⁺CD56^{neg}; right panel: gated on SSC/FSC/LiveCD45⁺CD56^{neg}CD7⁺CD5⁺.

Additionally, we have further characterized CD32⁺ cell-derived T-lymphoid output using artificial thymic organoid (ATO)⁵, which provides a more efficient *in vitro* model to study T-cell late-stage development and maturation. Using this strategy, we now describe (page 8, Extended Data Figure 3g) that CD32⁺ cells derive CD3⁺ T-cells that acquire the expression of the classical maturation and activation markers CD45RA, CD25 and CD27. In addition, we show in Extended Data Figure 3f that CD32⁺ cells can generate CD3⁺ T-cell lineages expressing either TCRαβ or TCRγδ (Extended Data Fig. 3f, page 8). Within the latter, some express the Vδ2 specific rearrangement that develops early in the human embryo and that is not observed when cord blood-derived CD34⁺ cells are used as controls (Extended Data Figure 3h, page 8)⁶.

Furthermore, in this revised version of the manuscript, we have included the analysis of the NK-lymphoid potential of WNTd CD32⁺ cells. As we now describe on page 8 and show in Extended Data Figure 3i (page 8), WNTd CD32⁺ cells can efficiently generate CD45⁺CD56⁺ cells belonging to the NK-cell lymphoid lineage.

We have also included new data regarding the hematopoietic differentiation of hPSC to the WNT independent (WNTi) extra-embryonic-like program that can be observed in the early yolk sac (YS). Also under these conditions, we observed that CD32 enriches for HECs. As NK-cells represent the defining lymphoid lineage of extra-embryonic hematopoiesis^{7,8}, we have characterized the NK-cell potential of CD32⁺ cells isolated from WNTi hPSC-derived hematopoietic cultures. As now described on page 13 and Extended Data Figure 8d, WNTi CD32⁺ cells display a robust NK-cell potential. As such, our results show that CD32⁺ HECs harbour the appropriate multilineage hematopoietic potential in accord with their ontogeny.

4. It is still unclear to me how the endothelium generated resembles molecularly *in vivo* AGM-like endothelium. It could be interesting to include a more in depth bioinformatic analysis using the Mikola lab's time course data instead of just the score card analysis and show how this compares from a trajectory standpoint. Furthermore, it could also be interesting to look at other marker genes for HSC-competent HE (for example CD27/Neur13) and see if this overlaps with CD32 expressing HE.

We further investigated the similarity of hPSC-derived CD32⁺ HECs with HECs identified bioinformatically in the human embryo using publicly available datasets^{9,10}. First, we checked whether *CD27* and *NEURL3*, two genes that track HSC-competent HECs in the mouse embryos¹¹⁻¹³, are conserved markers of HECs in the human embryo. As we show below, the

expression of *CD27* or *NEURL3* cannot be detected in the single cell transcriptome analysis of CS14-16 aorta-gonad-mesonephros region (AGM), recently published by Calvanese et al¹⁰, stages that are thought to include HSC-competent HECs⁷. This indicates that markers that have been identified to be specific HEC markers in the mouse embryo do not necessarily track in human development. This observation underscores an important issue that we will try to address in future studies, falling outside the scope of the current manuscript, which is why we are not presenting these data in this study.

Figure rebuttal 3: Feature plots showing *CD27* and *NEURL3* expression across clusters of the single-cell data set of CS14-CS16 AGM cells, published by Calvanese et al¹⁰. CS14-CS16 AGM are thought to comprise HSC-competent HECs⁷. This analysis shows that *CD27* and *NEURL3* transcripts cannot be detected in CS14-CS16 AGM cells.

We therefore opted for a bioinformatic assessment of similarity between HECs transcriptionally identified in the human embryo (including both YS and AGM) and hPSC-derived WNTd CD32⁺ cells. We and others have shown that the WNTd hPSC differentiation method yields AGM-like multipotent HECs^{4,14,15}. As expected, both AGM cells and hPSC-derived WNTd CD32⁺ cells express *HOXA9* and *HOXA10*, two defining genes of intra-embryonic hematopoietic development¹⁵, which are not detected in YS cells (Extended Data Figure 5 b-d, page 9). Thus, we can conclude that WNTd CD32⁺ cells resemble intra-embryonic-like rather than extra-embryonic cells. We next interrogated the degree of transcriptional similarity between hPSC-derived CD32⁺ cells and HECs from AGM regions isolated at different stages, spanning from CS10 to CS16. As we describe in the revised manuscript (page 9, Extended Data Figure 5e), the transcriptome of WNTd CD32⁺ cells harbors high similarity to a subfraction of transcriptionally defined HECs of AGM across the different developmental stages, including those when HSC-competent HECs are supposed to be found (CS13-16). Of note, hPSC-derived WNTd DLL4⁺ cells exhibit the greatest transcriptional similarity to CS14-16 embryonic arterial endothelium. This further supports our hypothesis that our WNTd hPSC differentiation strategy yields endothelial cells that are reminiscent of those found *in vivo* in the CS14-16 AGM.

Reviewer #2:

Remarks to the Author:

To capture rare and transient hemogenic endothelial cells (HECs) for characterizing endothelial-to-hematopoietic transition in human embryos, the authors performed functional and transcriptomic analyses in human primary tissues and human PSC hematopoietic cultures. They identify CD32 (encoded by FCGR2B), together with CD34 and ACE, as surface marker combinations to enrich HECs *in vivo* and *in vitro*. In the human dorsal aorta and yolk sac at 26-30 dpf, CD32 shows co-staining with classical HEC markers, such as Runx1, CD34, and ACE, in the endothelial cells; meanwhile, sorted CD32⁺ endothelial fraction shows higher hematopoietic outputs than the CD32^{neg} fraction after OP9-DL1 co-culture. In human PSC hematopoietic culture, CD32-enriched HEC fraction shows robust multilineage differentiation potential. Finally, the authors reveal that the acquisition of hematopoietic fate of CD32⁺ HECs is independent of Notch signaling.

Overall, this study provides some insights of the molecular characteristics during endothelial-to-hematopoietic transition (EHT) in human embryos and hPSC system *in vitro*. In this regard, however, EHT has been already transcriptionally characterized at single cell level in a large number of previous studies using human embryos and human PSC differentiation cultures.

Major concerns:

1. How does the author demonstrate that the CD32⁺ HEC is bona fide HEC (HSC primed) or EMP primed HEC? Whether the HECs labelled by CD32 in human embryos or human PSC hematopoietic cultures can generate transplantable HSCs after hematopoietic induction remains unknown.

To functionally prove that HECs yield hematopoietic stem cells (HSCs) in the human setting is a multifaceted problem, including, but not limited to, establishing the appropriate assay and permissive culture conditions to HSC specification from HECs.

In this regard:

1. As extensively reported by the Medvinsky group¹⁶, within the human embryo, at the very stage when HSCs are being generated from HECs, only cells that already express CD45 can be assayed in xenografts. Therefore human HECs, including the CD32⁺ cells that we describe, that by definition are CD45^{neg}, will fail to engraft. This is expected as transplant in a recipient is an assay for HSCs that is not suited for HECs.
2. Contrary to the mouse model, currently there are not culture conditions permissive to HSC specification from human HECs, including from human embryonic explant cultures. This is well documented by the Medvinsky group¹⁷, that shows how the conditions that work to specify *in vitro* HSCs from murine embryonic HECs cannot be successfully translated to human samples. Thus, until such conditions are identified (which falls outside the scope of this study), it cannot be ascertained which hemogenic endothelial population, CD32⁺ or CD32^{neg}, gives rise to HSCs. Similarly, protocols for the derivation of HSCs from hPSCs are not available yet. Therefore, testing the HSC-competency of hPSC-derived CD32⁺ cells is not feasible at the moment and goes beyond the scope of this study.
3. Experiments using embryo derived HECs (CD32⁺ and CD32^{neg}) require availability of human embryos at a very specific stage. It must be noted that, after the pandemic, in the institutions with which we collaborate many of the procedures that allowed the collection of human embryonic material have moved from inpatient to outpatient settings. As such, the number of samples available dropped dramatically to very few per year. In the 6-month timeframe of these revisions, we did not receive any properly staged embryos.

However, in this revised manuscript, we have characterised in-depth the transcriptional similarity of hPSC-derived WNTd CD32⁺ cells with *in vivo* HECs identified in YS and AGM at different embryonic stages. As expected, both AGM cells and hPSC-derived CD32⁺ cells

express *HOXA9* and *HOXA10*, two defining genes of intra-embryonic hematopoietic development¹⁵, which are not detected in YS cells (Extended Data Figure 5 b-d, page 9). These results are in accord with previous studies showing that the WNTd hematopoietic differentiation of hPSCs yields cells resembling AGM cells⁷. Thus, we can conclude that WNTd CD32⁺ cells resemble intra-embryonic rather than extra-embryonic cells. We next interrogated the degree of transcriptional similarity between hPSC-derived CD32⁺ cells and HECs from AGM regions isolated at different stages, spanning from CS10 to CS16, using available datasets^{9,10}. As we describe in the revised manuscript (page 9, Extended Data Figure 5e), the transcriptome of CD32⁺ cells harbors high similarity to a subfraction of transcriptionally defined HECs of AGM across the different developmental stages, including those when HSC-competent HECs are supposed to be found (CS13-16). Of note, hPSC-derived WNTd DLL4⁺ cells exhibit the greatest transcriptional similarity to CS14-16 embryonic arterial endothelium. This further supports our hypothesis that our hPSC differentiation strategy yields endothelial cells that are reminiscent of those found *in vivo* in the CS14-16 AGM.

As EMPs in the mouse embryo are generated in the YS, in this revised manuscript we have added the characterization of WNTi hPSC-derived hematopoietic cultures that recapitulate early YS hematopoiesis and that yield HECs, whose gene expression and potential are similar to the EMP hematopoietic program^{7,14}. As shown in Extended Data Figure 8c (page 13), we describe that CD32 enriches for HECs also under these conditions.

Collectively, these results suggest that the expression of CD32 marks a unique HEC state common to HECs of all ontogenies, including HSC-competent HECs. However, we have also commented in the discussion (page 15) that we could not formally test whether HSCs are generated through a CD32⁺ HEC intermediate.

2. At 23 dpf, the authors show that ACE expression is observed in the mesenchyme surrounding the DA; however, at 27 dpf, the ACE expression in the sub-aortic mesenchyme cannot be observed and is observed in the endothelial cells lining the ventral wall of DA (Fig. 1a). How to explain the changes of location or fate of the ACE+ mesenchyme from 23 dpf to 27 dpf? What is the relationship between ACE+ mesenchyme and ACE+ endothelium?

We have previously described how the localization of ACE⁺ cells change during development in the human embryo¹⁸. As shown in Figure 1a, at 23 days of development, cells expressing ACE are concentrated in the dorsal mesenteric mesenchyme, which surrounds the dorsal aorta, and are always localized in the caudal portion of the embryo. Starting from 26 days, ACE expression can be detected in the endothelial cells lining the ventral side of the dorsal aorta.

While the panels in Figure 1 display the absence of ACE expression in the subaortic mesenchyme precursors at 27 days of development, other sections and other embryos at the same stage of development show ACE expression both in mesenchyme and in the ventral endothelium of the DA, as shown here in the Figure rebuttal 4. As such, we have amended the sentence describing these data on page 4.

Figure rebuttal 4 (next page): ACE expression in a transversal section of a 27-day human embryo. ACE marks rare cells scattered in the mesenchyme surrounding the dorsal aorta (DA) (white arrowhead) and some endothelial cells in the ventral side of DA (black arrowheads).

We have shown previously that intra-embryonic hematopoiesis in the human embryo emerges from precursors that can be found in the human splanchnopleura as early as 19 days of development⁸. We have also shown that hematopoietic potential in the 24–26-day-old human embryo is completely restricted to ACE⁺ cells isolated from the splanchnopleura¹⁸. Notably, vascular endothelial cells sorted from the same territory at the same time are not yet hematopoietic¹⁸. This suggests a colonization of the ventral wall of the aorta by ACE⁺ precursors of the HEC lineage migrating through the subaortic mesenchyme. However, formal proof of this hypothesis is lacking. We have added a sentence on the possible relationship between ACE⁺ mesenchyme and ACE⁺ endothelium on page 4.

3. In the human DA, CD32 and RUNX1 show co-localization in the HECs; however, CD32 and RUNX1C-EGFP show no co-staining in the day 8 CD34⁺CD43^{neg}CD73^{neg}CD184^{neg}DLL4^{neg} cells (containing HECs) in human PSC hematopoietic cultures. How to explain this difference?

In vertebrates Runx1, the pivotal regulator of developmental hematopoiesis, is transcribed from two promoters, the distal P1 and proximal P2, which yield different Runx1 isoforms^{19,20}. While the antibody used in Figure 2c binds to all the different human RUNX1 isoforms, the hPSC reporter line used for Figures 3c, 3d and 3f is specific for the *RUNX1C* isoform controlled by the distal P1 promoter, which is activated only as HECs initiate the EHT and become hematopoietic progenitors^{4,15}. This is clarified on page 7.

4. Given that 6 HEC clusters (annotated in single-cell transcriptome atlas) form a

developmental continuum (Monocle3-based trajectory analysis, Fig. 4b) featured by arterial identity loss and hematopoietic identity gain, it remains controversial whether these HEC clusters are considered as 6 heterogeneous subtypes or 6 developmental stages.

In the first version of this manuscript, we have used Monocle3 to delineate the developmental progression of cells within clusters 0, 1, 2, 11, 16, and 17 (as shown in Figure 4b as well as Extended Data Figure 6d in this revised version). Monocle3 revealed a predominant, uninterrupted trajectory stretching from cluster 0 to cluster 17, accompanied by only two small branching paths whose terminal node is not a cluster marked by the expression of hematopoietic genes, as illustrated in the UMAP in Figure 4b and Extended Data Figure 6d and e. Furthermore, Monocle3 facilitated the estimation of the pseudotime, a metric indicating the extent of a cell's advancement through a developmental process. Tracking the pseudotime values against progression along this main trajectory, we mapped individual cells' pseudotime scores onto the UMAP (Extended Data Figure 6d). This highlighted a consistent incremental trend in pseudotime scores, which were minimal in cluster 0 and increased steadily towards cluster 17, demonstrating a uniform gradient without discrete divisions. The absence of both multiple branching trajectories and distinct pseudotime gradients towards cells expressing hematopoietic genes indicates that, under these conditions, these six clusters likely represent sequential developmental stages rather than distinct heterogeneous subtypes.

To further explore the potential lineage connections and developmental pathways, we conducted trajectory inference analysis utilizing additional tools. Specifically, we consulted Dynverse, a resource that evaluates over 50 different trajectory inference methods, to determine the most suitable approach for pseudotime analysis tailored to our experimental conditions. Dynverse recommended the partition-based graph abstraction (PAGA)-tree method, which we then applied to corroborate the findings obtained from Monocle3. The PAGA-tree method constructs an inherently unbiased pseudotemporal trajectory by analyzing the transcriptional resemblances among cells. As required by the PAGA-tree method, we purposefully designated the cells of cluster 0 as the starting point of this trajectory. Consistent with Monocle3's insights, our analysis via PAGA-tree identified a principal developmental pathway extending from cluster 0 to cluster 17, characterized by only minor branches (Extended Data Figure 6f, g, page 10). This pathway exhibited a singular pseudotime gradient with scores incrementing from the lowest, at cluster 0, to the highest, at cluster 17. These observations align with the Monocle3-derived conclusion, further supporting our interpretation that the six clusters represent distinct developmental stages.

Additionally, we have also employed CellOracle, given its capacity to perform pseudotime analysis. Indeed, although CellOracle is primarily designed for the analysis of gene regulatory networks and the prediction of regulatory gene functions in single cells, its extension into pseudotime analysis provided a means to validate our findings from Monocle3 and PAGA-tree. The pseudotime patterns identified by CellOracle aligned with those from the other methods (Extended Data Fig 6h, page 10), adding a layer of confirmation to our initial results. The consistency of the achieved results enhanced our confidence in the developmental trajectory we have proposed. The agreement between different analytical approaches highlights a principal trajectory emerging from cluster 0 and advancing towards cluster 17, depicting a unified developmental sequence with minimal deviations. Overall, this congruence supports our interpretation of the 6 clusters as progressive stages of HEC development.

5. Is CD32 a conserved marker for enrichment of HECs in mammals, including mice?

Whether CD32 and in general Fc receptors expression and activity are conserved markers of HECs in mammals, or even vertebrates, is a very interesting question. However, this falls beyond the scope of our current studies.

Reviewer #3:

Remarks to the Author:

In this manuscript, Scarfò et al. identify CD32 as a marker of HEC and use it to purify and molecularly profile HECs in human embryos and hPSCs. The authors isolate and profile CD34+ACE+ HECs from human embryos, showing that CD32 is co-expressed on a subset of HECs. CD32+ HECs are committed to hematopoietic fate and undergo EHT in a Notch-independent fashion. This is a well-developed and technically excellent study that impressively uses human embryos to uncover a specific marker of HECs. Despite molecular characterization of CD32+ HECs, the biological insights into how hematopoietic cell fate is specified or approaches for improving hematopoietic output are somewhat limited. Specific comments below.

1. The authors perform single cell RNAseq analysis of CD32+ HECs from hPSCs. However, it remains unclear what are the mechanisms and pathways driving hematopoietic fate specification, which can be harnessed to regulate EHT and/or hematopoietic output from hPSCs. The finding that the CD32+ HEC is Notch-independent is interesting, though it essentially confirms published data showing that Notch activity and dependence progressively declines during EHT (e.g. Lizama et al. 2015; Richard et al. 2013).

We agree that others have shown that the process of emergence of blood cells become gradually Notch-independent in its final steps. However, with this study we added new resolution to this rapid and transient process and identified the demarcation of the Notch-dependent and Notch-independent stages, which is characterized by CD32 expression. Notably, we have provided evidence for the first time that the emergence of blood cells becomes a Notch-independent process even before the actual morphological remodeling and the expression of hematopoietic genes can be observed, i.e. earlier than previously thought.

Since CD32 marks HECs that are fully committed to the blood fate, we exploited its expression to identify pathways driving HEC specification and differentiation, whose modulation can increase the hematopoietic output from hPSC cultures. By contrasting the transcriptomic profiles of WNTd CD32⁺ (HECs) and DLL4⁺ (arterial-like) hPSC-derived populations using ORA, we observed that CD32⁺ cells are positively associated with BMP signaling GO terms (Extended Data Figure 7a, Supplementary Table 3j, page 12). This indicates that CD32⁺ HECs experience a higher level of BMP signaling activation compared to hPSC-derived arterial cells. We therefore hypothesized that the development of CD32⁺ HECs may require active BMP signaling. To test this, we added either recombinant BMP4 or the BMP signaling chemical inhibitor LDN-193189 to WNTd hPSC-derived differentiations during the VEGF-driven HEC specification stage. Critically, while the addition of BMP4 resulted in around a 2-fold increase of the proportion of CD32⁺ cells, BMP signaling inhibition severely impaired their formation (Extended Data Figure 7b and 7c, page 12). If BMP signaling positively regulates HEC specification, as suggested by the observed increase of the CD32⁺ subpopulation, BMP treatment should result in a higher proportion of HECs present in the day 8 total CD144⁺ endothelial population, which would be expected to generate more hematopoietic progenitors. Indeed, CD144⁺ cells isolated from BMP-treated cultures gave rise to 2-fold more CD45⁺ cells after HEC culture. On the other hand, BMP signaling inhibition almost abrogated the hematopoietic potential of day 8 CD144⁺ cells (Extended Data Fig 7d and 7e, page 12).

We next leveraged the hPSC-derived HEC scRNAseq dataset we generated to identify pathways regulating *RUNX1*⁺ HEC differentiation. We focused on the progression to *FCGR2B*⁺ cluster (i.e., from clusters 0, 1, 2 to cluster 11), given that CD32 marks a unique HEC stage. ORA revealed that this progression is associated with a downregulation of Rho/ROCK signaling (Extended Data Fig. 7f, Supplementary Table 3k, page 12). We then tested whether the addition the chemical ROCK inhibitor Y-27632 during the HEC culture could increase the hematopoietic output from hPSCs. Indeed, ROCK signaling inhibition increased the frequency clonogenic progenitors generated during HEC culture (Extended

Data Fig. 7g and 7h, page 13), likely by synchronizing or facilitating the progression of HEC towards blood fate.

Collectively, these results indicate that HEC specification and differentiation require stage-specific BMP signaling activation and Rho/ROCK signaling inhibition, respectively, and highlight the value of using CD32 to study HEC biology.

2. CD32 marks only the penultimate stage of EHT. This raises the question of what are the earlier stages of HEC ontogeny and can more specific markers of those stages be identified.

In Supplementary Table 3a we provide the complete list of the markers we have identified for each individual cluster. While some surface proteins are present in this list, unfortunately none of the markers listed display a restricted expression to a specific cluster as CD32 does for cluster 11. As such, our bioinformatic analysis was unable to reveal specific markers for the other five transcriptional subpopulations of HECs.

3. Is CD32⁺ an obligate stage of HEC ontogeny? In Fig. S4e, HEC precursors at a specific stage do not appear to uniformly express CD32, raising the possibility that only some HECs transition through a CD32⁺ stage during EHT.

We agree with the reviewer that *FCGR2B* expression is not uniformly observed in cells belonging to cluster 11, as depicted in the feature plot displayed in Extended Data Figure 4e (Extended Data Figure 6c in the current revised version). This variation could be attributed to dropout events, which are frequently encountered in single-cell analyses. Such events may arise from the inherently low levels of mRNA within individual cells or from suboptimal mRNA capture during the library preparation phase. Furthermore, the inherent stochasticity of mRNA expression can also lead to the diversity in expression patterns. These factors highlight the complexities involved in ensuring consistent gene expression detection in single-cell studies.

In the first version of this manuscript, we have used Monocle3 to delineate the developmental progression of cells within clusters 0, 1, 2, 11, 16, and 17 (as shown in Figure 4b as well as Extended Data Figure 6d). Monocle3 revealed a predominant, uninterrupted trajectory stretching from cluster 0 to cluster 17, accompanied by only two small branching paths whose terminal node is not a cluster marked by the expression of hematopoietic genes, as illustrated in the UMAP in Figure 4b and Extended Data Figure 6d and e. Furthermore, Monocle3 facilitated the estimation of the pseudotime, a metric indicating the extent of a cell's advancement through a developmental process. Tracking the pseudotime values against progression along this main trajectory, we mapped individual cells' pseudotime scores onto the UMAP (Extended Data Figure 6d). This highlighted a consistent incremental trend in pseudotime scores, which were minimal in cluster 0 and increased steadily towards cluster 17, demonstrating a uniform gradient without discrete divisions. The absence of both multiple branching trajectories and distinct pseudotime gradients towards cells expressing hematopoietic genes indicates that under these conditions HECs give rise to blood cells via a CD32⁺ intermediate.

To further explore the potential lineage connections and developmental pathways, we conducted trajectory inference analysis utilizing additional tools. Specifically, we consulted Dynverse, a resource that evaluates over 50 different trajectory inference methods, to determine the most suitable approach for pseudotime analysis tailored to our experimental conditions. Dynverse recommended the partition-based graph abstraction (PAGA)-tree method, which we then applied to corroborate the findings obtained from Monocle3. The PAGA-tree method constructs an inherently unbiased pseudotemporal trajectory by analyzing the transcriptional resemblances among cells. As required by the PAGA-tree method, we purposefully designated the cells of cluster 0 as the starting point of this trajectory. Consistent with Monocle3's insights, our analysis via PAGA-tree identified a principal developmental pathway extending from cluster 0 to cluster 17, characterized by only minor branches

(Extended Data Figure 6f and 6g, page 10). This pathway exhibited a singular pseudotime gradient with scores incrementing from the lowest at cluster 0 to the highest at cluster 17. These observations align with the Monocle3-derived conclusion, further reinforcing the notion that HECs under these conditions transit through a CD32⁺ stage.

Additionally, we have also employed CellOracle, given its capacity to perform pseudotime analysis. Indeed, although CellOracle is primarily designed for the analysis of gene regulatory networks and the prediction of regulatory gene functions in single cells, its extension into pseudotime analysis provided a means to validate our findings from Monocle3 and PAGA-tree. The pseudotime patterns identified by CellOracle aligned with those from the other methods (Extended Data Fig. 6h, page 10), adding a layer of confirmation to our initial results. The consistency of the achieved results enhanced our confidence in the developmental trajectory we have proposed. The agreement between different analytical approaches highlights a principal trajectory emerging from cluster 0 and advancing towards cluster 17, depicting a unified developmental sequence with minimal deviations. This congruence supports the hypothesis that the process of blood emergence is characterized by a CD32⁺ stage.

To further elucidate the role of CD32 in the ontogeny of HECs, we undertook an in-silico perturbation analysis of CD32 across the six identified HEC clusters. This was facilitated by CellOracle²¹, a computational tool designed for constructing cluster-specific gene regulatory networks (GRNs) through the integration of scRNAseq expression profiles and pseudotime analyses. Using the CellTalkDB dataset²², we established a foundational GRN and integrated CD32 and its interacting partners, according to the data available in the STRING database. By simulating a knockout (KO) of CD32 and observing the resultant effects on both direct and indirect gene targets within the CellOracle-inferred GRN, we identified a shift in the developmental trajectory of HEC differentiation, particularly noting a reversion in differentiation at cluster 11 (Extended Data Fig 6j and 6k, pages 11-12). These effects were quantitatively assessed through perturbation scores (PS), which measure the direction and magnitude of changes induced by the perturbation in comparison to the natural differentiation process (that is, computing the inner product between the two vectors). Visualization of these effects as a vector field over a digitized grid (Extended Data Fig. 6l, pages 11-12) revealed a prevalent distribution of negative PS values within clusters 1, 2, 11, and 16. This pattern suggests that a perturbation of CD32 is likely to affect the differentiation process at these critical stages of HEC development.

Based on these data, we tested whether CD32 plays a functional role in HEC specification or maturation using a genetic model. We reasoned that if the CD32⁺ stage is an obligated step for HEC maturation, the genetic loss of CD32 should have consequences in the generation of blood cells. We inserted a cassette that expresses short hairpin RNAs against *FCGR2B* under the control of the constitutive chicken actin (CA) promoter¹ in the AAVS1 “safe harbor” locus of H1 human embryonic stem cells (Extended Data Figure 6m). This targeted integration of the cassette results in the stable CD32 gene expression knock-down (KD) during the entire differentiation. Using these CD32 KD hESCs, we demonstrate that indeed CD32 function is required for the emergence of blood cells during hPSC differentiations. In fact, CD32 KD cells display a severely reduced capacity of generating CD45⁺ cells. These results are described on page 12 and shown in figures 4g and Extended Data Fig. 4p of the revised manuscript.

Collectively our studies suggest that CD32 is required for the robust generation of blood cells and HECs transit through a CD32⁺ stage in order to generate blood. However, we do not know whether the residual hematopoietic output in CD32 KD cells is due to CD32-independent hematopoiesis or if it is the result of an incomplete repression of CD32 expression. As such, while we demonstrate that CD32 plays a role in the emergence of blood cells, we cannot formally rule out that HECs can generate hematopoietic cells independently of CD32 signaling and we have discussed this aspect on page 15.

4. In human embryo experiments, it remains unclear if CD32 marks HECs that gives rise to

HSCs. CFC studies in Fig. 2f show that CD32⁺ cells are more hemogenic compared to CD32⁻, however it remains possible that HSCs arise from a distinct HEC population which may be CD32 negative.

To functionally prove that HECs yield hematopoietic stem cells (HSCs) in the human setting is a multifaceted problem, including, but not limited to, establishing the appropriate assay and having culture conditions that are permissive to HSC specification from HECs.

In this regard:

1. As extensively reported by the Medvinsky group¹⁶, within the human embryo, at the very stage when HSC are being generated from HECs, only cells that already express CD45 can be assayed in xenografts. Therefore human HECs, including the CD32⁺ cells that we describe, that by definition are CD45^{neg}, will fail to engraft. This is expected as transplant in a recipient is an assay for HSCs that is not suited for HECs.
2. Contrary to the mouse model, currently there are not culture conditions permissive to HSC specification from human HECs, including from human embryonic explant cultures. This is well documented by the Medvinsky group¹⁷, that shows how the conditions that work to specify *in vitro* HSCs from murine embryonic HECs cannot be successfully translated to human samples. Thus, until such conditions are identified (which falls outside the scope of this study), it cannot be ascertained which hemogenic endothelial population, CD32⁺ or CD32^{neg}, gives rise to HSCs. Similarly, protocols for the derivation of HSCs from hPSCs are not available yet. Therefore, testing the HSC-competency of hPSC-derived CD32⁺ cells is not feasible at the moment and goes beyond the scope of this study.
3. Experiments using embryo derived HECs (CD32⁺ and CD32^{neg}) require availability of human embryos at a very specific stage. It must be noted that, after the pandemic, in the institutions with which we collaborate many of the procedures that allowed the collection of human embryonic material have moved from inpatient to outpatient settings. As such, the number of samples available dropped dramatically to very few per year. In the 6-month timeframe of these revisions, we did not receive any properly staged embryos.

However, in this revised manuscript, we have characterised in depth the transcriptional similarity of hPSC-derived WNTd CD32⁺ cells with *in vivo* HECs identified in YS and AGM at different embryonic stages. We and others have shown that the WNTd hPSC differentiation method yields AGM-like multipotent HECs^{4,14,15}. As expected, both AGM cells and hPSC-derived CD32⁺ cells express *HOXA9* and *HOXA10*, two defining genes of intra-embryonic hematopoietic development¹⁵, which are not detected in YS cells (Extended Data Fig 5b-d, page 9). Thus, we can conclude that CD32⁺ cells resemble intra-embryonic-like rather than extra-embryonic cells. We next interrogated the degree of transcriptional similarity between hPSC-derived CD32⁺ cells and HECs from AGM regions isolated at different stages, spanning from CS10 to CS16. As we describe in the revised manuscript (page 9, Extended Data Figure 5e), the transcriptome of CD32⁺ cells highly similar to a subfraction of transcriptionally defined HECs of AGM across the different developmental stages, including those when HSC-competent HECs are supposed to be found (CS13-16). Of note, hPSC-derived DLL4⁺ cells exhibit the greatest transcriptional similarity to CS14-16 embryonic arterial endothelium. This further supports our hypothesis that our hPSC differentiation strategy yields endothelial cells that are reminiscent of those found *in vivo* in the CS14-16 AGM.

In addition, we now provide new data highlighting that CD32 marks a unique stage of HECs across different hematopoietic programs. In fact, we have also characterized CD32⁺ HECs isolated from WNTi hPSC-derived hematopoietic cultures that recapitulate early YS hematopoiesis¹⁴. As shown in Extended Data Figure 8c (page 13), CD32 enriches for HECs also under these conditions.

Collectively, these results suggest that the expression of CD32 marks a unique HEC state common to HECs of all ontogenies, including HSC-competent HECs. However, we have

also commented in the discussion (page 15) that we could not formally test whether HSCs are generated through a CD32⁺ HEC intermediate.

Minor comments

1. Fig. 2e shows that nearly ~80% of CD34⁺ cells in the YS are also CD32⁺ (vs. 5% in AGM and PSCs). What is a possible reason for this apparent abundance of CD32⁺ in YS? Are these CD32⁺ cells also hemogenic?

This is an interesting question that would require the use of additional human embryos for more refined analysis of the expression of other markers in YS CD32⁺ cells (including, and not limited to, RUNX1, endothelial vs mesenchymal as well as arterial vs venous markers). In addition, to determine the frequency of HECs among the high proportion of CD32⁺ cells found in the YS, *ex vivo* functional experiments at limiting dilution should be performed. Unfortunately, during the 6-month timeframe of these revisions, we did not receive any embryo of the proper stage. Therefore, we can only speculate a response on this point. As the YS at this stage is a particularly active hematopoietic site, it is possible although not likely, that HECs might represent the predominant proportion of the total endothelial cell fraction. Alternatively, CD32 in the YS might have a broader endothelial expression than in dorsal aorta, with HECs confined to the CD73^{neg}CD184^{neg} fraction, similarly to what we have now described for WNT independent (WNTi) extra-embryonic like hPSC-derived CD32⁺ cells (Extended Data Figure 8, page 13).

2. Fig. 3c shows that CD32⁺ cells are RUNX1Cneg, however single cell analysis in Fig. 4 shows that they are RUNX1+?

In vertebrates Runx1, the pivotal regulator of developmental hematopoiesis, is transcribed from 2 promoters, the distal P1 and proximal P2, which yield different Runx1 isoforms^{19,20}. While the antibody used for figure 2c binds to all the different human RUNX1 isoforms, the hESC reporter line used for figures 3c, 3d and 3f is specific for the RUNX1C isoform controlled by the distal P1 promoter, which is activated as HECs initiate the EHT and become hematopoietic progenitors^{4,15}. We describe this on page 7. Since RUNX1 expression is detected at the transcriptomic level in CD32⁺ cells, very likely they are positive for the RUNX1 isoforms under the control of the proximal P2 promoter. Alternatively, in these cells RUNX1 expression is controlled post-transcriptionally.

3. Fig. 3e images demonstrating that CD32⁺ cells are bona fide EC are important and could be enlarged to better show classical endothelial morphology.

In this revised version of the manuscript, we have now replaced the top panel of figure 3e with a picture taken with higher magnification.

References Rebuttal

1. Tiyaboonchai, A. *et al.* Utilization of the AAVS1 safe harbor locus for hematopoietic specific transgene expression and gene knockdown in human ES cells. *Stem Cell Res.* **12**, 630–637 (2014).

2. Robert-Moreno, À. *et al.* Impaired embryonic haematopoiesis yet normal arterial development in the absence of the Notch ligand Jagged1. *Embo J* **27**, 1886–1895 (2008).
3. Yamamizu, K. *et al.* Convergence of Notch and β -catenin signaling induces arterial fate in vascular progenitors. *The Journal of Cell Biology* **189**, 325–338 (2010).
4. Ditadi, A. *et al.* Human definitive haemogenic endothelium and arterial vascular endothelium represent distinct lineages. *Nat Cell Biol* **17**, 580–591 (2015).
5. Montel-Hagen, A. *et al.* Organoid-Induced Differentiation of Conventional T Cells from Human Pluripotent Stem Cells. *Cell Stem Cell* (2019) doi:10.1016/j.stem.2018.12.011.
6. Atkins, M. H. *et al.* Modeling human yolk sac hematopoiesis with pluripotent stem cells. *J Exp Med* **219**, e20211924 (2021).
7. Dege, C. *et al.* Potently Cytotoxic Natural Killer Cells Initially Emerge from Erythro-Myeloid Progenitors during Mammalian Development. *Dev Cell* **53**, 229-239.e7 (2020).
8. Tavian, M., Robin, C., Coulombel, L. & Péault, B. The human embryo, but not its yolk sac, generates lympho-myeloid stem cells: mapping multipotent hematopoietic cell fate in intraembryonic mesoderm. *Immunity* **15**, 487–95 (2001).
9. Zeng, Y. *et al.* Tracing the first hematopoietic stem cell generation in human embryo by single-cell RNA sequencing. *Cell Res* **29**, 881–894 (2019).
10. Calvanese, V. *et al.* Mapping human haematopoietic stem cells from haemogenic endothelium to birth. *Nature* 1–7 (2022) doi:10.1038/s41586-022-04571-x.
11. Ning, X. *et al.* Divergent expression of *Neur13* from hemogenic endothelial cells to hematopoietic stem progenitor cells during development. *J Genet Genomics* (2023) doi:10.1016/j.jgg.2023.05.006.
12. Hou, S. *et al.* Embryonic endothelial evolution towards first hematopoietic stem cells revealed by single-cell transcriptomic and functional analyses. *Cell Res* **30**, 376–392 (2020).
13. Li, Y., Gao, L., Hadland, B., Tan, K. & Speck, N. A. CD27 marks murine embryonic hematopoietic stem cells and type II prehematopoietic stem cells. *Blood* **130**, 372–376 (2017).
14. Sturgeon, C. M., Ditadi, A., Awong, G., Kennedy, M. & Keller, G. Wnt signaling controls the specification of definitive and primitive hematopoiesis from human pluripotent stem cells. *Nat Biotechnol* **32**, 554–561 (2014).
15. Ng, E. S. *et al.* Differentiation of human embryonic stem cells to HOXA⁺ hemogenic vasculature that resembles the aorta-gonad-mesonephros. *Nat Biotechnol* **34**, 1168–1179 (2016).
16. Ivanovs, A., Rybtsov, S., Anderson, R. A., Turner, M. L. & Medvinsky, A. Identification of the Niche and Phenotype of the First Human Hematopoietic Stem Cells. *Stem Cell Reports* **2**, 449–456 (2014).
17. Easterbrook, J. *et al.* Analysis of the Spatiotemporal Development of Hematopoietic Stem and Progenitor Cells in the Early Human Embryo. *Stem Cell Reports* (2019) doi:10.1016/j.stemcr.2019.03.003.

18. Sinka, L., Biasch, K., Khazaal, I., Péault, B. & Tavian, M. Angiotensin-converting enzyme (CD143) specifies emerging lympho-hematopoietic progenitors in the human embryo. *Blood* **119**, 3712–3723 (2012).
19. Bee, T. *et al.* Alternative Runx1 promoter usage in mouse developmental hematopoiesis. *Blood Cells, Molecules, and Diseases* **43**, 35–42 (2009).
20. Ghozi, M. C., Bernstein, Y., Negreanu, V., Levanon, D. & Groner, Y. Expression of the human acute myeloid leukemia gene AML1 is regulated by two promoter regions. *Proc. Natl. Acad. Sci.* **93**, 1935–1940 (1996).
21. Kamimoto, K. *et al.* Dissecting cell identity via network inference and in silico gene perturbation. *Nature* 1–10 (2023) doi:10.1038/s41586-022-05688-9.
22. Shao, X. *et al.* CellTalkDB: a manually curated database of ligand–receptor interactions in humans and mice. *Brief. Bioinform.* **22**, (2020).

Decision Letter, first revision:

Our ref: NCB-A50996A

26th January 2024

Dear Andrea,

Thank you for submitting your revised manuscript "CD32 allows capturing blood cells emergence in slow motion during human embryonic development" (NCB-A50996A). It has now been seen by the original referees and their comments are below. As you will see by going through the comments, reviewer #2 continued to raise some concerns and we therefore discussed these concerns with reviewer #3, hence the delay in getting back to you, which I sincerely apologize for. You can find the view of reviewer #3 on the remaining concerns raised by reviewer #2 in the section of their report "ADDITIONAL COMMENTS TO REMAINING ISSUES RAISED BY REVIEWER #2". Taking everything into account, the paper has improved in revision, and therefore we'll be happy in principle to publish it in Nature Cell Biology, pending minor revisions to satisfy the referees' final requests and to comply with our editorial and formatting guidelines.

If the current version of your manuscript is in a PDF format, please email us a copy of the file in an editable format (Microsoft Word or LaTeX)-- we cannot proceed with PDFs at this stage.

We are now performing detailed checks on your paper and will send you a checklist detailing our editorial and formatting requirements in about a week. Please do not upload the final materials and make any revisions until you receive this additional information from us. When you do receive this checklist, apart from the issues related to journal policies and style, we will expect you to also provide an additional point-by-point response to the remaining reviewer comments and:

- address the remaining points by referee #1
- address the minor comments by referee #2
- address the first part of point #2 by referee #2: "given that the staining pattern at 27 dpf in the Figure 1a is not representative, we suggest that the authors can revise it with a new panel".
- **textually** address the second part of point #2 by referee #2 ("Although the authors have explained that the ACE+ cells in the subaortic mesenchyme are likely precursors of aortic ACE+ HECs, whether the mesenchymal ACE+ cells can also regulate the hematopoietic fate acquisition of aortic ACE+ cells is still questionable"), as suggested by referee #3: "On the other hand, while I also agree that whether the mesenchymal ACE+ cells can also regulate the hematopoietic fate acquisition of aortic ACE+ cells remains unclear, the questions sounds somewhat peripheral and could be addressed in the text".
- **textually** address point #1 by referee #2 ("Referring to a previous study by Daley's lab (Sugimura et al. Nature. 2017. PMID: 28514439), the injection of human PSC-derived transgenic HECs into NSG mice enables to evaluate multi-lineage hematopoietic chimerism. We suggest that the authors can apply the transplantation assay to demonstrate whether application of CD32 can capture HSPC-competent HECs"), acknowledging the difficulty of these experiments and that further in vivo functional evidence are to be provided in future studies.
- **textually** address point #6 by referee #2 ("In addition, the newly added in silico perturbation and

functional validation confirmed the role of CD32 in whole HEC subclusters. However, this contradicts the observation that the expression of FCGR2B was specifically higher in HEC subcluster 11 but not uniformly expressed in all HEC cells. In response to Question 3 from Reviewer #3, the author suggests that the claim about a uniform expression of FCGR2B not being observed in cluster 11 could be due to drop out events commonly encountered in single-cell analyses. This claim maybe not correct. To address this point, they should consider data imputation or utilize other sequencing technologies, such as Smart-seq2, to directly address this question and also avoid misleading"), as suggested by referee #3: "The issue of whether or not CD32+ late HE is an obligatory stage in hematopoietic ontogeny should be raised in the discussion".

- On the contrary, as also indicated in the previous round of review, while we find insightful the comment by referee #2 regarding potential conservation (point #5), we consider it beyond the scope of this study and addressing it will not be necessary.

Thank you again for your interest in Nature Cell Biology. Please do not hesitate to contact me if you have any questions.

Best wishes,
Stelios

Stylianos Lefkopoulos, PhD
He/him/his
Senior Editor, Nature Cell Biology
Springer Nature
Heidelberger Platz 3, 14197 Berlin, Germany

E-mail: stylianos.lefkopoulos@springernature.com
Twitter: @s_lefkopoulos
LinkedIn: [linkedin.com/in/stylianos-lefkopoulos-81b007a0](https://www.linkedin.com/in/stylianos-lefkopoulos-81b007a0)

Reviewer #1 (Remarks to the Author):

I wanted to give credit to the authors for experimentally addressing all of my comments. The manuscript is a lot more complete now, and the new data add robustness to the manuscript's main findings. I think the ms will be of significant impact to the human hematopoietic community.

I just had a couple of comments that could be addressed simply by changes in the text:

1) I am still somewhat confused about what Cd32 is marking. The authors describe in the text that 'CD32 expression in day 8 WNTd CD34+CD43negCD73negCD184negDLL4neg cells demarcates endothelial cells that do not express RUNX1C-EGFP'. This would suggest that in this context CD32 precedes RUNx1 expression. However, in the text referring to the other figures and in their model in figure 4 the authors clearly show that CD32 is expressed in a committed HEC (which express RUNx1). Which one is it? I think the authors need to be very careful with they wording here.

2)The ms would benefit from having a better description of the CD32KD cell line as far as the different populations that are altered in their differentiation protocols. Is the KD simply affection CD45 expression or is EC differentiation affected as well?

Reviewer #2 (Remarks to the Author):

Major concerns:

1. For the major concern 1, the authors explain that xenograft assay is not suited for HECs and that the culture condition for HSC specification from HECs is not established in vitro. Referring to a previous study by Daley's lab (Sugimura et al. Nature. 2017. PMID: 28514439), the injection of human PSC-derived transgenic HECs into NSG mice enables to evaluate multi-lineage hematopoietic chimerism. We suggest that the authors can apply the transplantation assay to demonstrate whether application of CD32 can capture HSPC-competent HECs.

2. For the major concern 2, given that the staining pattern at 27 dpf in the Figure 1a is not representative, we suggest that the authors can revise it with a new panel. Although the authors have explained that the ACE+ cells in the subaortic mesenchyme are likely precursors of aortic ACE+ HECs, whether the mesenchymal ACE+ cells can also regulate the hematopoietic fate acquisition of aortic ACE+ cells is still questionable.

3. For the major concern 3, the authors have provided an additional clarification to resolve the concern.

4. For the major concern 4, the authors have provided additional bioinformatic analyses to resolve the concern.

5. For the major concern 5, the authors are supposed to perform additional bioinformatic analyses of published datasets to determine whether the expression of CD32-encoding gene, FCGR2B, is also detected in the HECs of mouse, zebrafish, or other vertebrate embryos. In fact, upon checking the expression of Fcgr2b in mouse AGM single-cell data (Left: PMID 32392346, Right: PMID 35414020), we observed a low expression in mouse EHT-related cells. This observation may suggest that this gene may not play a significant role in mouse EHT. Additionally, the absence of a homologous gene in zebrafish further supports the notion that this gene may only function in humans.

6. In addition, the newly added in silico perturbation and functional validation confirmed the role of CD32 in whole HEC subclusters. However, this contradicts the observation that the expression of FCGR2B was specifically higher in HEC subcluster 11 but not uniformly expressed in all HEC cells. In response to Question 3 from Reviewer #3, the author suggests that the claim about a uniform expression of FCGR2B not being observed in cluster 11 could be due to drop out events commonly encountered in single-cell analyses. This claim maybe not correct. To address this point, they should consider data imputation or utilize other sequencing technologies, such as Smart-seq2, to directly address this question and also avoid misleading.

Minor concerns:

1. There is an inconsistency in the use of abbreviation of single-cell RNA sequencing (scRNA-seq) as noted in lines 228 and 234, and mixed use of CD32 or FCGR2B expression throughout the manuscript as well.

2. In Extended Data Figure 4b of revised MS, using the Dotplot in Seurat to visualize the gene expression between two groups may cause misleading interpretations. For instance, genes like GJA5, MECOM, HOXA9, and CD44 may appear to have a larger difference than they actually do. To address this concern, the authors may consider using violin plots to display these results, as they provide a more accurate representation of the distribution and potential overstatement of differences between the two groups.

Reviewer #3 (Remarks to the Author):

In the revised manuscript, the authors have addressed my comments in a satisfactory manner. Notably, the authors have used the CD32 marker to identify the role of BMP and ROCK pathways in hematopoietic specification, and provide evidence that CD32 itself is required for hematopoiesis. Although the authors were not able to establish that CD32+ HEC is the precursor of HSCs in vivo, it is indeed a challenging experiment, and their collective results argue that CD32 is likely an obligatory stage of hematopoietic ontogeny. I have no additional comments.

ADDITIONAL COMMENTS TO REMAINING ISSUES RAISED BY REVIEWER #2

- For their point #1: I am not convinced that this experiment would demonstrate the utility and fidelity of CD32 as a marker of HSC-competent HE. This system relies on ectopic expression of transcription factors to enforce the HSC potential in non-HSC-competent iPSC-derived HE. It would not test whether CD32 marks HSC-competent HE during normal hematopoietic development.
- For their point #2: given that the staining pattern at 27 dpf in the Figure 1a is not representative, I agree it sounds reasonable that the authors can revise it with a new panel. On the other hand, while I also agree that whether the mesenchymal ACE+ cells can also regulate the hematopoietic fate acquisition of aortic ACE+ cells remains unclear, the questions sounds somewhat peripheral and could be addressed in the text.
- For their point #5: It seems reasonable for the authors to include some information about CD32/Fcgr2b expression in the mouse AGM based on the published datasets. I agree that functional validation in the mouse system is beyond the scope.
- For their point #6: I agree with reviewer #2 that the claim about dropout in single cell data (Fig. S6c) may or may not be accurate. I am skeptical that other technologies e.g., Smart-seq2 would provide higher coverage to resolve this question. Ultimately, the data in Fig. 4 show that CD32neg cells have some residual hematopoietic potential, however this does not undermine the authors' claim that CD32+ is a near-obligatory marker of late HEC. The issue of whether or not CD32+ late HE is an obligatory stage in hematopoietic ontogeny should be raised in the discussion.

Decision Letter, final checks:

Our ref: NCB-A50996A

8th February 2024

Dear Dr. Ditadi,

Thank you for your patience as we've prepared the guidelines for final submission of your Nature Cell Biology manuscript, "CD32 allows capturing blood cells emergence in slow motion during human embryonic development" (NCB-A50996A). Please carefully follow the step-by-step instructions provided in the attached file, and add a response in each row of the table to indicate the changes that you have made. Ensuring that each point is addressed will help to ensure that your revised manuscript can be swiftly handed over to our production team.

In recognition of the time and expertise our reviewers provide to Nature Cell Biology's editorial process, we would like to formally acknowledge their contribution to the external peer review of your manuscript entitled "CD32 allows capturing blood cells emergence in slow motion during human embryonic development". For those reviewers who give their assent, we will be publishing their names alongside the published article.

Nature Cell Biology offers a Transparent Peer Review option for new original research manuscripts submitted after December 1st, 2019. As part of this initiative, we encourage our authors to support increased transparency into the peer review process by agreeing to have the reviewer comments, author rebuttal letters, and editorial decision letters published as a Supplementary item. When you submit your final files please clearly state in your cover letter whether or not you would like to participate in this initiative. Please note that failure to state your preference will result in delays in accepting your manuscript for publication.

Cover suggestions

COVER ARTWORK: We welcome submissions of artwork for consideration for our cover. For more information, please see our guide for cover artwork.

Nature Cell Biology has now transitioned to a unified Rights Collection system which will allow our Author Services team to quickly and easily collect the rights and permissions required to publish your work. Approximately 10 days after your paper is formally accepted, you will receive an email in providing you with a link to complete the grant of rights. If your paper is eligible for Open Access, our Author Services team will also be in touch regarding any additional information that may be required to arrange payment for your article.

Please note that *Nature Cell Biology* is a Transformative Journal (TJ). Authors may publish their research with us through the traditional subscription access route or make their paper immediately

open access through payment of an article-processing charge (APC). Authors will not be required to make a final decision about access to their article until it has been accepted. Find out more about Transformative Journals

Please use the following link for uploading these materials:
[Redacted]

Best regards,

Kendra Donahue
Staff
Nature Cell Biology

On behalf of

Stylios Lefkopoulos, PhD
He/him/his
Senior Editor, Nature Cell Biology
Springer Nature
Heidelberger Platz 3, 14197 Berlin, Germany

E-mail: stylios.lefkopoulos@springernature.com
Twitter: [@s_lefkopoulos](https://twitter.com/s_lefkopoulos)
LinkedIn: [linkedin.com/in/stylios-lefkopoulos-81b007a0](https://www.linkedin.com/in/stylios-lefkopoulos-81b007a0)

Reviewer #1:

Remarks to the Author:

I wanted to give credit to the authors for experimentally addressing all of my comments. The manuscript is a lot more complete now, and the new data add robustness to the manuscript's main findings. I think the ms will be of significant impact to the human hematopoietic community.

I just had a couple of comments that could be addressed simply by changes in the text:

1) I am still somewhat confused about what Cd32 is marking. The authors describe in the text that 'CD32 expression in day 8 WNTd CD34+CD43negCD73negCD184negDLL4neg cells demarcates endothelial cells that do not express RUNX1C-EGFP'. This would suggest that in this context CD32 precedes RUNX1 expression. However, in the text referring to the other figures and in their model in figure 4 the authors clearly show that CD32 is expressed in a committed HEC (which express RUNX1). Which one is it? I think the authors need to be very careful with they wording here.

2)The ms would benefit from having a better description of the CD32KD cell line as far as the different populations that are altered in their differentiation protocols. Is the KD simply affection CD45 expression or is EC differentiation affected as well?

Reviewer #2:

Remarks to the Author:

Major concerns:

1. For the major concern 1, the authors explain that xenograft assay is not suited for HECs and that the culture condition for HSC specification from HECs is not established in vitro. Referring to a previous study by Daley's lab (Sugimura et al. Nature. 2017. PMID: 28514439), the injection of human PSC-derived transgenic HECs into NSG mice enables to evaluate multi-lineage hematopoietic chimerism. We suggest that the authors can apply the transplantation assay to demonstrate whether application of CD32 can capture HSPC-competent HECs.

2. For the major concern 2, given that the staining pattern at 27 dpf in the Figure 1a is not representative, we suggest that the authors can revise it with a new panel. Although the authors have explained that the ACE+ cells in the subaortic mesenchyme are likely precursors of aortic ACE+ HECs, whether the mesenchymal ACE+ cells can also regulate the hematopoietic fate acquisition of aortic ACE+ cells is still questionable.

3. For the major concern 3, the authors have provided an additional clarification to resolve the concern.

4. For the major concern 4, the authors have provided additional bioinformatic analyses to resolve the concern.

5. For the major concern 5, the authors are supposed to perform additional bioinformatic analyses of published datasets to determine whether the expression of CD32-encoding gene, FCGR2B, is also detected in the HECs of mouse, zebrafish, or other vertebrate embryos. In fact, upon checking the expression of Fcgr2b in mouse AGM single-cell data (Left:PMID 32392346,Right: PMID 35414020), we observed a low expression in mouse EHT-related cells. This observation may suggest that this gene may not play a significant role in mouse EHT. Additionally, the absence of a homologous gene in

zebrafish further supports the notion that this gene may only function in humans.

6. In addition, the newly added in silico perturbation and functional validation confirmed the role of CD32 in whole HEC subclusters. However, this contradicts the observation that the expression of FCGR2B was specifically higher in HEC subcluster 11 but not uniformly expressed in all HEC cells. In response to Question 3 from Reviewer #3, the author suggests that the claim about a uniform expression of FCGR2B not being observed in cluster 11 could be due to drop out events commonly encountered in single-cell analyses. This claim maybe not correct. To address this point, they should consider data imputation or utilize other sequencing technologies, such as Smart-seq2, to directly address this question and also avoid misleading.

Minor concerns:

1. There is an inconsistency in the use of abbreviation of single-cell RNA sequencing (scRNA-seq) as noted in lines 228 and 234, and mixed use of CD32 or FCGR2B expression throughout the manuscript as well.
2. In Extended Data Figure 4b of revised MS, using the Dotplot in Seurat to visualize the gene expression between two groups may cause misleading interpretations. For instance, genes like GJA5, MECOM, HOXA9, and CD44 may appear to have a larger difference than they actually do. To address this concern, the authors may consider using violin plots to display these results, as they provide a more accurate representation of the distribution and potential overstatement of differences between the two groups.

Reviewer #3:

Remarks to the Author:

In the revised manuscript, the authors have addressed my comments in a satisfactory manner. Notably, the authors have used the CD32 marker to identify the role of BMP and ROCK pathways in hematopoietic specification, and provide evidence that CD32 itself is required for hematopoiesis. Although the authors were not able to establish that CD32+ HEC is the precursor of HSCs in vivo, it is indeed a challenging experiment, and their collective results argue that CD32 is likely an obligatory stage of hematopoietic ontogeny. I have no additional comments.

ADDITIONAL COMMENTS TO REMAINING ISSUES RAISED BY REVIEWER #2

- For their point #1: I am not convinced that this experiment would demonstrate the utility and fidelity of CD32 as a marker of HSC-competent HE. This system relies on ectopic expression of transcription factors to enforce the HSC potential in non-HSC-competent iPSC-derived HE. It would not test whether CD32 marks HSC-competent HE during normal hematopoietic development.
- For their point #2: given that the staining pattern at 27 dpf in the Figure 1a is not representative, I agree it sounds reasonable that the authors can revise it with a new panel. On the other hand, while I also agree that whether the mesenchymal ACE+ cells can also regulate the hematopoietic fate acquisition of aortic ACE+ cells remains unclear, the questions sounds somewhat peripheral and could be addressed in the text.
- For their point #5: It seems reasonable for the authors to include some information about CD32/Fcgr2b expression in the mouse AGM based on the published datasets. I agree that functional

validation in the mouse system is beyond the scope.

- For their point #6: I agree with reviewer #2 that the claim about dropout in single cell data (Fig. S6c) may or may not be accurate. I am skeptical that other technologies e.g., Smart-seq2 would provide higher coverage to resolve this question. Ultimately, the data in Fig. 4 show that CD32neg cells have some residual hematopoietic potential, however this does not undermine the authors' claim that CD32+ is a near-obligatory marker of late HEC. The issue of whether or not CD32+ late HE is an obligatory stage in hematopoietic ontogeny should be raised in the discussion.

Author Rebuttal, first revision:

Dear Andrea,

Thank you for submitting your revised manuscript "CD32 allows capturing blood cells emergence in slow motion during human embryonic development" (NCB-A50996A). It has now been seen by the original referees and their comments are below. As you will see by going through the comments, reviewer #2 continued to raise some concerns and we therefore discussed these concerns with reviewer #3, hence the delay in getting back to you, which I sincerely apologize for. You can find the view of reviewer #3 on the remaining concerns raised by reviewer #2 in the section of their report "ADDITIONAL COMMENTS TO REMAINING ISSUES RAISED BY REVIEWER #2". Taking everything into account, the paper has improved in revision, and therefore we'll be happy in principle to publish it in Nature Cell Biology, pending minor revisions to satisfy the referees' final requests and to comply with our editorial and formatting guidelines.

We are thrilled to learn that Nature Cell Biology is interested in our study. In this revised version of the manuscript, we have addressed all remaining reviewer comments. Below in red, you can find our point-by-point response.

- address the remaining points by referee #1

1) I am still somewhat confused about what Cd32 is marking. The authors describe in the text that 'CD32 expression in day 8 WNTd CD34+CD43negCD73negCD184negDLL4neg cells demarcates endothelial cells that do not express RUNX1C-EGFP'. This would suggest that in this context CD32 precedes RUnx1 expression. However, in the text referring to the other figures and in their model in figure 4 the authors clearly show that CD32 is expressed in a committed HEC

(which express RUNX1). Which one is it? I think the authors need to be very careful with their wording here.

As we mentioned in our previous round of revisions, in vertebrates the pivotal regulator of developmental hematopoiesis Runx1 is transcribed from two promoters. These two promoters, the distal P1 and proximal P2, yield different Runx1 isoforms. The distal promoter P1 drives the expression of the RUNX1C isoform in HECs as they start to become hematopoietic progenitors and display hematopoietic characteristics, including the visible remodeling that leads to the characteristic “rounding” of emerging hematopoietic cells.

In our manuscript, the antibody used in Figure 2c binds to all the different human RUNX1 isoforms. In addition, the 3'-based RNA sequencing is unable to clearly distinguish these two isoforms. Instead, the hPSC reporter line used for Figures 3c, 3d and 3f is specific for the *RUNX1C* isoform controlled by the distal P1 promoter, i.e. the one expressed as HEC initiate the hematopoietic remodelling.

In light of our results, our model proposes that CD32 is upregulated in cells expressing RUNX1 isoforms under the control of the proximal P2 promoter. These cells will then progress to a *RUNX1C*⁺ state. In addition to the description on page 7 (line 140), we have now clarified this on page 14 (line 358-363).

2) The ms would benefit from having a better description of the CD32KD cell line as far as the different populations that are altered in their differentiation protocols. Is the KD simply affected CD45 expression or is EC differentiation affected as well?

The strategy we adopted to silence *FCRG2B* did not affect the generation of endothelial cells. We have now added a sentence in the result section on page 12 (line 277).

- address the minor comments by referee #2

1. There is an inconsistency in the use of abbreviation of single-cell RNA sequencing (scRNA-seq) as noted in lines 228 and 234, and mixed use of CD32 or FCGR2B expression throughout the manuscript as well.

We have carefully checked the manuscript and used *FCGR2B* when referring to gene or transcript and CD32 when referring to protein.

2. In Extended Data Figure 4b of revised MS, using the Dotplot in Seurat to visualize the gene expression between two groups may cause misleading interpretations. For instance, genes like GJA5, MECOM, HOXA9, and CD44 may appear to have a larger difference than they actually do. To address this concern, the authors may consider using violin plots to display these results, as they provide a more accurate representation of the distribution and potential overstatement of differences between the two groups.

We have now added the violin plots as requested, in Extended Data Figure 5a.

- address the first part of point #2 by referee #2: "given that the staining pattern at 27 dpf in the Figure 1a is not representative, we suggest that the authors can revise it with a new panel".

We are showing a different panel of a cross-section of a 27 dpf human embryo in the revised Figure 1a.

- *textually* address the second part of point #2 by referee #2 ("Although the authors have explained that the ACE+ cells in the subaortic mesenchyme are likely precursors of aortic ACE+ HECs, whether the mesenchymal ACE+ cells can also regulate the hematopoietic fate acquisition of aortic ACE+ cells is still questionable"), as suggested by referee #3: "On the other hand, while I also agree that whether the mesenchymal ACE+ cells can also regulate the hematopoietic fate acquisition of aortic ACE+ cells remains unclear, the questions sounds somewhat peripheral and could be addressed in the text".

This point is discussed on page 4 (line 72).

- *textually* address point #1 by referee #2 ("Referring to a previous study by Daley's lab (Sugimura et al. Nature. 2017. PMID: 28514439), the injection of human PSC-derived transgenic HECs into NSG mice enables to evaluate multi-lineage hematopoietic chimerism. We suggest that the authors can apply the transplantation assay to demonstrate whether application of CD32 can capture HSPC-competent HECs"), acknowledging the difficulty of these experiments and that further in vivo functional evidence are to be provided in future studies.

This aspect is addressed in the discussion on page 15 (line 377).

- *textually* address point #6 by referee #2 ("In addition, the newly added in silico perturbation and functional validation confirmed the role of CD32 in whole HEC subclusters. However, this contradicts the observation that the expression of FCGR2B was specifically higher in HEC subcluster 11 but not uniformly expressed in all HEC cells. In response to Question 3 from Reviewer #3, the author suggests that the claim about a uniform expression of FCGR2B not being observed in cluster 11 could be due to drop out events commonly encountered in single-cell analyses. This claim maybe not correct. To address this point, they should consider data imputation or utilize other sequencing technologies, such as Smart-seq2, to directly address this question and also avoid misleading"), as suggested by referee #3: "The issue of whether or not CD32+ late HE is an obligatory stage in hematopoietic ontogeny should be raised in the discussion"

This aspect is addressed in the discussion on page 15 (line 378).

- On the contrary, as also indicated in the previous round of review, while we find insightful the comment by referee #2 regarding potential conservation (point #5), we consider it beyond the scope of this study and addressing it will not be necessary.

Final Decision Letter:

Dear Andrea,

I am pleased to inform you that your manuscript, "CD32 captures committed haemogenic endothelial cells during human embryonic development", has now been accepted for publication in *Nature Cell Biology*. Congratulations to you and all the authors!

Over the next few weeks, your paper will be copyedited to ensure that it conforms to *Nature Cell Biology* style. Once your paper is typeset, you will receive an email with a link to choose the appropriate publishing options for your paper and our Author Services team will be in touch regarding any additional information that may be required.

Publication is conditional on the manuscript not being published elsewhere and on there being no announcement of this work to any media outlet until the online publication date in *Nature Cell Biology*.

Please note that *Nature Cell Biology* is a Transformative Journal (TJ). Authors may publish their research with us through the traditional subscription access route or make their paper immediately

open access through payment of an article-processing charge (APC). Authors will not be required to make a final decision about access to their article until it has been accepted. Find out more about Transformative Journals

If you have not already done so, we strongly recommend that you upload the step-by-step protocols used in this manuscript to the Protocol Exchange (www.nature.com/protocolexchange), an open online resource established by Nature Protocols that allows researchers to share their detailed experimental know-how. All uploaded protocols are made freely available, assigned DOIs for ease of citation and are fully searchable through nature.com. Protocols and Nature Portfolio journal papers in which they are used can be linked to one another, and this link is clearly and prominently visible in the online versions of both papers. Authors who performed the specific experiments can act as primary authors for the Protocol as they will be best placed to share the methodology details, but the Corresponding Author of the present research paper should be included as one of the authors. By uploading your Protocols to Protocol Exchange, you are enabling researchers to more readily reproduce or adapt the methodology you use, as well as increasing the visibility of your protocols and papers. You can also establish a dedicated page to collect your lab Protocols. Further information can be found at www.nature.com/protocolexchange/about

With kind regards,
Stelios

Stylianos Lefkopoulos, PhD
He/him/his

Senior Editor, Nature Cell Biology
Springer Nature
Heidelberger Platz 3, 14197 Berlin, Germany

E-mail: stylianos.lefkopoulos@springernature.com
Twitter: @s_lefkopoulos
LinkedIn: [linkedin.com/in/stylianos-lefkopoulos-81b007a0](https://www.linkedin.com/in/stylianos-lefkopoulos-81b007a0)
